

# Environmental conditions in the North Atlantic sector of the Arctic during the HALO–(AC)³ campaign

Andreas Walbröl[1], Janosch Michaelis[2], Sebastian Becker[3], Henning Dorff[4], Irina Gorodetskaya[5,6], Benjamin Kirbus[3], Melanie Lauer[1], Nina Maherndl[3], Marion Maturilli[7], Johanna Mayer[8], Hanno Müller[3], Roel A. J. Neggers[1], Fiona M. Paulus[1], Johannes Röttenbacher[3], Janna E. Rückert[9], Imke Schirmacher[1], Nils Slättberg[7], André Ehrlich[3], Manfred Wendisch[3], and Susanne Crewell[1]

[1]Institute for Geophysics and Meteorology, University of Cologne, Cologne, Germany
[2]Alfred Wegener Institute, Helmholtz Center for Polar and Marine Research, Bremerhaven, Germany
[3]Leipzig Institute for Meteorology, Leipzig University, Leipzig, Germany
[4]Meteorological Institute of Hamburg, University of Hamburg, Hamburg, Germany
[5]Centre for Environmental and Marine Studies, University of Aveiro, Aveiro, Portugal
[6]Interdisciplinary Centre of Marine and Environmental Research (CIIMAR), University of Porto, Matosinhos, Portugal
[7]Alfred-Wegener-Institut, Helmholtz Center for Polar and Marine Research, Potsdam, Germany
[8]German Aerospace Centre (Deutsches Zentrum für Luft- und Raumfahrt (DLR)), Oberpfaffenhofen, Germany
[9]Institute of Environmental Physics, University of Bremen, Bremen, Germany

**Correspondence:** Andreas Walbröl (a.walbroel@uni-koeln.de)

**Abstract.**

The airborne field campaign HALO–(AC)³ took place from 07 March to 12 April 2022 and was designed to observe the transformation of air masses during their meridional transport in the North Atlantic sector of the Arctic. We evaluate the meteorological and sea ice conditions during the campaign based on the European Centre for Medium–Range Weather Forecasts (ECMWF) Reanalysis v5 (ERA5), satellite data, and atmospheric soundings with respect to climatology and describe special synoptic events. HALO–(AC)³ started with a warm period (11–20 March) where strong southerly winds prevailed that caused moist and warm air intrusions (MWAIs). Two MWAIs were detected as Atmospheric Rivers (ARs). Compared to the ERA5 climatology (1979–2022), record breaking vertically integrated poleward heat and moisture fluxes averaged over 75.0–81.5° N were found. The related warm and moist air masses reached the central Arctic, causing the highest rainfall rates over the sea ice northwest of Svalbard recorded since the beginning of the ERA5 climatology. Subsequently, the cold period of HALO–(AC)³ started after the passage of a Shapiro–Keyser cyclone on 21 March when the wind regime turned to northerlies, advecting colder air into the Fram Strait. Until 08 April, marine cold air outbreaks (MCAOs) prevailed, including two strong MCAO events on 21–26 March and 01–02 April. In between, aged subpolar warm air was advected to the Fram Strait with northeasterly and easterly winds. On average, the campaign period was warmer than the climatology, especially due to the exceptionally strong ARs during the warm period. In the Fram Strait, the sea ice concentration (SIC) was within the 10–90th percentiles of the climatology over the entire campaign duration. During the warm period, SIC was strongly reduced and an untypically large polynya for this season opened north of Svalbard. During the cold period, the polynya was closed again and above average SICs were found. We describe the environmental conditions of a Polar Low and far north reaching cirrus in detail, which will be subjects of research using HALO's remote sensing suite in the future. We also present the perspective on the HALO–(AC)³



weather conditions from the research site Ny–Ålesund, where orographic effects caused by the Svalbard archipelago and temporal shifts of atmospheric signals due to the propagation of synoptic systems must be taken into account. Overall, our study may serve as basis for future analyses of the data collected during HALO–(AC)³ and to compare synoptic conditions with other field campaigns.



## 1 Introduction

The Arctic experiences a drastic temperature increase almost 4 times stronger compared to the rest of the globe (Rantanen et al., 2022). This enhanced warming is known as Arctic Amplification and is caused by several feedback mechanisms (Serreze et al., 2009; Screen and Simmonds, 2010; Serreze and Barry, 2011). While the contribution of some feedback mechanisms to Arctic Amplification can be regarded as scientific consensus (i.e., sea ice–albedo feedback, Serreze et al., 2009), others are not yet sufficiently explored and their contribution is uncertain (i.e., influence of clouds, lapse rate feedback, linkages between

Arctic and mid–latitudes, Wendisch et al., 2023). As the Arctic warms faster than the mid–latitudes, the meridional temperature gradient decreases, likely resulting in a weaker jet stream with enhanced tendency to meander (more southward and northward excursions, Francis and Vavrus, 2015). Consequently, the frequency of poleward moist and warm air intrusions (MWAIs) and southward cold air outbreaks (CAOs) is expected to increase (Pithan et al., 2018; Mewes and Jacobi, 2019). You et al. (2022) found a positive trend in the frequency and duration of atmospheric blocking over the Barents Sea especially in winter. As

blocking situations are associated with meandering jets, both frequency and duration of MWAIs reaching the Arctic through the North Atlantic sector are increasing.

MWAIs frequently transport large amounts of heat and moisture into the Arctic through the Atlantic sector (Woods and Caballero, 2016). When the warm air glides onto the cold Arctic dome, deep cloud systems including mixed–phase clouds form. Woods et al. (2013) and Woods and Caballero (2016) stress that intense MWAIs have a large effect on the downward

terrestrial radiation at the surface, contributing to the warming of the Arctic. Such MWAIs precondition the sea ice for the melting season, resulting in lower sea ice extent at the end of the summer (Kapsch et al., 2013, 2019). Numerical models still struggle to represent the mixed–phase clouds and the transformation processes of the air masses well (Pithan et al., 2014; Cohen et al., 2020). MWAIs can often be linked with filamentary bands of extreme moisture transport, known as Atmospheric Rivers (Newell et al., 1992). Atmospheric Rivers (ARs) are responsible for over 90 % of the poleward moisture transport across the

mid–latitudes and are frequently accompanied by extreme winds and precipitation (Nash et al., 2018). In the Arctic, ARs can cause snow accumulation, as well as the opposite, melting of snow and sea ice due to enhanced downward terrestrial radiation from the clouds (Neff et al., 2014; Komatsu et al., 2018; Mattingly et al., 2018, 2020; Bresson et al., 2022; Viceto et al., 2022). In a warming climate, ARs are expected to shift polewards and intensify due to the increased moisture load (Ma et al., 2020).

CAOs are often responsible for severe weather events in the high– and mid–latitudes and mainly occur in winter and spring

(Fletcher et al., 2016; Pithan et al., 2018). Here, we focus on marine CAOs (MCAOs) where cold and dry air is advected southwards from the sea ice covered Arctic ocean over the ice–free warm waters of the North Atlantic (Fram Strait, Greenland and Norwegian Seas). The strong temperature contrast between the ocean and the lower tropospheric air leads to intense heat releases (turbulent fluxes of sensible and latent heat) that are responsible for 60–80 % of the oceanic heat losses in that region (Papritz and Spengler, 2017). In the destabilized atmospheric boundary layer, cloud streets form and develop into open cells

as the air mass moves southwards and takes up heat and moisture. This cloud evolution is not well represented in models but it is an important feature for the performance of climate models (Pithan et al., 2018). MCAOs can also cause the formation of Arctic mesoscale cyclones (Polar Lows, Rasmussen and Turner, 2003; Zahn and von Storch, 2008), which may locally lead to



heavy precipitation and strong winds. The formation processes of Polar Lows and their relation to MCAOs are still not fully understood, partly due to the sparsity of measurements in the Arctic (Moreno-Ibáñez et al., 2021).

So far, observations of air mass transformations in the Arctic have mostly been conducted from a fixed local position (Eulerian view). Only a few aircraft based samplings of air mass properties over a limited regional area have been reported (i.e., Wendisch et al., 2019; Mech et al., 2022). The Eulerian point of view does not permit the required observations of air mass transformation processes along their meridional pathway. Therefore, following Pithan et al. (2018), a quasi–Lagrange approach following air masses to and from the Arctic motivated the field campaign HALO–(AC)[3] within the Transregional

Collaborative Research Center TRR 172 (TRR 172) on Arctic Amplification: Climate Relevant Atmospheric and Surface Processes and Feedback Mechanisms (AC)[3]. The campaign was designed to obtain quasi–Lagrange observational data of air mass transformations during MWAIs and MCAOs to gain process understanding and evaluate the performanceof weather and climate models (Wendisch et al., 2021).

During HALO–(AC)[3], extensive remote sensing and in situ measurements have been performed between the Norwegian
Sea and the North Pole with three research aircraft between 11 March and 12 April 2022. The High Altitude and LOng-range research aircraft (HALO) operated by the German Aerospace Center (DLR, Ziereis and Gläßer, 2006; Stevens et al., 2019) and was based in Kiruna during HALO–(AC)[3]. HALO is a modified Gulfstream G550, which has a sufficient operating range (9000 km in altitudes up to 15 km) for quasi–Lagrange air mass observations. For HALO–(AC)[3], it was equipped with a similar instrumental payload as during the EUREC4A campaign (Stevens et al., 2019; Konow et al., 2021). The research aircraft Polar
5 and Polar 6 (P5 and P6) were based in Longyearbyen and operated by the Alfred Wegener Institute, Helmholtz Centre for Polar and Marine Research (AWI, Wesche et al., 2016) in a similar coordinated way as during the ACLOUD campaign (Wendisch et al., 2019).

In this study, we focus on the analysis of the environmental conditions during HALO–(AC)[3] and their climatological context to serve as comprehensive reference in future studies. After presenting the data and methods in Sect. 2, we describe the two
dominating periods and the classification of certain synoptic events based on meteorological parameters in Sect. 3.1. We also provide a view on the conditions from Ny–Ålesund (Sect. 3.2), where future studies will use the extensive measurements from the French–German Atmospheric Observatory AWIPEV. Sea ice conditions in the North Atlantic sector of the Arctic are presented in Sect. 3.3. We further analyse the origin and strength of the MWAIs and ARs (Sect. 4.1), the conditions during the MCAO periods (Sect. 4.2), the formation and environmental characteristics of the Polar Low (Sect. 4.3), and Arctic cirrus
clouds (Sect. 4.4), before concluding our study in Sect. 5.



**Table 1.** Geographical coordinates for measurement regions as displayed in Fig. 1. Note that the northern part of the northern region in the central Arctic is extended to the west.

|  | Northern region | Central region | Southern region |
|---|---|---|---|
| Northern limit in deg N | 89.3 (84.5) | 81.5 | 75.0 |
| Southern limit in deg N | 81.5 | 75.0 | 70.6 |
| Western limit in deg E | -54.0 (-9.0) | -9.0 | 0.0 |
| Eastern limit in deg E | 30.0 | 16.0 | 23.0 |

## 2 Data and Methods

### 2.1 Study domain

As the research flights during HALO–(AC)[3] were conducted in the Norwegian, Greenland, and Barents Seas, the Fram Strait, and central Arctic Ocean in early spring, we also confined our analysis to this region. Both region and season for this campaign

were selected because the largest fraction of MWAIs enters the Arctic through the North Atlantic during winter and early spring (Johansson et al., 2017; You et al., 2022). Based on the flight tracks, illustrated in Fig. 1, we selected three domains for which averages of several meteorological variables were calculated in this study. Table 1 summarizes the geographical coordinates of the domains. From north to south the domains are referred to as northern region (central Arctic north of Svalbard and Greenland), central region (Fram Strait and Greenland Sea), and southern region (Greenland and Norwegian Seas between

Svalbard and Norway). A particular focus lies on the central region as most flights were carried out in this domain. Within the central region, flight activities of P5 and P6 remained within 500 km of the northwestern coast of Svalbard due to their limited operating range. HALO covered the entire pathway of MWAIs and MCAOs between Greenland, Svalbard and Norway up to the central Arctic.

### 2.2 Measurements from HALO and at Ny–Ålesund

HALO is equipped with a dropsonde launch system (Airborne Vertical Atmospheric Profiling System, Hock and Franklin, 1999) and RD94 dropsondes were used to obtain profiles of temperature, relative humidity, and pressure with an accuracy of 0.2 K, 2 %, and 0.4 hPa, respectively, between the aircraft flight altitude and the surface (Vaisala, a; George et al., 2021). Wind profiles are derived from the temporal change of the dropsonde's position measured by the GPS receiver.

Measurements gathered at the Atmospheric Observatory of the French–German AWIPEV research base in Ny–Ålesund
(Svalbard, 78.92° N, 11.92° E) provide an additional view on the environmental conditions during HALO–(AC)[3] in the central region. At AWIPEV, ground based meteorological measurements (Maturilli et al., 2013; Maturilli, 2020a) are complemented by vertical profiles of wind, temperature, pressure, and relative humidity from daily radiosondes (Maturilli and Kayser, 2016, 2017; Maturilli, 2020b). The currently used Vaisala RS41 radiosondes have uncertainties of temperature, pressure, and relative humidity measurements of 0.2–0.4 K, 0.6–1.0 hPa, and 3–4 %, respectively (Vaisala, b). During the campaign period,



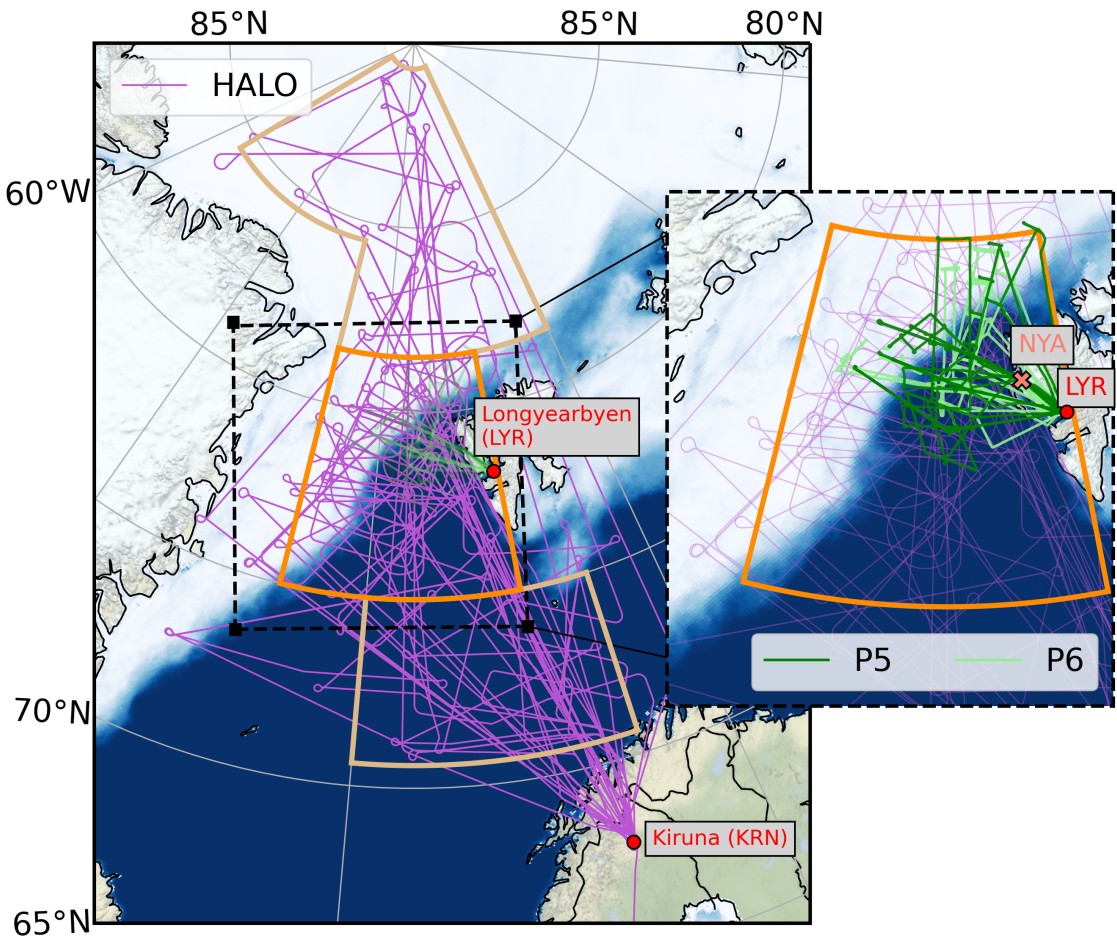

**Figure 1.** Flight tracks of HALO (purple), as well as P5 and P6 (dark and light green, zoomed domain in the right panel) during HALO–$(AC)^3$. For orientation, the mean sea ice concentration from 07 March to 12 April 2022 based on AMSR2 satellite data (Spreen et al., 2008) is showed as filled contours. The three main measurement regions (southern region, central region, and northern region) are marked as yellow and orange boxes. The central region is highlighted.

the launch frequency of the radiosondes was increased to every 6 hours. From these vertical profiles, we identified the thermal tropopause according to the WMO definition as the lowest level at which the temperature lapse rate falls below $2\,\mathrm{K\,km^{-1}}$ and does not exceed this value for the next $2\,\mathrm{km}$. The integrated water vapour (IWV) was calculated from profiles of air pressure and specific humidity, which was derived from the temperature and relative humidity measurements. In addition to the radiosonde data, we used data from the CL–51 ceilometer (Maturilli and Ebell, 2018; Maturilli, 2022) to assess the cloud conditions at Ny–Ålesund.



## 2.3 Satellite observations of sea ice

We focused on sea ice concentration (SIC), i.e., the percentage of a satellite pixel (grid cell in ERA5) covered by sea ice, derived from satellite measurements to describe the sea ice conditions during HALO–(AC)[3]. In general, we separated observations for areas with and without sea ice (latter is also termed open water) through the threshold of 15 % SIC. Starting with the launch of the Numbus–7 satellite in 1978, passive microwave radiometers from space provide a long–term observation of sea ice (Spreen and Kern, 2016). To put SIC during HALO–(AC)[3] into a climatological context, we used the OSI SAF Global Sea Ice Concentration Climate Data Record (SIC CDR v2.0), namely the product OSI–450 from 1979 to 2015, and the complementary Interim Climate Data Record OSI–430–b from 2016 onwards (OSI SAF, 2017; Copernicus Climate Change Service (C3S), 2020). It is based on satellite microwave data from a series of instruments: SMMR (1979–1987), SSM/I (1987–2008), and SSMIS (2006–today). The dynamic hybrid algorithm is described in Lavergne et al. (2019) and it processes frequency channels at 19 and 37 GHz. The datasets are available daily with a grid spacing of 25 km×25 km.

To study the evolution of SIC during HALO–(AC)[3] at a higher spatial resolution, we used the merged MODIS–AMSR2 SIC dataset, available daily at 1 km resolution. It is based on passive microwave (AMSR2 sensor on GCOM–W1 (JAXA)) and thermal infrared (MODIS sensor on AQUA (NASA)) satellite data. The merged product (Ludwig et al., 2020) takes advantage of the high spatial resolution of thermal infrared, where SIC computed from temperature anomalies (Drüe and Heinemann, 2004), and of the independence of cloud cover of microwaves. SIC is computed from polarization differences at 89 GHz between open ocean and sea ice (ASI algorithm, Spreen et al., 2008).

## 2.4 Reanalysis data

For the analysis of the weather development, we used the ERA5 reanalysis from the European Centre for Medium–Range Weather Forecasts (ECMWF), which has a horizontal resolution of 31 km with 137 vertical model levels and offers hourly data output from 1950 until present (Hersbach et al., 2018a, b, 2020). Climatological comparisons are based on the years 1979–2022 to cover the inter–annual variability of atmospheric conditions, unless specified differently. In the following, we describe the processing of ERA5 data and derivation of indices calculated for our analysis.

### 2.4.1 Basic meteorological parameters

Daily means of basic meteorological variables, such as the mean sea level pressure, 2 m air temperature, 10 m wind speed, IWV, total daily precipitation (in liquid and frozen form), were calculated and averaged for each of the three measurement regions for the description of the synoptic evolution. Grid points with a land fraction $\geq 0$ were excluded. We respected the increasing data point density with increasing latitudes (due to the $0.25° \times 0.25°$ grid) by computing regional averages weighted by the cosine of latitudes. Additionally, we computed averages of vertically integrated meridional heat (IHT$_{\text{north}}$) and moisture flux (IVT$_{\text{north}}$) over the ocean grid points of the central region latitudes for information about the time and location of MWAIs.

We assessed the weather conditions during HALO–(AC)[3] with respect to the climatological reference period with anomalies of temperature at 850 hPa and 2 m height, 500 hPa geopotential height, and IWV. Evaluating the temperature anomalies at



these two levels reveals the vertical extent of the temperature anomalies in the lower troposphere. We computed means over the climatology years of these variables for three (sub–)periods of HALO–(AC)³: 11–20 March, 21 March–12 April, 07 March–12 April. We obtained the anomalies by subtracting the long–term means from the mean of the respective time period in 2022 so that positive (negative) values correspond to anomalies above (below) the long–term mean.

### 2.4.2 Detection of Atmospheric Rivers and moist and warm air intrusions

The northward component of vertically integrated water vapour transport ($IVT_{north}$) is used to detect periods of increased poleward water vapour transport. Woods and Caballero (2016) defined moist air intrusions into the Arctic using $IVT_{north}$ at $70°$ N above certain threshold ($200 \, \mathrm{Tg \, d^{-1} \, deg^{-1}}$) over a duration of at least 1.5 days and a zonal extent of at least $9°$. Here, we adapted this definition to our campaign site and defined MWAIs with positive daily means of $IVT_{north}$ averaged over the central region. An MWAI is considered weak when this variable does not exceed $100 \, \mathrm{kg \, m^{-1} \, s^{-1}}$. Values above this threshold suggest strong MWAI conditions. Additionally, we computed the climatological mean, as well as the 25–75$^{th}$ and 10–90$^{th}$ percentiles over the climatology years for each day.

In this study, ARs were identified with a global algorithm by Guan and Waliser (2015) in its revised version (Guan et al., 2018). IVT must exceed the 85$^{th}$ percentile of the season and grid cell, and must reach an absolute value of $100 \, \mathrm{kg \, m^{-1} \, s^{-1}}$, while fulfilling geometrical requirements (length $> 2000 \, \mathrm{km}$, length to width ratio of $> 2$, IVT direction within $45°$ of the detected AR axis). To account for the lower moisture content in the polar atmosphere, the absolute IVT threshold is reduced to $50 \, \mathrm{kg \, m^{-1} \, s^{-1}}$. In case a detected AR at the 85$^{th}$ percentile threshold does not fulfill other requirements (e.g., related to its structure or geometry), higher IVT thresholds are successively tested (85–95$^{th}$ in steps of 2.5). This procedure ensures detection of an AR that is embedded in a wide area of enhanced moisture transport with respect to the local climatology.

### 2.4.3 Marine cold air outbreak index

A key feature of MCAOs are strong temperature decreases with increasing height especially in the lower troposphere. It forms the basis for the MCAO index $M$ to identify them and to quantify their strength. The MCAO index $M$ was calculated following Papritz and Spengler (2017) and Dahlke et al. (2022):

$$M = \theta_{SKT} - \theta_{850} \tag{1}$$

with $\theta_{SKT}$ ($\theta_{850}$) as the potential skin temperature (potential temperature at $850 \, \mathrm{hPa}$). It was averaged over the central region, excluding grid points classified as land according to the ERA5 land–sea–mask. Furthermore, grid points with skin temperatures below $271.5 \, \mathrm{K}$ (e.g., over assumed sea ice) were also masked (Dahlke et al., 2022) and excluded from further processing. As for $IVT_{north}$, we computed the mean, 25–75$^{th}$ and 10–90$^{th}$ percentiles for the climatology.

Note that both Papritz and Spengler (2017) and Dahlke et al. (2022) defined the Fram Strait box slightly different for their calculations of $M$ ($75°$ N—$80°$ N, $10°$ W–-$10°$ E). Following Papritz and Spengler (2017), MCAO conditions are present when $M > 0 \, \mathrm{K}$ and its strength can be classified as weak ($0 \, \mathrm{K} < M \leq 4 \, \mathrm{K}$), moderate ($4 \, \mathrm{K} < M \leq 8 \, \mathrm{K}$), strong ($8 \, \mathrm{K} < M \leq 12 \, \mathrm{K}$) or very strong ($M > 12 \, \mathrm{K}$).



### 2.4.4 Lower tropospheric parameters for marine cold air outbreak analysis

For a detailed analysis of the MCAO conditions over the campaign (Sect. 4.2), we used hourly ERA5 data of wind, temperature, and relative humidity at 850, 925, and 1000 hPa, as well as cloud cover, precipitation and surface turbulent heat fluxes (sensible and latent). All grid points classified as land were neglected. All pressure level variables were temporally averaged over 6 hour intervals and spatially averaged over each of the three measurement regions. Similar averaging was applied to turbulent flux data, where we used the mean values over 1 hour at each grid point instead of the instantaneous values (Hersbach et al., 2020). Because our focus is on the intensity of the upward fluxes (positive sign) over open water during the MCAOs, we only considered the central and southern region, for which we applied spatial averaging neglecting sea ice surfaces (SIC $\geq 15\,\%$), respectively.

For low, medium and high–level cloud cover, we computed the median value including 5, 25, 75, and $95^{\text{th}}$ percentiles, also for 6 hour intervals. Moreover, cloud cover was confined to open water surfaces only (SIC $< 15\,\%$) as ERA5 tends to overestimate especially low–level cloud cover over sea ice surfaces (Di Biagio et al., 2021). For precipitation, we used the spatially averaged accumulated values over 6 hour intervals (only for the central and southern region), where we distinguished between open water and sea ice covered surfaces (SIC $< 15\,\%$ and SIC $\geq 15\,\%$, respectively). For precipitation in terms of snowfall, we further distinguished between large–scale and convective precipitation. The latter is defined in ERA5 as the frozen precipitation resulting from the convection scheme and thus from subgrid–scale convection. Total precipitation is then the sum of the convective and the large–scale part (Hersbach et al., 2020).

### 2.4.5 Environmental conditions of Polar Lows

We analyse the environment for Polar Low formation with a set of conditions (C1–C6) suggested by Radovan et al. (2019) and Terpstra et al. (2016): C1: sea surface temperature (SST) $-$ 500 hPa temperature, C2: SST $-$ 2 m temperature, C3: mean lapse rate below 850 hPa, $C4_{\text{i}}$: mean relative humidity below 950 hPa, $C4_{\text{ii}}$: mean relative humidity between 850 and 950 hPa, C5: 10 m wind speed (here, gust), C6: 500 hPa geopotential height anomaly.

Condition C1 is a measure of the convection potential as it evaluates the mean temperature gradient between the surface and the mid–troposphere. Large differences between the 2 m temperature and the SST (C2) promote strong sensible heat fluxes, stressing the air–sea interaction. Condition C3 is evaluated as the vertical mean of the vertical potential temperature gradient $\partial\theta/\partial z$ (with $\theta$ as potential temperature and $z$ as height) between the surface and the 850 hPa pressure level. Positive values of C3 represent an increase of potential temperature with height. At least conditionally unstable conditions (unstable (stable) for moisture–saturated (unsaturated) air parcels) should be present ($\partial\theta/\partial z \leq 3\,\text{K}\,\text{km}^{-1}$). The relative humidity conditions $C4_{\text{i}}$ and $C4_{\text{ii}}$ are an important measure for the availability of moisture for latent heat release, a major energy source of a Polar Low. Maximum 10 m wind speed (C5) is an indicator of the Polar Low's strength. The geopotential height condition (C6) is selected to ensure the presence of an upper level trough.

For conditions C1–C4, the $75^{\text{th}}$ percentile, for C5, the maximum, and for C6, the mean in a 200 km radius around the Polar Low's centre were computed. We decided to use the maximum 10 m wind gust instead of mean wind due to the coarse





**Table 2.** Criteria to evaluate environmental conditions of a Polar Low in a $200\,\mathrm{km}$ radius around its centre according to Radovan et al. (2019).

| Condition | Meaning | How evaluated | Threshold |
|---|---|---|---|
| C1 | Sea surface temperature $-\ 500\,\mathrm{hPa}$ temperature | $75^{\mathrm{th}}$ percentile | $\geq 43\,\mathrm{K}$ |
| C2 | Sea surface temperature $-\ 2\,\mathrm{m}$ temperature | $75^{\mathrm{th}}$ percentile | $\geq 6\,\mathrm{K}$ |
| C3 | Mean lapse rate below $850\,\mathrm{hPa}$ | $75^{\mathrm{th}}$ percentile | $\leq 3\,\mathrm{K\,km^{-1}}$ |
| C4$_{\mathrm{i}}$ | Mean relative humidity below $950\,\mathrm{hPa}$ | $75^{\mathrm{th}}$ percentile | $\geq 75\,\%$ |
| C4$_{\mathrm{ii}}$ | Mean relative humidity between 850 and $950\,\mathrm{hPa}$ | $75^{\mathrm{th}}$ percentile | $\geq 82\,\%$ |
| C5 | $10\,\mathrm{m}$ wind gust | maximum | $\geq 15\,\mathrm{m\,s^{-1}}$ |
| C6 | $500\,\mathrm{hPa}$ geopotential height anomaly | mean | $\leq -160\,\mathrm{gpm}$ |

resolution of ERA5. The climatology of the $500\,\mathrm{hPa}$ geopotential height for C6 consists of all days of April between 1979 and 2022 for each grid point within the $200\,\mathrm{km}$ radius. Subsequently, the geopotential height anomaly was computed as the

difference between the conditions during HALO–(AC)$^3$ and the climatology, and C6 was obtained by averaging over the $200\,\mathrm{km}$ circle. Table 2 summarizes the conditions and their thresholds (Rasmussen and Turner, 2003; Zahn and von Storch, 2008; Radovan et al., 2019).

Furthermore, we calculated vorticity from 10 dropsonde measurements taken within 1 hour arranged on a circular pattern. Following Lenschow et al. (2007) and Bony and Stevens (2019), we assumed horizontal variations in the wind field to be linear

and expanded the wind vector with taylor expansion around the centre of the circle. The wind vector $\boldsymbol{v}$ can be expressed as

$$\boldsymbol{v} = \boldsymbol{v_0} + \frac{\partial \boldsymbol{v}}{\partial \vartheta} + \frac{\partial \boldsymbol{v}}{\partial \phi} \tag{2}$$

for spherical coordinates $\vartheta$ and $\phi$ and the average wind velocity vector of all dropsondes $\boldsymbol{v_0}$. By performing least–squares regression, the horizontal gradients $\frac{\partial \boldsymbol{v}}{\partial \vartheta}$ and $\frac{\partial \boldsymbol{v}}{\partial \phi}$ were obtained to calculate the relative vorticity

$$\zeta = \frac{1}{r \sin \vartheta_0} \left( \frac{\partial v}{\partial \phi} - \frac{\partial u \sin \vartheta}{\partial \vartheta} \right). \tag{3}$$

**2.4.6 Trajectory calculations**

Trajectory calculations were performed using LAGRANTO (Sprenger and Wernli, 2015) and wind fields from ERA5 (Hersbach et al., 2020) retrieved on native resolution ($0.25° \times 0.25°$, 1 hour, 137 model levels). To assess the origin of sampled cirrus air masses, 72 hour back–trajectories for the northernmost sections of the flights on 11 and 12 April were computed. Trajectories were initialized at the pressure levels $350$–$550\,\mathrm{hPa}$ in $50\,\mathrm{hPa}$ steps, where the single–layer cirrus had been observed by HALO.

Furthermore, 72 hour back– and 48 hour forward–trajectories were computed to identify the pathways of the MWAIs (ARs) on 13 and 15 March. These trajectories were initialized along the $77.5°\,\mathrm{N}$ latitude with $1°$ zonal spacing between $20°\,\mathrm{W}$ and $13°\,\mathrm{E}$ at 18:00 UTC on the respective days at the pressure levels 700, 850, and $925\,\mathrm{hPa}$.



## 3 Meteorological and sea ice conditions during HALO–(AC)³

### 3.1 Analysis and identification of synoptic events

In this section, we characterize weather events during HALO–(AC)³ and assess their strength with climatological context. At first, we separate the campaign into two major periods based on IVT$_{north}$ and the MCAO index ($M$) in the central region in Fig. 2. The campaign started with a warm and humid period with positive IVT$_{north}$ and negative values of $M$, coinciding with enhanced static stability, from 11 to 20 March 2022. The warm period was followed by a cold and dry period from 21 March to 12 April 2022, where IVT$_{north}$ ($M$) was almost exclusively negative (positive).

Within the warm and cold periods, we classified a large variety of weather events based on IVT$_{north}$ and $M$, the large scale pressure constellation, and regional averages of basic meteorological variables from ERA5 (Fig. 3 and 4). IHT$_{north}$ and IVT$_{north}$, averaged over the central region latitudes, were used to spatially and temporally localize meridional heat and moisture transport corresponding with MWAIs and MCAOs (Fig. 5). For the warm, cold, and the entire HALO–(AC)³ period, we present anomalies of temperature at $850\,\mathrm{hPa}$ and $2\,\mathrm{m}$ above the surface, geopotential height at $500\,\mathrm{hPa}$, and IWV with respect to the
climatology (Fig. 6).

#### 3.1.1 Warm period

We start our analysis on 07 March to capture the initial conditions of HALO–(AC)³. Before the warm period started, southerly and southwesterly winds dominated over the southern and central regions (07–09 March 2022, Fig. 3d). The corresponding general circulation pattern was initially characterized by a high surface pressure system extending from the North Sea to
Baltic Sea and Scandinavia. The surface high was connected to a ridge in the $500\,\mathrm{hPa}$ geopotential height. Low surface pressure and low $500\,\mathrm{hPa}$ geopotential height were located over Greenland and the North Atlantic. A weak MWAI (IVT$_{north}$ < $100\,\mathrm{kg\,m^{-1}\,s^{-1}}$) associated with the southwesterly flow barely reached the central region on 08–09 March (Fig. 3a). No signal can be seen in the central Arctic regarding temperature and IWV increases during that time (Fig. 3c, e). Subsequently, on 10–11 March, a weak high pressure system at the surface and ridge in the $500\,\mathrm{hPa}$ geopotential height temporarily caused northerly
winds between two lows over the Fram Strait. The brief interruption of the southerly flow is reflected in the $2\,\mathrm{m}$ temperature drop (Fig. 3c) on 10–11 March.

At the beginning of the warm period (11–13 March), the high pressure ridge over Scandinavia intensified and established blocking (Omega Block) over most of Europe. Intense MWAIs are connected to blocking situations over the eastern border of a large basin (here, North Atlantic), redirecting cyclones northward as the typical eastward propagation is blocked (Woods
et al., 2013). Consequently, warm and moist air masses originating from the North Atlantic were transported towards our three main measurement areas, driven by several low pressure systems that formed between Iceland and eastern Greenland.

Besides the before mentioned weak MWAI event, we identified a much stronger event during the first week of the campaign with area–averaged IVT$_{north}$ up to $160\,\mathrm{kg\,m^{-1}\,s^{-1}}$. The event was detected as AR and passed through the measurement regions on 12–13 March. It was connected with a more pronounced and narrow meridional flow due to the intensified pressure contrast
between Scandinavia and Greenland (Fig. 4a). Exceptionally warm and moist air masses were transported from the British Isles



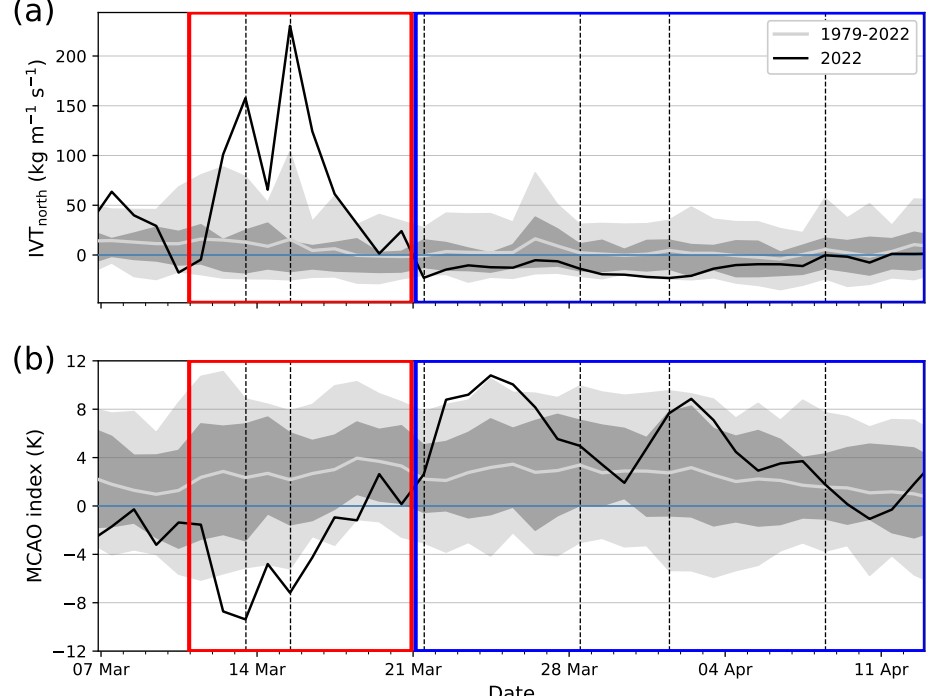

**Figure 2.** Daily means of (a) northwards component of integrated water vapour transport (IVT$_{north}$), (b) Marine Cold Air Outbreak (MCAO) index ($M$) based on ERA5 and averaged over the central region for HALO–(AC)$^3$ (black line). Daily values of the climatological mean (1979–2022, grey line), as well as the 10–90$^{th}$ and 25–75$^{th}$ percentiles are illustrated as light and dark grey shading. The red (blue) box indicates the warm (cold) period dominated by MWAIs (MCAOs). Vertical dashed black lines mark the days shown in Fig. 4.

over the Fram Strait towards the North Pole. As a result, equivalent potential temperature values at $850\,hPa$ in the northern region were elevated by up to $30\,K$ compared to later parts of the campaign (Fig. 4a). In the same region, IWV reached the maximum value of the campaign with an area–average above $8\,kg\,m^{-2}$ on 13 March and declined afterwards (Fig. 3e). The $2\,m$ temperature in the central Arctic continued to increase until the following day, reflecting the typical delay of the surface
warm front (corresponding to the MWAI / AR event) compared to higher altitudes.

At the beginning of the second campaign week, on 14 March, the mean wind speed over the measurement areas increased (Fig. 3d). This was due to a rising pressure gradient over the Norwegian and Greenland Seas between a steering low that reached southern Greenland and the persisting high pressure ridge over Scandinavia (Fig. 4b). The confluent tropospheric southwesterly flow between the two pressure systems formed and sustained a new AR between 15 and 16 March (Fig. 3a).
The new AR had a similar orientation as the MWAI / AR event on 12–13 March (Fig. 4a, b). In the central Arctic Ocean, the strongest warming occurred on 16 March and resulted in the highest $2\,m$ temperature averaged over all ERA5 grid points north of 80° N (latitude–weighted) for all days in March between 1979 and 2022 (not shown). Simultaneously, the central





**Figure 3.** Left column: (a) Calendar of March and April 2022 showing the categorized and colour-coded weather conditions. Bold font indicates days when research flights were performed. Right column: Regional averages (lines) and standard deviation (shading) of ERA5 based time series of (b) mean sea level pressure, (c) 2 m temperature, (d) 10 m wind (lines indicate speed, barbs indicate direction and speed), (e) integrated water vapour (IWV), and (f) daily accumulated precipitation, separated into rain (no transparency) and snow (partly transparent). Regional averages are performed for the southern (S, red), central (C, green), and northern (N, blue) region shown in Fig. 1. Vertical black lines indicate days shown in Fig. 4.





region featured the largest latitude–averaged $IHT_{north}$ and $IVT_{north}$ with a maximum of $9.44 \cdot 10^{10}\,\mathrm{W\,m^{-1}}$ and $388\,\mathrm{kg\,m^{-1}\,s^{-1}}$, respectively (Fig. 5).

Compared to climatological values, the latitude–averaged $IHT_{north}$ and $IVT_{north}$ show new records for the days between 07 March and 12 April, surpassing the previous records of $9.32 \cdot 10^{10}\,\mathrm{W\,m^{-1}}$ and $384\,\mathrm{kg\,m^{-1}\,s^{-1}}$ from 1996, respectively. The area–averaged $IVT_{north}$ over the central region also strongly exceeds the climatological $90^{th}$ percentile with a maximum value of around $230\,\mathrm{kg\,m^{-1}\,s^{-1}}$ (Fig. 2a). It should be noted that this does not necessarily reflect the most intense MWAI during HALO–(AC)[3], but only the strongest fluxes passing the central region.

While the mean $2\,\mathrm{m}$ temperature of the northern region started to decrease on 15 March, the standard deviation in Fig. 3c indicates temperatures close to $0\,^{\circ}\mathrm{C}$ until 16 March. Beyond the one standard deviation, temperatures were above freezing in some parts of the northern region. Thus, it is consistent that liquid precipitation was documented over the sea ice in the northern region (Fig. 3f, analysed in greater detail in Sect. 4.1). The northern region also showed the highest area–averaged daily precipitation on 13–16 March. In the central region, the area–averaged $2\,\mathrm{m}$ temperature was above freezing and a high
fraction of rainfall to the total daily precipitation was found.

  After the AR crossed Svalbard on 16 March, $2\,\mathrm{m}$ temperatures in the measurement areas decreased while the wind turned to southwesterlies (Fig. 3c, d). Low pressure systems were dominating especially over the central and southern region so that precipitation further increased (Fig. 3f). The latitude–averaged $IHT_{north}$ and $IVT_{north}$ remained positive in the central region but were strongly reduced because the AR propagated eastwards and dissolved (Fig. 5). The moisture flux decreased faster than
the heat flux, illustrating that the air masses behind the AR were still relatively warm but much drier.

  On 19 March, one of the low pressure systems moved from the central Fram Strait to the eastern side of Svalbard. The resulting northerly off–ice flow over the northeastern Fram Strait led to slightly negative latitude–averaged $IHT_{north}$ and $IVT_{north}$, and thus a weak MCAO over the northeastern Fram Strait (Fig. 2b, 3a, and 5). Moreover, the large–scale pressure constellation was changing considerably since a surface based high pressure system formed over Greenland. However, at upper levels, low
geopotential persisted, promoting the formation of a strong low pressure system near Iceland on 20 March. The low quickly propagated towards the northeast and was characterized by rapid deepening ($> 25\,\mathrm{hPa}$ core pressure decrease in 24 hours) and a frontal structure representative of a Shapiro–Keyser cyclone (Shapiro and Keyser, 1990). Unlike the Norwegian cyclone model, its warm front keeps circling around the centre, forming a warm core, instead of being caught up by the cold front (known as occlusion). On the eastern flank of the Shapiro–Keyser cyclone, $IHT_{north}$ and $IVT_{north}$ were similarly high as on 12
March (Fig. 5). However, as the moisture transport sufficiently dissipated before reaching the southern or central region, we did not classify this as an AR. Because of the low amount of heat in the cold air to the west of the Shapiro–Keyser cyclone, strongly negative $IHT_{north}$ was absent.

  In the climatological perspective, the prevailing pressure gradient between Scandinavia and Greenland, reflected in the extremely strong $500\,\mathrm{hPa}$ geopotential height anomalies ($> 10\,\mathrm{gpdm}$ over Scandinavia, $< -10\,\mathrm{gpdm}$ over Greenland, Fig.
6h), was responsible for the associated warming. Moreover, we found positive temperature anomalies at $850\,\mathrm{hPa}$ of more than $7\,\mathrm{K}$ over the Fram Strait and up to $9\,\mathrm{K}$ over Scandinavia and around Franz Josef Land over the warm period (Fig. 6b). At $2\,\mathrm{m}$, the warming was amplified to more than $16\,\mathrm{K}$ above climatology over Franz Josef Land and more than $10\,\mathrm{K}$ over the



Greenland Sea and Fram Strait (Fig. 6e). The bottom–amplified warming is consistent to the findings of Woods and Caballero (2016), confirming the reduction of the Arctic boundary layer inversion strength through intense MWAIs. Additionally, the high amounts of moisture carried by the ARs resulted in positive IWV anomalies of more than $2\,\mathrm{kg\,m^{-2}}$ over the southern, central, and parts of the northern region (up to $4\,\mathrm{kg\,m^{-2}}$ in the Norwegian and Greenland Seas, Fig. 6k).

### 3.1.2 Cold period

The intense Shapiro–Keyser cyclone marked a turning point in the dominating weather conditions over all measurement areas at the beginning of the third campaign week (21–27 March 2022). Due to anomalously high pressure over Greenland (even exceeding the 90th percentile on some days between 21 and 31 March, not shown), a strong zonal pressure gradient resulted over the Fram Strait, which caused northerly winds. Figure 4c shows the position of the low pressure system and the advection of the cold air masses for 21 March. As Kolstad et al. (2009) showed, there is a connection between such a circulation pattern and the onset of MCAOs in that region.

With the high amounts of water vapour circled around the cyclone's centre, we found the strongest southward latitude–averaged $\mathrm{IVT_{north}}$ on the cyclone's western flank ($-90\,\mathrm{kg\,m^{-1}\,s^{-1}}$, Fig. 5b). As the cyclone stayed over the Barents Sea while being filled up, it got access to cold, dry air from the sea ice covered Arctic Ocean, causing a quick reduction of absolute $\mathrm{IVT_{north}}$ values. The negative $\mathrm{IHT_{north}}$ and northerly winds associated with the cold air advection (Fig. 5a and 3d) favoured the development of an MCAO period. Convective cloud streets emerged over the Fram Strait as cold Arctic air masses were transported over the relatively warm ocean. In all three measurement regions, the area–averaged $2\,\mathrm{m}$ temperature including standard deviation remained below $0\,^\circ\mathrm{C}$ in the third campaign week (Fig. 3c). IWV dropped to the lowest area–averaged values in the northern region ($1.4$–$2.4\,\mathrm{kg\,m^{-2}}$, Fig. 3e). The moderate to strong MCAO conditions ($5\,\mathrm{K} < M < 10\,\mathrm{K}$) lasted for the entire third campaign week, partly exceeding the $90^\mathrm{th}$ percentile of the climatology (see Fig. 2b). Towards the end of the week, the MCAOs weakened (decreasing MCAO index $M$) in the western Fram Strait due to increasing pressure and calming winds, which turned more towards easterlies.

The fourth week of the campaign (28 March – 03 April) was predominantly marked by a northeasterly and easterly flow (Fig. 3d). While a high pressure system was still located over Greenland, a new anticyclone formed over the central Arctic and a low pressure complex developed over northwestern Russia (see Fig. 4d). The latter two systems advected relatively warm air masses from different subpolar regions (partly through the central Arctic) to the measurement areas. On their way north, the air masses were cooled over the sea ice covered Arctic Ocean. We classified this event as an aged warm air period (Fig. 3a) during which we observed mainly three recurring patterns. First, the easterly winds over Svalbard resulted in cloud–free areas west of the island due to lee effects. Second, converging winds over the central Fram Strait led to increased convective cloud formation. Third, some cloud streets developed in a narrow band over the Fram Strait in the westernmost ice–free area due to a less disturbed off–ice flow. Both position and intensity of these patterns varied from day to day. Southward advection of relatively warm air led to stronger southward heat and moisture fluxes (more negative $\mathrm{IHT_{north}}$ and $\mathrm{IVT_{north}}$) between 27 March and 03 April compared to the week before (Fig. 5a). Averaged over the central region, $\mathrm{IVT_{north}}$ was partly below the $25^\mathrm{th}$ percentile (Fig. 2a).







**Figure 4.** Maps of mean sea level pressure (white contour lines), $500\,\text{hPa}$ geopotential height (black contour lines), and $850\,\text{hPa}$ equivalent–potential temperature (shading and grey contours) from ERA5 data for representative days of the main weather conditions at 12 UTC. (a) and (b) represent the moist and warm air intrusions with AR characteristics, (c) shows the Shapiro–Keyser cyclone that marked the beginning of the cold air outbreak period, (d) and (e) represent the persistent northeasterlies with varying cold air advection strength, and (f) features the Polar Low west of Svalbard. The $15\,\%$ sea ice concentration from AMSR2 (Spreen et al., 2008) is displayed as blue contour line.



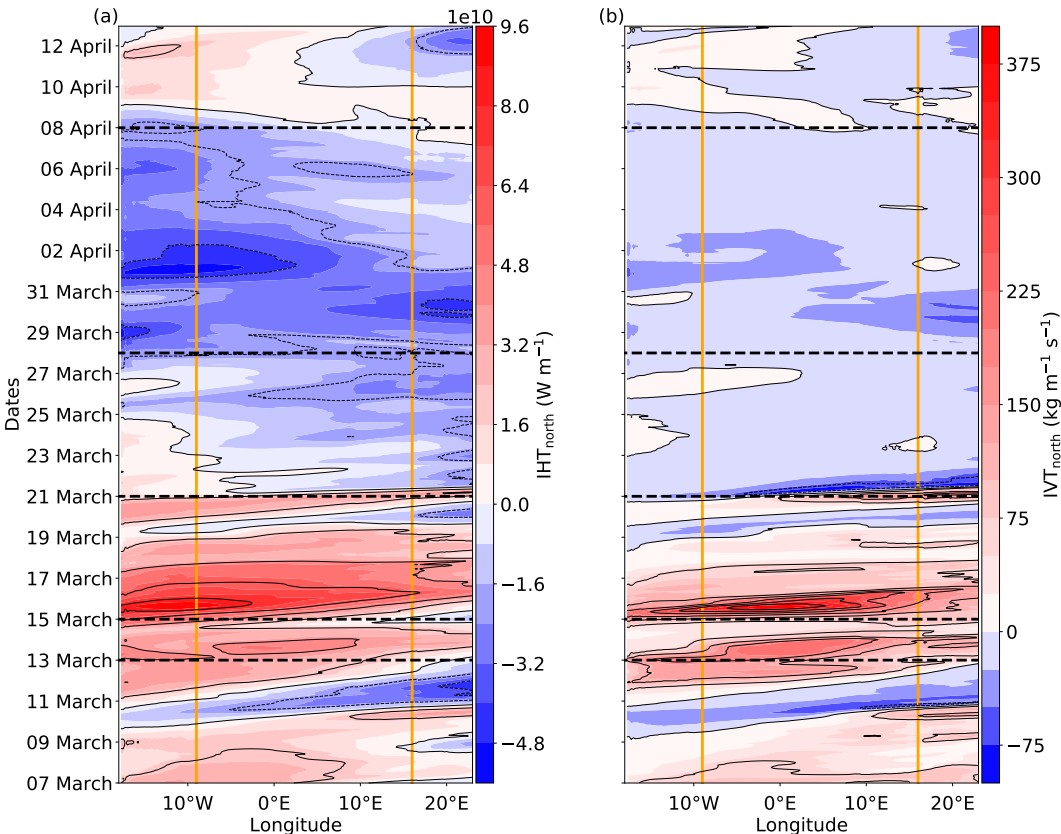

**Figure 5.** Hovmöller diagram of vertically integrated meridional fluxes of (a) heat (IHT$_{north}$), and (b) moisture (IVT$_{north}$) during HALO–(AC)$^3$, averaged over the central region latitudes. Positive (negative) values indicate northward (southward) fluxes. Orange vertical lines indicate the longitude boundaries of the central region. Important synoptic events shown in Fig. 4 are marked by horizontal dashed lines. Data is based on ERA5.

Another strong MCAO event ($M$ up to $9\,\mathrm{K}$, exceeding the 75$^{th}$ percentile) occurred on 01–02 April, associated with a weak cyclone that formed at the southern edge of Svalbard ahead of a trough in the $500\,\mathrm{hPa}$ geopotential (Fig. 4e). Temporarily, it advected slightly colder air masses with more intense northerly winds compared to the days before (Fig. 3d). The intensified

350    northerly winds resulted in the strongest southward latitude–averaged heat flux of the campaign (Fig. 5a). With a value of $-5.32 \cdot 10^{10}\,\mathrm{W\,m^{-1}}$, it is the 99.8$^{th}$ percentile of the southward latitude–averaged heat flux climatology.

In the first half of the fifth campaign week (04–10 April), the weather conditions in the measurement areas were similar to the previous week. The large system of low $500\,\mathrm{hPa}$ geopotential, which was responsible for the strong MCAO on 01–02 April, moved to northern Scandinavia and intensified. As a result, the main wind direction turned slightly more towards easterlies

355    in the southern and central region. Consequently, the off–ice flow over the Fram Strait, and the southward heat and moisture



fluxes weakened ($IHT_{north}$ and $IVT_{north}$ closer to 0, Fig. 5), reducing the MCAO activity. Gradually, 2 m temperatures and IWV increased over all three measurement regions (Fig. 3c, e).

An upper–level trough reached the Fram Strait from the east on 07 April, which led to another weak MCAO (Fig. 3a and 2b). Moreover, the positive vorticity of the trough together with the MCAO promoted the formation of a mesoscale cyclone with features of a Polar Low (details in Sect. 4.3). The Polar Low caused slightly positive latitude–averaged $IHT_{north}$ as warm air was advected northward from the North Atlantic in the eastern Fram Strait, ending a period of consistently negative meridional heat fluxes (Fig. 5a). In the following two days, the Polar Low moved to the northwest and dissipated over the sea ice northeast of Greenland due to the loss of atmospheric instability and strongly reduced upward sensible and latent heat fluxes (Fig. 4f).

Parallel to the evolution of the Polar Low, the low pressure complex, whose core had moved to northern Scandinavia, formed a region of warm and moist air advection in conjunction with a high pressure ridge over western Siberia. The moist air advection from Siberia, passing over southern Novaya Zemlya towards the Norwegian Sea, led to an increase of IWV above $10\,\mathrm{kg\,m^{-2}}$ over parts the southern region (Fig. 3e). On the final days of the campaign (10–12 April), the moist air was advected northwards to the sea ice covered western Fram Strait with a southerly upper–level flow (see also $IHT_{north}$ in Fig. 5a). Cirrus clouds observed during research flights (see Sect. 4.4) indicate the moist air at high altitudes. However, the corresponding large–scale circulation pattern during these days was not clearly attributable to one of the event classes we defined.

Overall, the period of pronounced MCAO activity lasting from 21 March to 12 April 2022 (as indicated by almost exclusively $M > 0\,\mathrm{K}$, Fig. 2b) reflects much more the typical, climatological state of the central region than the previous MWAI events. Climatologically, the full period from 07 March to 12 April is characterized by weak MCAO conditions ($M$ between 1 and 3 K). With a duration of 5–6 days, the first HALO–(AC)[3] MCAO period (21–26 March) was slightly longer than the mean duration of MCAO events (3–4 days, see Terpstra et al., 2021).

During the cold period, negative 500 hPa geopotential height anomalies (and low surface pressure) were located over Scandinavia and northwestern Russia, and positive geopotential height anomalies (and high surface pressure) over the central Arctic and Greenland (Fig. 6i). The persistence of the northerly winds due to the anomalous pressure constellation during the cold period explains the longevity of the MCAO conditions. Negative temperature anomalies at 2 m and at 850 hPa were found in the central Fram Strait, Norwegian Sea and parts of the Greenland Sea (Fig. 6c, f). Reduced IWV compared to the climatology overlaps well with the negative 850 hPa temperature anomalies (Fig. 6l). The Arctic Ocean north of Greenland and the central Fram Strait featured the strongest negative 2 m temperature anomalies with $-5.5\,\mathrm{K}$ and $-4.5\,\mathrm{K}$, respectively. Positive temperature anomalies at 850 hPa were restricted to regions that were not affected by cold air advection (central Arctic, eastern Barents Sea). Interestingly, the central Arctic Ocean showed negative temperature anomalies at 2 m and positive anomalies at 850 hPa, suggesting enhanced static stability compared to the climatological mean (Fig. 6c, f). In the area between Svalbard and Franz Josef Land, the opposite applies (positive anomalies at the surface, negative anomalies at 850 hPa). In the central Arctic, where the temperature anomalies at different height levels cancel each other out, we found only slightly negative IWV anomalies (Fig. 6l).





### 3.1.3 Climatological assessment of the full HALO–(AC)³ period

The 2 m temperature anomalies for the entire HALO–(AC)³ period (07 March–12 April) were positive in most regions shown in Fig. 6d. Only in the central Fram Strait and north of Greenland negative 2 m temperature anomalies down to $-1.2$ K and $-2.4$ K were found, coinciding with the regions of strongest negative 2 m temperature anomalies during the cold period. Over the sea ice in the western Fram Strait, 2 m temperature anomalies were slightly positive, indicating that the extremely positive anomalies of $> 8$ K of the warm period were not completely balanced out by negative anomalies during the longer cold period (Fig. 6d–f). Temperature anomalies at 850 hPa were positive in all regions shown in Fig. 6. The strongest anomalies with 2–3.5 K were located in the central Arctic Ocean and northern Barents Sea, coinciding with the regions where 850 hPa temperature anomalies were positive in the cold period (Fig. 6a, c). Positive temperature anomalies combined with high surface pressure over Scandinavia (central Arctic) in the warm (cold) period resulted in overall positive 500 hPa geopotential height anomalies over these regions (Fig. 6g). Geopotential height anomalies at 500 hPa were only close to 0 over the east coast of Greenland, where low geopotential also remained during the first part of the cold period despite high surface pressure due to extremely low temperatures.

Over the central region, we found small northward heat fluxes ($\mathrm{IHT_{north}}$ of $5.4 \cdot 10^8 \, \mathrm{W \, m^{-1}}$) and moisture fluxes ($\mathrm{IVT_{north}}$ of $18 \, \mathrm{kg \, m^{-1} \, s^{-1}}$). Consequently, the thermal energy input into the Arctic through this pathway was slightly positive and the brief but intense MWAIs / ARs overcompensated the long MCAO period. Similarly, the dry air masses associated with the MCAOs were not sufficient to balance out the extreme moisture transport related to the MWAIs / ARs during the warm period.

### 3.2 Ny–Ålesund

Ny-Ålesund, with its fixed location in the northwest of Svalbard, adds a Eulerian view to the synoptic events described above. Describing the relationship between the weather conditions in the HALO–(AC)³ regions and Ny-Ålesund helps to find the context in future studies that might include the additional observations at Ny-Ålesund. The atmospheric conditions at Ny–Ålesund were also divided into a warm (11–21 March) and a colder period (22 March–12 April), which is shifted by one day compared to the full central region (Fig. 7).

The warm air advection related to the first AR led to an increase of 2 m temperature from about $-14\,^{\circ}\mathrm{C}$ to $+2\,^{\circ}\mathrm{C}$ within 19 hours (meteorological tower measurements, not shown). As a consequence of high surface pressure ($> 1010$ hPa), and high temperatures and moisture load (IWV$> 10 \, \mathrm{kg \, m^{-2}}$, Fig. 7b), the troposphere extended up to a tropopause height up to 12.9 km (measured by the 12 UTC radiosonde on 12 March, Fig. 7a). Lower surface pressure resulted in reduced tropopause heights on 15–16 March although the highest temperature and humidity were observed on these days (IWV of $14.5 \, \mathrm{kg \, m^{-2}}$ on 15 March at 18 UTC, Fig. 7b). The 2 m temperature remained above freezing for five days and even reached a new record high temperature for all days of March on 15 March ($5.5\,^{\circ}\mathrm{C}$).

The warm period was characterized by constant low–level cloud cover over Ny–Ålesund detected by the ceilometer, interrupted only by a few hours of high–level cloud occurrence in connection with a high pressure ridge passing the site in the night

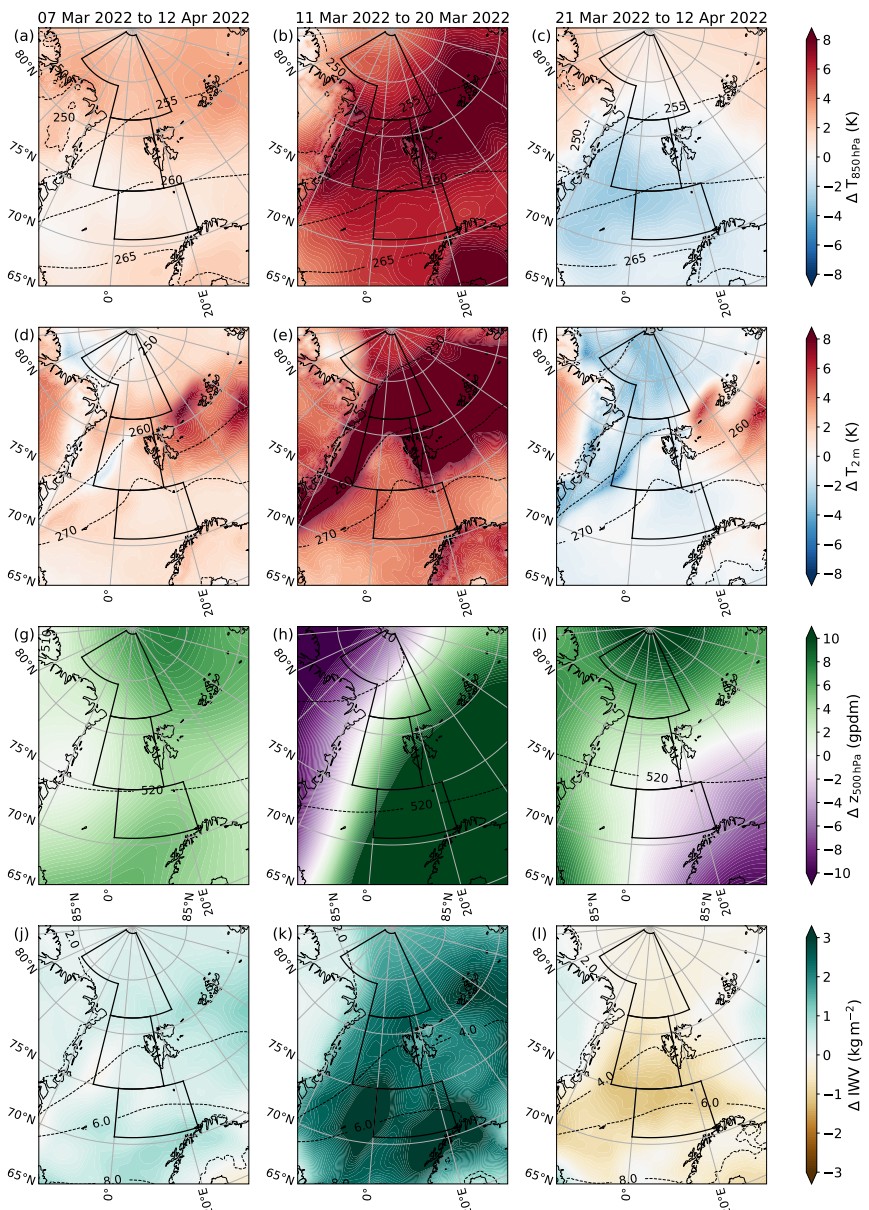

**Figure 6.** Anomaly maps of (a–c) temperature at $850\,\mathrm{hPa}$ ($\Delta\mathrm{T}_{850\,\mathrm{hPa}}$), (d–f) $2\,\mathrm{m}$ temperature ($\Delta\mathrm{T}_{2\,\mathrm{m}}$), (g–i) geopotential height at $500\,\mathrm{hPa}$ ($\Delta\mathrm{z}_{500\,\mathrm{hPa}}$), and (j–l) vertically integrated water vapour ($\Delta\mathrm{IWV}$) based on ERA5 for the entire campaign, the warm, and the cold period in the left, middle, and right column. The coloured contours show the long-–term mean subtracted from the mean of the time period in 2022 so that positive (negative) values correspond to anomalies above (below) the long-term mean. The dashed black isolines describe the long—term climatology in each panel. The three measurement regions (shown in Fig. 1) are illustrated as black boxes.





of 14–15 March (not shown). The situation changed during the course of 17 March, when the cloud deck started to dissolve and the $2\,\mathrm{m}$ temperature dropped below $0\,^\circ\mathrm{C}$.

The influence of the Shapiro–Keyser cyclone (see Sect. 3.1) delayed the beginning of the cold period over Svalbard compared to the central region by about one day. Northerly winds corresponding to the MCAO period led to extremely dry conditions
with IWV down to $1.1\,\mathrm{kg\,m^{-2}}$ on 24 March at 06 UTC (Fig. 7b), which is below the 3[rd] percentile of all radiosondes between 07 March and 12 April 1993–2022. At Ny–Ålesund, the MCAO conditions between 23 and 29 March were mostly associated with clear sky and $2\,\mathrm{m}$ temperatures below $-15\,^\circ\mathrm{C}$. Aged warm air reached Ny–Ålesund on 28 March with more humidity, increasing IWV up to $4\,\mathrm{kg\,m^{-2}}$. After the subsequent stronger MCAO on 01–02 April, clear sky conditions due to orographic lee effects (foehn) associated with the calm easterly winds persisted until 05 April.

Generally, the Ny–Ålesund observations provide a ground based view on the atmospheric events that occurred in the Svalbard region during HALO–(AC)[3]. Because of orographic effects especially in the cold period, the perspective of Ny–Ålesund is not representative for the entire central region, and much less for the southern and northern regions. Additionally, the temporal delay related to the propagation of synoptic systems from other areas to Ny–Ålesund during the warm period must be kept in mind when considering other observations from Ny–Ålesund in relation to HALO–(AC)[3] in future studies.

### 3.3   Sea ice conditions

The different synoptic patterns changed the sea ice conditions in the measurement areas. In Fig. 8a–c, we present the spatial distribution of SIC for three selected time periods. We selected 09–11 March, 14–16 March, and 10–12 April 2022 to show SIC before and during the MWAIs as well as after the MCAO period. SIC was $> 90\,\%$ in almost the entire northern region in the first period (Fig. 8a). In the central region, this was true for about $30\text{--}40\,\%$ of the area with a southwest–northeast oriented
sea ice edge, approximately from the lower left to the upper right corner of the box. The southern region was almost completely ice–free over the entire campaign duration (Fig. 8a–c).

The MWAI / AR events on 12–13 and 15 March showed a clear impact on the sea ice due to a reduction of SIC in the northwestern part of the central region (Fig. 8b). Averaged over the central region, SIC reached its minimum for the campaign duration, which is below the climatological mean but within the 10–90[th] quantiles (Fig. 8d). We assume that ice dynamics
related to strong winds caused the decrease, but ice melt cannot be excluded as well because temperatures were above freezing and liquid precipitation was observed over sea ice. Rapid ice melt has been attributed to warm air advections and induced increases of heat flux in summer (Tjernström et al., 2015; Woods and Caballero, 2016). After the two AR events, an unusually large polynya opened from north of Svalbard to Franz Josef Land (Fig. 8b). Although the liquid precipitation on snow alters the signal of the microwave radiometry and might have increased the uncertainty of the SIC product (Stroeve et al., 2022; Rückert
et al., 2023) that SIC reduction was obvious in visual satellite images as well (e.g., NASA Worldview, not shown). The events during HALO–(AC)[3] and the wealth of measurements on water vapour, clouds and precipitation are thus an interesting case for future research about the interplay of atmosphere and sea ice and the role of strong MWAIs and ARs on SIC during springtime.

After the AR events, the sea ice conditions recovered (Fig. 8d) mostly due to sea ice transport from the central Arctic towards the Fram Strait. Also the large polynya between Svalbard and Franz Josef Land was closed again (Fig. 8c). The southward



**Figure 7.** Time series of Ny–Ålesund radiosondes of (a) temperature profiles (shading), height of thermal tropopause (black line), and wind barbs in selected levels, and (b) specific humidity profiles (shading) and resulting IWV (black line, right axis). Vertical dashed black lines indicate the days shown in Fig. 4. The red (blue) box indicates the shifted warm (cold) period.

transport can be attributed to the high MCAO activity between 21 March and 07 April, where northerly and northeasterly winds and near–surface temperatures below freezing dominated. For most of the campaign time, the average SIC in the central region was higher compared to the climatology (Fig. 8d).

We found less SIC with respect to the climatology south of 75° N and west of 0° E as well as northeast and southeast of Svalbard and a higher SIC in the central and western Fram Strait (Fig. 8e). The positive SIC anomaly in the central and western
Fram Strait overlaps well with the negative 2 m temperature anomalies seen in Fig. 6d, f. During HALO–(AC)³, there was no





sea ice at the west coast of Svalbard, an area that according to the climate data record used to be covered by some ice, on average. For the northern and southern regions, the mean SIC for the entire campaign period did not differ more than 5 % on average compared to the climatological mean.

Note that uncertainties in the marginal ice zone are especially large due to weather related filters and as a result of temporal and spatial interpolations. This mainly affects the area of the central region where the estimated uncertainties provided with the SIC product (total standard uncertainty) can reach up to 40 % at the ice edge, mostly due to smearing uncertainty (gridding of the swaths, i.e., areas imaged on the surface, from several daily overflights), and different satellite footprints at different frequency channels.



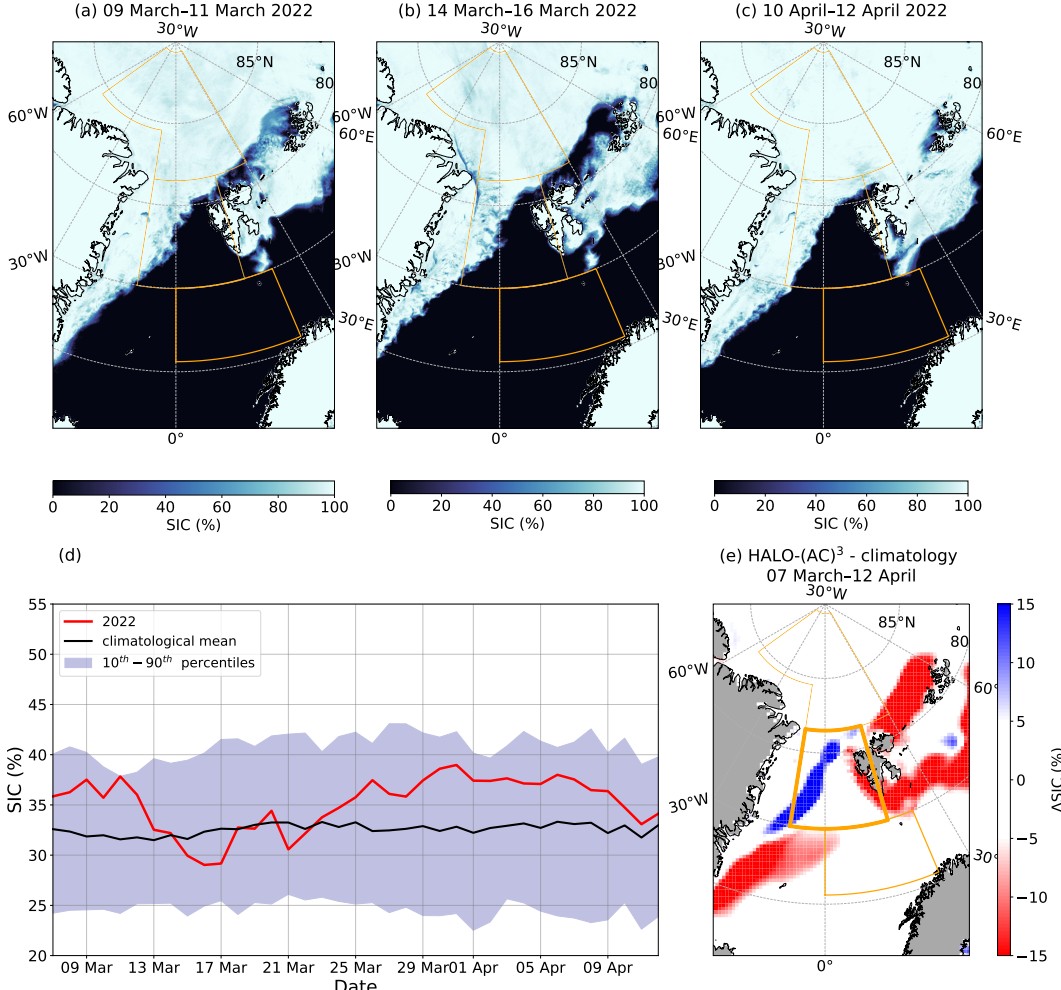

**Figure 8.** Average sea ice concentration (SIC) for two three–day time periods during the HALO–(AC)$^3$ campaign: (a) prior to the first AR event, (b) during the second AR event, (c) at the end of the campaign. Data is from the MODIS-AMSR2 product at 1 km grid resolution. (d): SIC time series averaged over the central region for HALO–(AC)$^3$, as well as the mean, and 10–90$^{th}$ percentiles of the 1979–2022 climatology. (e): Time–averaged SIC anomalies from the climatological mean. Note that here only differences larger 5 % are considered due to the uncertainties of the satellite product in the marginal ice zone. Data for (d) and (e) is from the OSI–SAF sea ice concentration climate data record Copyright © 2018 EUMETSAT.





## 4 Specific events during HALO–(AC)[3]

### 4.1 Warm air intrusions and Atmospheric Rivers


The first two campaign weeks were characterized by four independent AR events (12–13 March, 14 March, 15 March, and 20 March) that were at least partly sampled by research aircraft in the three measurement regions. Back trajectory analyses with LAGRANTO (Sect. 2.4.6) revealed that the air masses of the 12–13 March event originated from mid–latitude regions of the North Atlantic and from central Europe (Fig. B1a). For the second AR event on 15 March, the air masses originated

almost exclusively from the western part of the mid–latitude North Atlantic (Fig. B1b). Maximum IVT (meridional and zonal) exceeded $400 \, \mathrm{kg \, m^{-1} \, s^{-1}}$ between 13 and 16 March in the central region (see Appendix A, Fig. A1).

Apart from these major MWAIs in conjunction with ARs, two more remarkable events of moisture transport were identified. The first one was associated with the Shapiro–Keyser cyclone (Sect. 3.1) and approached our region of interest on 20 March with a maximum IVT exceeding $500 \, \mathrm{kg \, m^{-1} \, s^{-1}}$. The Shapiro–Keyser cyclone, however, steered the AR away from the sea

ice edge to the southeast of Svalbard and dissipated sufficiently before entering the southern or central region. The second moisture transport event, associated with the low pressure complex over Scandinavia from 08 to 10 April, was confined to the southern region and much more zonally oriented but too weak to be detected as AR.

We compare the strength of AR events during HALO–(AC)[3] by analysing them with respect to the relation between IVT and latitude over the North Atlantic and Arctic Ocean (50–90° N, 60° W–40° E) (Fig. 9). Note that a different region was

selected to extend the view on the North Atlantic, one of the major pathways of ARs (Guan and Waliser, 2017; Nash et al., 2018). Along their northward propagation, ARs generally decline in intensity. If otherwise ARs start to form in the Arctic, their moisture supply is reduced so that their intensification is limited (Papritz et al., 2021). Therefore the number of strong AR events decreases meridionally.

North of 70.6° N (southern boundary of the southern region), only few ARs exceed an IVT of $250 \, \mathrm{kg \, m^{-1} \, s^{-1}}$ when averaged

over the detected AR area (Fig. 9). In the climatology, the majority has mean values around $150 \, \mathrm{kg \, m^{-1} \, s^{-1}}$. The AR events on 12–13, and 15 March represent strong cases in terms of mean IVT for all time steps as they lie outside the 25[th] percentile in Fig. 9. On 15 March at 18 UTC, the AR averaged IVT even exceeded the 10[th] percentile while being centered at 76° N. While the AR centre from the event on 12–13 March was located most northerly before it dissipated, the AR from 15 March was characterized by the most intense moisture transport with mean IVT of around $250 \, \mathrm{kg \, m^{-1} \, s^{-1}}$ when centered at 70° N,

decreasing to $200 \, \mathrm{kg \, m^{-1} \, s^{-1}}$ at 75° N. As these AR events had meridionally elongated structures, the outflow region reached up to the central Arctic while their centres were located at 70–78° N (Fig. 9). During their poleward propagation, the moisture transport decreased so that they no longer fulfilled the detection requirements.

The AR events in the warm period were associated with anomalous amounts of precipitation. In Fig. 10, we show the hourly averaged precipitation rate from 11 to 20 March 2022 as well as for the same days of the climatology. Further, the fraction of

the precipitation rate in 2022 to that in the climatology is illustrated. Along the east coast of Greenland, the total precipitation and snowfall rate partly exceeded $1 \, \mathrm{mm \, h^{-1}}$ (Fig. 10a, d), around six times higher compared to climatology (Fig. 10c, f). Over the North Atlantic, Fram Strait, and central Arctic Ocean, total precipitation and snowfall rates exceeded the climatology by





a factor of 3. In parts of the Norwegian and Barents Seas, total precipitation and snowfall rates were below the climatological

mean. Record breaking rainfall rates were found northwest of Svalbard, exceeding the climatology up to a factor of 36 (Fig.

10i). However, the actual rainfall rate in this area was below $0.25\,\mathrm{mm\,h^{-1}}$ and the high deviation is caused by even lower

rainfall rates in the climatology.

In summary, the ARs observed during HALO–(AC)³ featured uncommonly high AR–averaged IVT values, higher than 90%

of all ARs in the climatology. Additionally, record breaking rainfall rates occurred over the sea ice northwest of Svalbard.

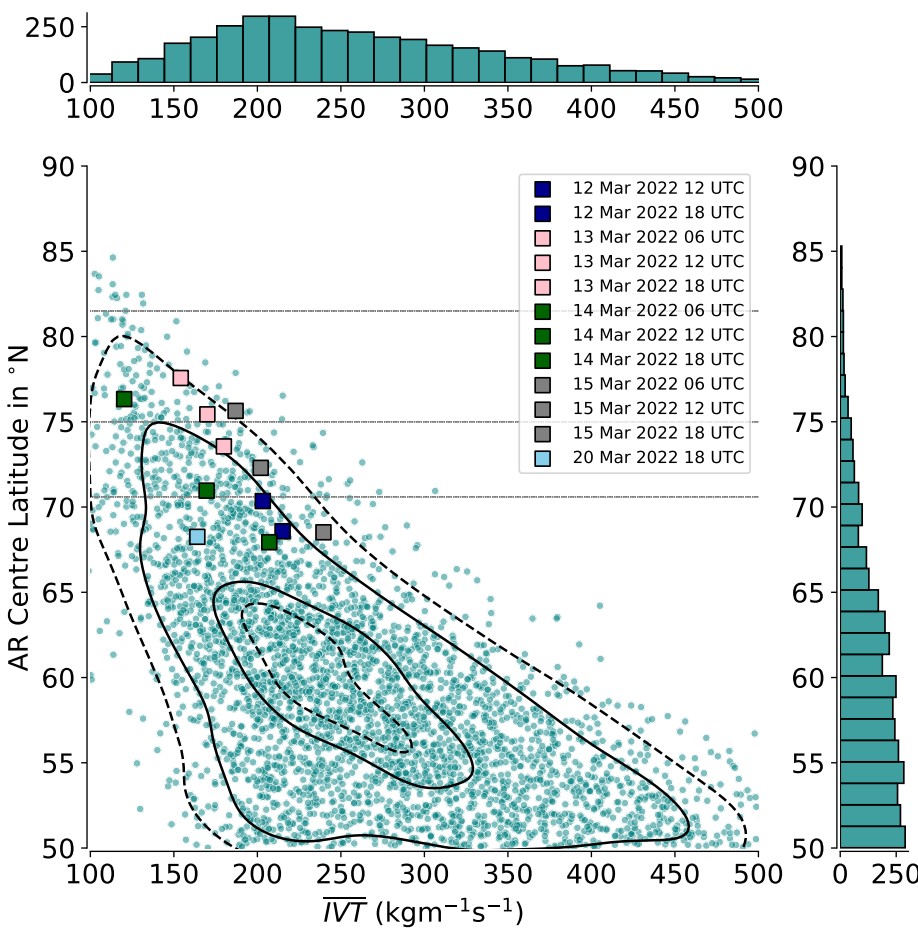

**Figure 9.** Six hourly climatological (1979–2022) distribution of central latitudes of Atmospheric Rivers (ARs) as a function of mean AR IVT

using an ERA5 based AR catalogue modified from Guan and Waliser (2015). Cases categorized as ARs during HALO–(AC)³ are illustrated

by coloured boxes for 06, 12, and 18 UTC of respective flight days. Black (dashed) lines indicate the 25th and 75th (10th and 90th) percentiles

of a kernel density estimation to visualize the shape of the histogram. Horizontal dotted grey lines indicate the latitude boundaries of the

three measurement regions (Table 1).

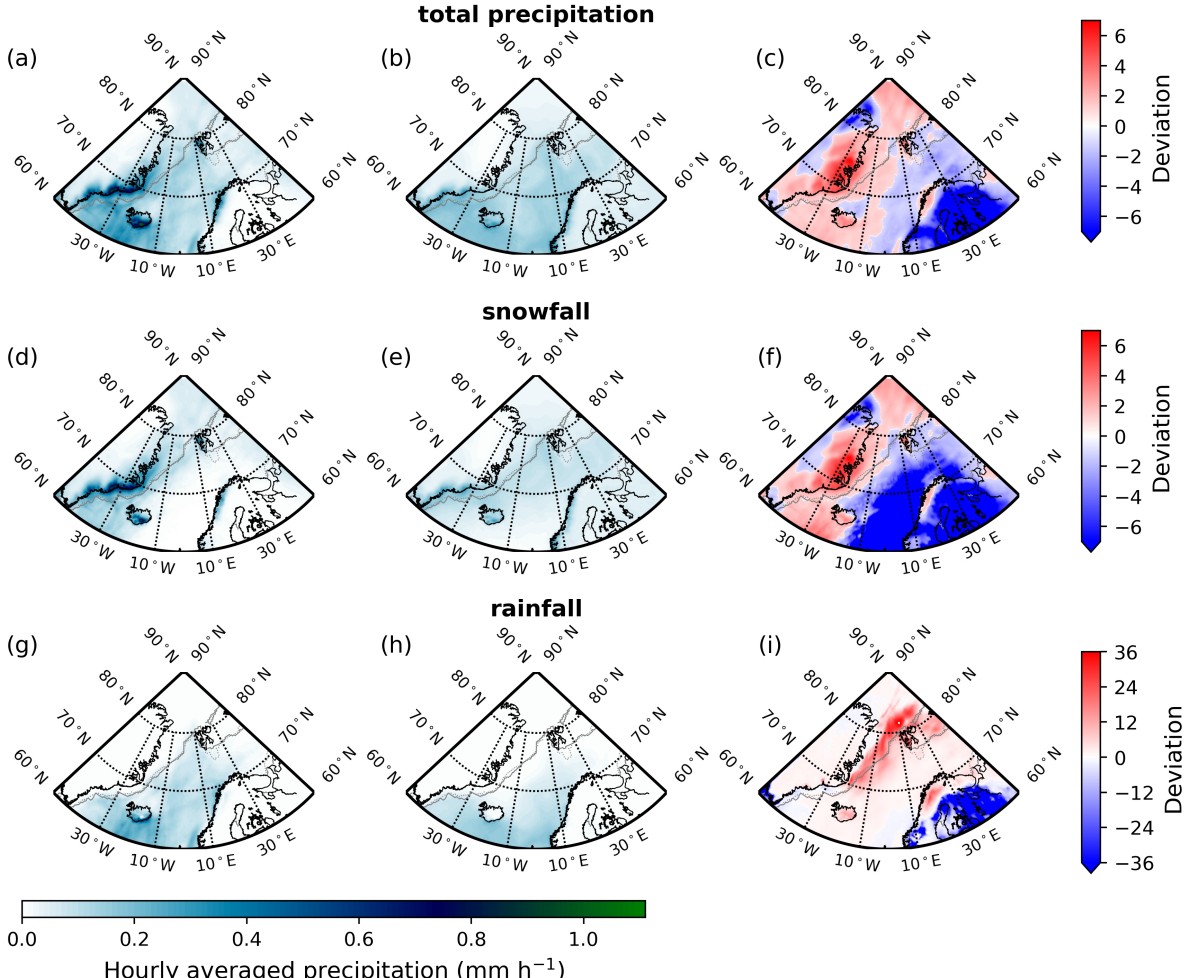

**Figure 10.** Hourly averaged precipitation, snowfall and rain rate ($\mathrm{mm\,h^{-1}}$) derived from ERA5 for (a, d, g) 11–20 March 2022, and (b, e, h) the climatology. In (c, f, i), we show the deviation from the climatology. Note that the rainfall over sea ice is barely visible in both (g) and (h). Positive deviations of precipitation rates are illustrated as the fraction of the average rate in 2022 to the rate of the climatology while negative deviations are given as its inverse.

## 4.2 Marine cold air outbreaks

The cold period included several MCAO events (21 March–07 April), which are analysed in detail specifically for the central region. Figure 11 shows the time series of atmospheric conditions as described in Sect. 2.4.4. To highlight the conditions in the MCAOs' source region, we also include the temporal evolution of the parameters averaged over the northern region (without extension to the west) in Appendix C (Fig. C1). The corresponding temporal development over the southern region is shown in Fig. C2 in Appendix C.





After the Shapiro–Keyser cyclone, which caused multi–layered clouds and intense snowfall on 21 March at the west coast of Svalbard (Fig. 11d–g), MCAO conditions were established. From 22 to 26 March, the contribution of convective to total snowfall increased over open water and mid– and high–level cloud cover decreased. Over sea ice, the total snowfall was low and almost exclusively stratiform. Liquid precipitation was not present except for 21 March. Low–level cloud cover remained high until 07 April, reflecting the presence of the atmospheric boundary layer clouds typically associated with MCAOs (Fig. 11d). However, some parts of the open water regions of the central region were not fully covered by low–level clouds as the 5% percentile was mostly between 0.2–0.4.

The strong MCAO events (21–26 March and 01–02 April) were characterized by northerly lower tropospheric winds and high values of domain–averaged turbulent fluxes of sensible and latent heat flux (Fig. 11a, h). The maximum values of sensible and latent heat coincided with the highest wind speeds and low temperatures (21–22 March, Fig. 11a, b, h). Note that over the southern region the sensible heat fluxes were slightly lower, whereas, on average, latent heat fluxes were higher than over the central region (Fig. C2). One possible explanation is that the lower tropospheric air temperature increased towards the south. Hence, sensible heat fluxes would become smaller while the latent heat fluxes would increase due to a rising saturation water vapour pressure at the air–sea interface (e.g., Hartmann et al., 1997).

During the weak MCAOs and aged warm air periods, northeasterly winds dominated (Fig. 11a). Compared to strong MCAO conditions, sensible heat fluxes were reduced by a factor of 2 while latent heat fluxes decreased to a lesser extent. Between 27–30 March and 03–05 April, when aged warm air was advected to the central region, precipitation decreased as the air mass dried and temperatures increased (Fig. 11b, c, g). Also, foehn effects due to the easterly winds over Svalbard might have contributed to drier and warmer conditions over parts of the Fram Strait as this effect can influence the atmospheric conditions more than 100 km off the coast (Shestakova et al., 2022). For the first aged warm air period, the southward advection of the air masses from the sea ice covered ocean took about one day. This can be seen when comparing the temporal evolution of the lower tropospheric temperature and relative humidity over the northern and central regions (e.g., low relative humidity from 29 to 30 March, Fig. C1a–c vs. 11a–c).

From 05 April on, the mean wind speed decreased and turned to northeast. Hence, more convergent regions developed in the lee of Svalbard with a rather diffuse low–level cloud structure (not shown). These converging winds in combination with a weak MCAO promoted the formation of a mesoscale cyclone in the following days, which is explained in greater detail in the next section.

During HALO–(AC)[3], different wind regimes and strengths of MCAOs were observed. MCAO strength lied partly outside the 90[th] percentile, partly well within the interquartile range of the climatology. Wind regimes ranged from northwest to northeast with different intensity. The large variety of MCAO conditions provides great opportunities for detailed cloud evolution and air mass transformation studies.

## 4.3 Polar low

The Polar Low formed close to the west coast of Svalbard where cold air transported by north to northeasterly winds converged with warmer North Atlantic air from south to southeasterly winds on 07 April, forming a baroclinic zone. During the





**Figure 11.** Temporal evolution of (a) wind, (b) air temperature (T), and (c) relative humidity (RH) on the 1000, 925, and 850 hPa pressure levels, and of (d) low–, (e) mid–, and (f) high–level cloud cover (CC), (g) precipitation, and (h) surface turbulent heat fluxes (STHF, upward is positive) over the central region for 21 March – 07 April 2022 derived from ERA5. For cloud cover, we show the median, the 25 and 75th percentiles, and the 5 and 95th percentiles of a 6 hour interval over open–water surfaces only as crosses, thick vertical lines, and thin vertical lines, respectively. In (g), we display area–averaged values of the accumulated 6 hour snowfall distinguished between convective and total precipitation, and open water (OW) and sea ice (SI). White crosses show the accumulated mean rainfall with more than $0.05\,\mathrm{mm}\,(6\,\mathrm{h})^{-1}$.



development phase of the Polar Low, the mean sea level pressure showed closed isobars from 08 April 2022 00 UTC on (Fig.
4f). We examine which environmental conditions for the Polar Low formation set by Radovan et al. (2019) were fulfilled. The
conditions C1–C6 explained in Sect. 2.4.5 and Table 2 are checked in a $200\,\mathrm{km}$ radius around the mean sea level pressure
minimum of the Polar Low at that time (Fig. 12):

The vertical temperature gradient conditions C1 and C2 are fulfilled with $44\,\mathrm{K}$ and $8.5\,\mathrm{K}$, respectively, while the lapse
rate condition (C3) was not fulfilled with $4.2\,\mathrm{K\,km^{-1}}$. Conditions C4$_\mathrm{i}$ and C4$_\mathrm{ii}$, which are a measure for the availability of
moisture for latent heat release, exceed the thresholds of $75\,\%$ and $82\,\%$ with $92.7\,\%$ and $93.9\,\%$. Near surface wind gusts
reached $16.6\,\mathrm{m\,s^{-1}}$ during the development phase of the Polar Low (00 UTC), which was sufficient to fulfill condition C5.
They increased to $18.8\,\mathrm{m\,s^{-1}}$ 9 hours later when the Polar Low was in its mature stage. If we considered the maximum
of the $10\,\mathrm{m}$ wind speed instead of the gust from ERA5 in the $200\,\mathrm{km}$ circle, the condition would not have been fulfilled.
However, using the Copernicus Arctic Regional Reanalysis (CARRA) from ECMWF with a $2.5\,\mathrm{km}{\times}2.5\,\mathrm{km}$ resolution, we
fond maximum wind speeds of more than $16\,\mathrm{m\,s^{-1}}$ ($19\,\mathrm{m\,s^{-1}}$) at 00 UTC (09 UTC). The trough connected to the Polar
Low was not sufficiently strong with respect to the climatology as the geopotential height anomaly (C6) was only $-80\,\mathrm{gpm}$.
Anomalously high geopotential around Svalbard and the central Arctic during that time might have obscured the strength of
the trough, resulting in the failure of condition C6.

Some of the conditions had a large regional variability within the $200\,\mathrm{km}$ circle. Condition C2 ranged from below the
threshold of $6\,\mathrm{K}$ in the southern part of the Polar Low to $10\,\mathrm{K}$ in the northern half. As the northwestern part of the circle lied
over sea ice, the vertically averaged lower tropospheric lapse rate (C3) indicated stable stratification and reached values up to
$10\,\mathrm{K\,km^{-1}}$. These high values of the lapse rate dominated the computation of the 75$^\mathrm{th}$ percentile although the lapse rate was
below the threshold in most parts of the circle.

At the mature stage (08 April 06–09 UTC), the region of maximum $\mathrm{SST}-\mathrm{T_{500\,hPa}}$ difference coincided well with the
observed convective clouds during the research flight with HALO in the eastern part of the Polar Low (see Appendix D,
Fig. D1). It also overlapped with the maximum precipitation rate indicated by ERA5 with $1.2\,\mathrm{mm\,h^{-1}}$ (not shown). Wind
gusts peaked at $18.8\,\mathrm{m\,s^{-1}}$ and were therefore stronger than during the development phase. ERA5 data also show updrafts of
$-1.75\,\mathrm{Pa\,s^{-1}}$ at $850\,\mathrm{hPa}$ in the northeastern half of the Polar Low (northwest to southeast) and slight descending motion in the
centre (not shown). It is likely that higher precipitation rates and updrafts occurred that were smoothed out by the relatively
coarse ERA5 data.

Dropsonde measurements show high values of relative vorticity in the lowest $2\,\mathrm{km}$ and above $6\,\mathrm{km}$, indicating cyclonic
rotation. While the lower tropospheric relative vorticity is related to the Polar Low, the upper tropospheric value is associated
with the trough in the geopotential height. Measured relative vorticity at $850\,\mathrm{hPa}$ ($1.5\,\mathrm{km}$ height) was about $3.5{\cdot}10^{-5}\,\mathrm{s^{-1}}$ and
therefore did not exceed the threshold suggested by Stoll (2022) ($> 20 \cdot 10^{-5}\,\mathrm{s^{-1}}$). However, the flight track did not perfectly
align with the Polar Low's centre and extended too far to the south and east of the Polar Low where the mean sea level pressure
isobars do not suggest a closed circulation (Fig. 13b). Negative relative vorticity to the east of the Polar Low, which is related
to lee effects, also influenced the calculation from the dropsonde measurements. ERA5 data shows a maximum of $22{\cdot}10^{-5}\,\mathrm{s^{-1}}$
at $850\,\mathrm{hPa}$ within the Polar Low on 08 April at 09 UTC.



The Polar Low observed during HALO–(AC)[3] fulfilled four of the six conditions. However, Polar Lows can also form
585    without fulfilling all conditions. According to the climatology of Polar Lows in Zahn and von Storch (2008), which covers the
years 1949–2005, our Polar Low was located unusually far in the north. Also Radovan et al. (2019) investigated 33 Polar Lows,
of which 32 were centered further to the south than our case. Despite its northerly position, the $10\,\mathrm{m}$ wind speed of our Polar
Low was larger than the average over the 33 Polar Lows.

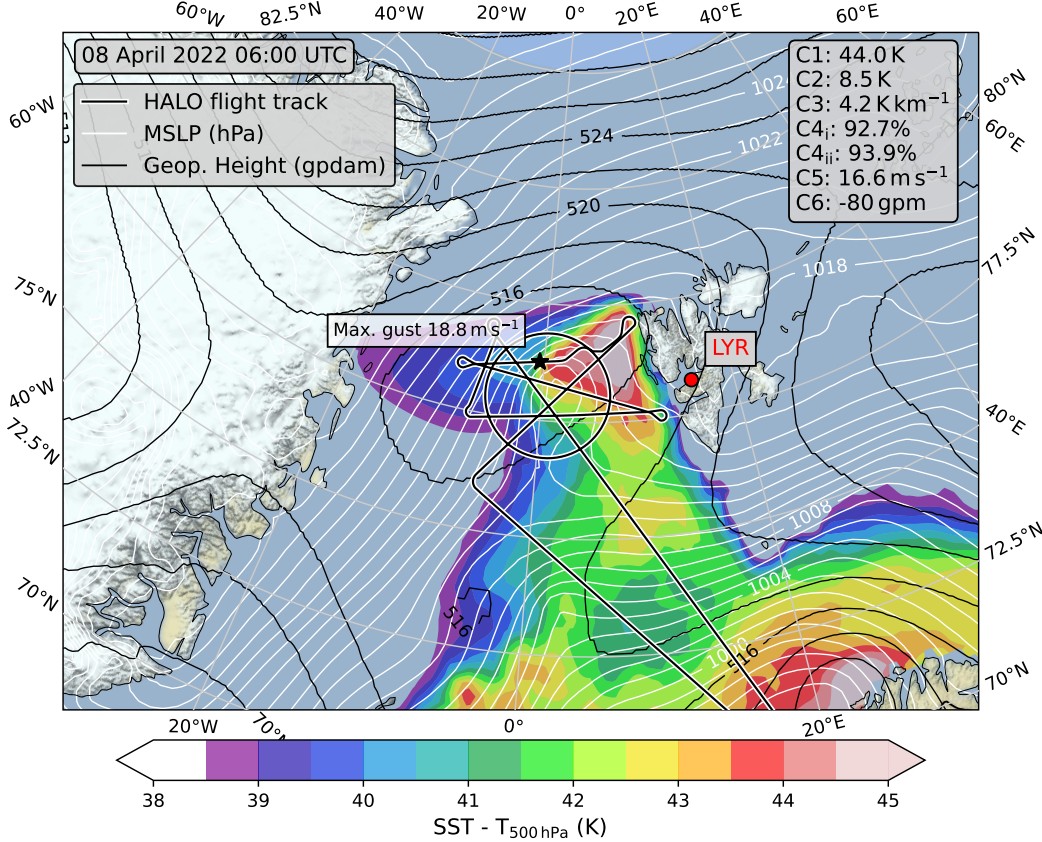

**Figure 12.** Mean sea level pressure (MSLP, white contour lines), $500\,\mathrm{hPa}$ geopotential height (black contour lines), and the difference
between sea surface temperature (SST) and $500\,\mathrm{hPa}$ temperature during the mature stage of the Polar Low on 08 April 2022 06:00 UTC.
Additionally, the maximum $10\,\mathrm{m}$ wind gust at 09:00 UTC (black star and text label) and the HALO flight track (black line with white
outline) are included. The Polar Low conditions C1 ($\mathrm{SST} - 500\,\mathrm{hPa}$ temperature), C2 ($\mathrm{SST} - T_{2\,\mathrm{m}}$), C3 (vertical mean of the vertical
potential temperature gradient below $850\,\mathrm{hPa}$), C4 (i: mean relative humidity below $950\,\mathrm{hPa}$, ii: mean relative humidity between 850 and
$950\,\mathrm{hPa}$), C5 (maximum $10\,\mathrm{m}$ wind speed), and C6 ($500\,\mathrm{hPa}$ geopotential height anomaly) analysed by Radovan et al. (2019) are given for
08 April 2022 00:00 UTC. Data is based on ERA5.





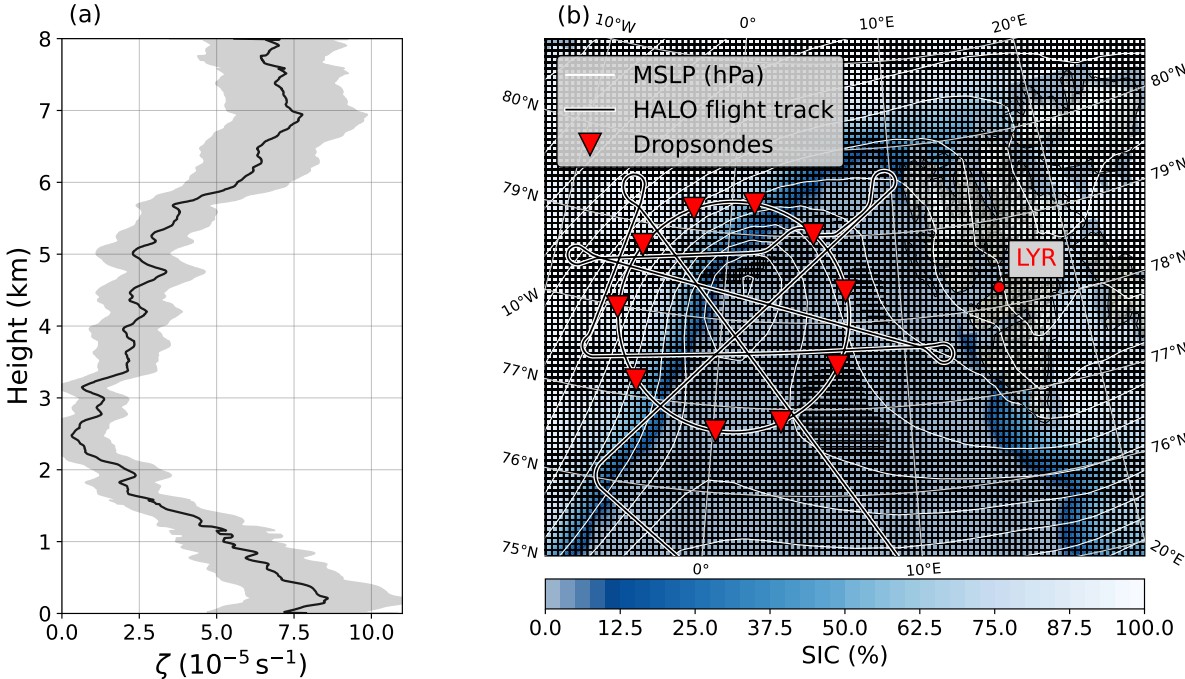

**Figure 13.** (a) Relative vorticity $\zeta$ computed from 10 dropsondes. (b) Location of the dropsondes (red triangles) launched from HALO while flying in a circle around the Polar Low. ERA5 based mean sea level pressure (white contour lines) and sea ice concentration (coloured filled contours) is shown for 08 April 2022 09 UTC. The HALO flight track is displayed as black line with white outline. Horizontal (horizontal and vertical) hatching indicates areas where ERA5 based 850 hPa relative vorticity is below $-10 \cdot 10^{-5}\,\mathrm{s}^{-1}$ (above $20 \cdot 10^{-5}\,\mathrm{s}^{-1}$).

### 4.4 Arctic cirrus over sea ice

The sequence of HALO flights between $10 - 12$ April 2022 aimed to characterize a pronounced field of Arctic cirrus, which entered the central Arctic through the Fram Strait and was finally located close to the North Pole. The initial synoptic state of this period was dominated by a low pressure system, which was centered over Scandinavia on 09 April, advecting warm and moist air westwards (see Sect. 3.1). Along approximately the $5°$ W meridian the westward flow converged with a cold air mass that formed over the Greenland ice sheet and flowed eastwards into the Fram Strait. Due to the convergence, a strong

vertical wind shear developed south of Greenland with low–level northeasterly winds and southeasterly winds at levels above 700 hPa along the east coast of Greenland. The southerly upper–level flow generated a poleward moisture transport, which was accompanied by cirrus and mid–level clouds west of Svalbard and north of Greenland. This upper–level flow continued during the sequence of three research flights conducted by HALO and roughly connected the air masses sampled during the three consecutive days.

The trajectories of the air masses sampled on 11–12 April are shown in Fig. 14a, c. While on 11 April two branches, one southerly and one easterly, merged into the area of observations, the trajectories of 12 April first followed the westward moist





air transport south of Svalbard and then turned northwards into the central Arctic. Figure 14b and d show the relative humidity over ice from all dropsondes launched in the central Arctic during the HALO flights on 11–12 April. The mean profile shows an enhanced moisture layer between $4\,\mathrm{km}$ and $8\,\mathrm{km}$. This layer is associated with higher wind speeds up to $25\,\mathrm{m\,s^{-1}}$ and a
shift in wind direction from a easterly flow in the lower $3\,\mathrm{km}$ to a southerly flow in the upper troposphere. Below $4\,\mathrm{km}$ altitude, the air mass was rather dry with relative humidity below $50\,\%$. In these lower altitudes, the stable cold and dry air mass in the central Arctic was supported by the outflow of cold and dry air from northern Greenland and prohibited the formation of clouds below the cirrus. The airborne measurements during this constellation of isolated, widespread cirrus over the sea ice in the central Arctic are well suited to investigate the evolution and radiative effect of Arctic cirrus over sea ice, which is not well
explored and of which only few observations exist (Hong and Liu, 2015; Marsing et al., 2023).

## 5 Conclusions

In this study, we described the weather and sea ice conditions during the HALO–(AC)³ aircraft campaign carried out from 07 March to 12 April 2022 in the North Atlantic sector of the Arctic. Our analysis focused on the three different measurement regions of the campaign: the sea ice covered Arctic Ocean north of Svalbard and Greenland (northern region), the Fram Strait
(central region), and the Greenland and Norwegian Seas between Svalbard and Scandinavia (southern region). Descriptions of the weather pattern were based on the ERA5 reanalysis and climatological context was given with respect to the years 1979–2022. Ny–Ålesund radiosondes and near–surface measurements allowed us to compare the conditions in the flight regions to those at the AWIPEV research base. We described the sea ice concentration (SIC) during HALO–(AC)³ with a high resolution satellite product combining infrared and microwave observations. Climatological context of the SIC was presented with
microwave satellite observations only.

The campaign was separated into a warm period (11–20 March), and a cold period (21 March–12 April). Vertically integrated meridional water vapour transport ($\mathrm{IVT_{north}}$) and the MCAO index $M$ averaged over the central region were used to classify the events of HALO–(AC)³ with highly positive (negative) $\mathrm{IVT_{north}}$ and negative (positive) $M$ during the warm (cold) phase. Within the respective periods, $\mathrm{IVT_{north}}$ and MCAO index $M$ exceeded the 90th percentile of the climatology on five and four
consecutive days, respectively.

The campaign started with a consistent southerly flow, which led to a strong MWAI on 12–13 March that was detected as an Atmospheric River (AR) and reached the central Arctic. It was followed by an even stronger AR on 15–16 March. Compared to the climatology, both ARs denoted higher IVT and were centered further in the north than $75\,\%$ (partly, $90\,\%$) of all ARs in the North Atlantic pathway. During the AR on 15–16 March, we found new records of vertically integrated meridional heat and
water vapour fluxes ($\mathrm{IHT_{north}}$ and $\mathrm{IVT_{north}}$) when averaged over the central region latitudes. The heat and moisture transport of the warm period, dominated by the relatively short–lived MWAIs / ARs, even surpassed the long period of southward transport during the cold period. We therefore stress the importance of MWAIs for the energy exchange between the mid–latitudes and the Arctic. Averaged over the warm period, record breaking rainfall rates were found over the sea ice northwest of Svalbard,



**Figure 14.** (a, c) Three day back trajectories of air masses in which cirrus were sampled on 11 and 12 April 2022. All trajectories are based on ERA5 reanalysis and end at the HALO flight track in cirrus level between 350 and 550 hPa. Surface pressure (thin black lines) and cloud cover (black shading) above 550 hPa from ERA5 are shown for 12 UTC at both days. The thick black line indicates the HALO flight track, and red crosses the location of dropsonde releases. In (b, d) we show the average profile and all individual profiles of relative humidity over ice measured during the two flights by dropsondes.





exceeding the climatology by a factor of 36. Not only is the high deviation remarkable, but also the occurrence of rain over sea

ice at this time of the year.

Starting on 18 March, the southerly flow became disrupted when a surface based high pressure system formed over Greenland. After the passage of a Shapiro–Keyser cyclone over Svalbard on 21 March, a northerly flow was established in the central region, which marked the transition to the cold period. We identified two strong MCAO periods (21–26 March and 01–02 April), where $M$ partly exceeded the 90th percentile. The strong MCAOs were associated with high upward fluxes of sensible

and latent heat in the unstable atmospheric boundary layer, which stress the interaction between the warm ocean and cold atmosphere. The ocean–atmosphere interactions in the boundary layer resulted in the typical high fraction of low–level cloud cover (cloud streets) and convective precipitation. After each strong MCAO period, the northerly flow was replaced by weaker northeasterly flow and aged warm air was advected to the measurement regions (28–31 March, 03–04 April). During the aged warm air period, precipitation faded in the central region due to warming and drying. West of Svalbard, cloud–free conditions

prevailed because of orographic lee effects (foehn).

An upper–level trough approached the central region with a weak MCAO and led to the development of a Polar Low in the Fram Strait on 07 April. We confirmed that four out of six criteria for Polar Low development were fulfilled by inspecting the environmental conditions. With the help of dropsondes launched along a circle, we found strong rotation confined to the lowest $2\,\mathrm{km}$ despite negative relative vorticity in the eastern part related to lee effects. The Polar Low was located unusually far in

the north. At the end of the cold period (10–12 April), warm and moist air was advected by a Scandinavian Low towards the southern region and diverted northwards through the Fram Strait. The high altitude southerly winds transported cirrus clouds over the sea ice with cold and dry air from the Greenland ice sheet below. This combination resulted in an isolated humidity layer including Arctic cirrus in absence of low– and mid–level clouds.

With respect to the climatology, the entire campaign was slightly warmer considering both $2\,\mathrm{m}$ and $850\,\mathrm{hPa}$ temperatures,

except for the central Fram Strait and north of Greenland. Here, $2\,\mathrm{m}$ temperature anomalies were negative. The influence of the ARs was clearly visible in the warm period with $850\,\mathrm{hPa}$ ($2\,\mathrm{m}$) temperatures of up to $9\,\mathrm{K}$ ($16\,\mathrm{K}$), and IWV more than $3\,\mathrm{kg\,m^{-2}}$ above the climatological mean. During the MCAO period, slightly positive $850\,\mathrm{hPa}$ temperature anomalies remained in the central Arctic, but the Fram Strait was up to $5.5\,\mathrm{K}$ colder and $1\,\mathrm{kg\,m^{-2}}$ drier than the climatology.

SICs were well within the 10–90th percentiles of the climatology during the HALO–(AC)$^3$ campaign. In the Fram Strait,

there was a remarkable reduction of SIC after the first AR event on 12–13 March due to dynamical forcing by wind and ocean, and possibly melt. Especially north and northeast of Svalbard, the sea ice cover retreated far towards the north leaving a widely open polynya even until Franz Josef Land. Afterwards during the MCAO period, temperatures stayed below the freezing point and northerly winds increased SIC above the climatological mean in that region. Only around Svalbard and at its southeastern coast, SICs were lower than the climatological mean values. We identified a strip of positive SIC anomaly in the central and

western Fram Strait, indicating that the sea ice edge was further to the south at this location compared to the climatology. The area of positive SIC anomaly coincides well with the negative $2\,\mathrm{m}$ temperature anomalies.

Upper atmospheric soundings and near–surface measurements at Ny–Ålesund generally represent the weather conditions observed in the HALO–(AC)$^3$ flight area well, when considering the temporal difference of up to 1 day due to the propagation



of synoptic systems. The additional measurements at the AWIPEV can be implemented into the data analysis of HALO–
(AC)[3] and provide a long–term climatological context to the airborne measurements. The radiosondes detected unusually high tropopause altitudes up to $12.9\,\mathrm{km}$ during the warm period and extremely dry conditions during the first strong MCAO period with IWV below the 3rd percentile ($1.1\,\mathrm{kg\,m^{-2}}$).

Compared to previous aircraft campaigns in the Arctic within (AC)[3] that focused on the evolution of (mixed–phase) clouds (ACLOUD; AFLUX and MOSAiC–ACA, Wendisch et al., 2019; Mech et al., 2022), we observed a larger variety of MCAO
conditions during HALO–(AC)[3]. The long phase of MCAOs with varying strength and different wind regimes provides opportunities for detailed MCAO studies making use of the airborne measurements. Also the sea ice edge was closer to Svalbard than during AFLUX (March–April 2019) so that ocean–ice transects could be performed more easily. We captured several MWAIs / ARs with unusual or even record breaking strength. With regard to the changing climate when exchanges between the mid–latitudes and the Arctic become more frequent and ARs are expected to shift poleward, the campaign provides a
unique opportunity to study stronger than average MWAIs / ARs. We conclude that the weather conditions were well suited to achieve the objectives of the HALO–(AC)[3] campaign.

*Code and data availability.* All codes used for the analyses presented in this study have been uploaded to GITHUB and connected with ZENODO for public access (*in preparation*). Sea ice concentration climatology data is found on OSI SAF (2017) and Copernicus Climate Change Service (C3S) (2020). The high resolution sea ice concentration dataset used for 07 March–12 April 2022 is based on the product from
the Institute of Environmental Physics, University of Bremen. The data is available at https://seaice.uni-bremen.de/data/modis_amsr2. ERA5 data on single and pressure levels can be accessed through Hersbach et al. (2018b) and Hersbach et al. (2018a). Ny–Ålesund radiosondes and ceilometer data have been published on PANGAEA (Maturilli and Kayser, 2016, 2017; Maturilli, 2020b, 2022). Near–surface meteorology data from Ny–Ålesund is available on PANGAEA as well (Maturilli, 2020a). Dropsonde data from used in this study can be found on ZENODO (*in preparation*).



## Appendix A: Integrated water vapour transport for moist and warm air intrusions

For the MWAIs / ARs on 13 and 15 March, and for the weaker MWAI on 10 April, we show the total IVT in Fig. A1. While the former two events were meridionally aligned, the latter had a rather zonal orientation. The strongest total IVT is found on 15 March, exceeding $400\,\mathrm{kg\,m^{-1}\,s^{-1}}$.

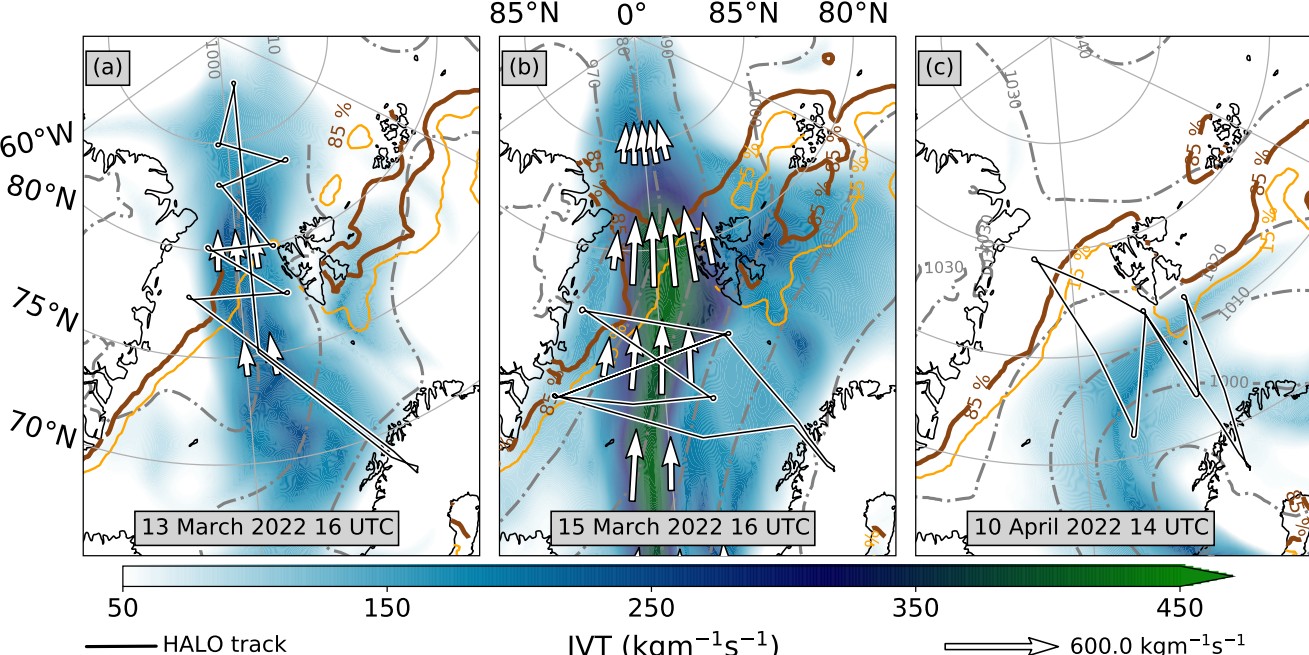

**Figure A1.** Total integrated water vapour transport (IVT, zonal and meridional component) for three moist and warm air intrusions / Atmospheric Rivers: (a) 13 March 2022 16 UTC, (b) 15 March 2022 16 UTC, (c) 10 April 2022 14 UTC. Quivers indicating the flow direction and strength, as well as the HALO flight track (black line with white outline) are also included. The orange (brown) line indicate the 15 % (85 %) sea ice concentration isoline. Grey dash–dotted contours show the mean sea level pressure. Data is based on ERA5.



## Appendix B: Moist and warm air intrusion trajectories

The general air flow on two selected MWAI / AR days is investigated. Three day back and two day forward trajectories shown in Fig. B1 are initialized on 13 and 15 March each at 18:00 UTC. The trajectories reveal that the air masses of the first MWAI day mainly originated from central Europe and the second case from the North Atlantic. Forward trajectories show that both MWAIs / ARs reached the central Arctic.

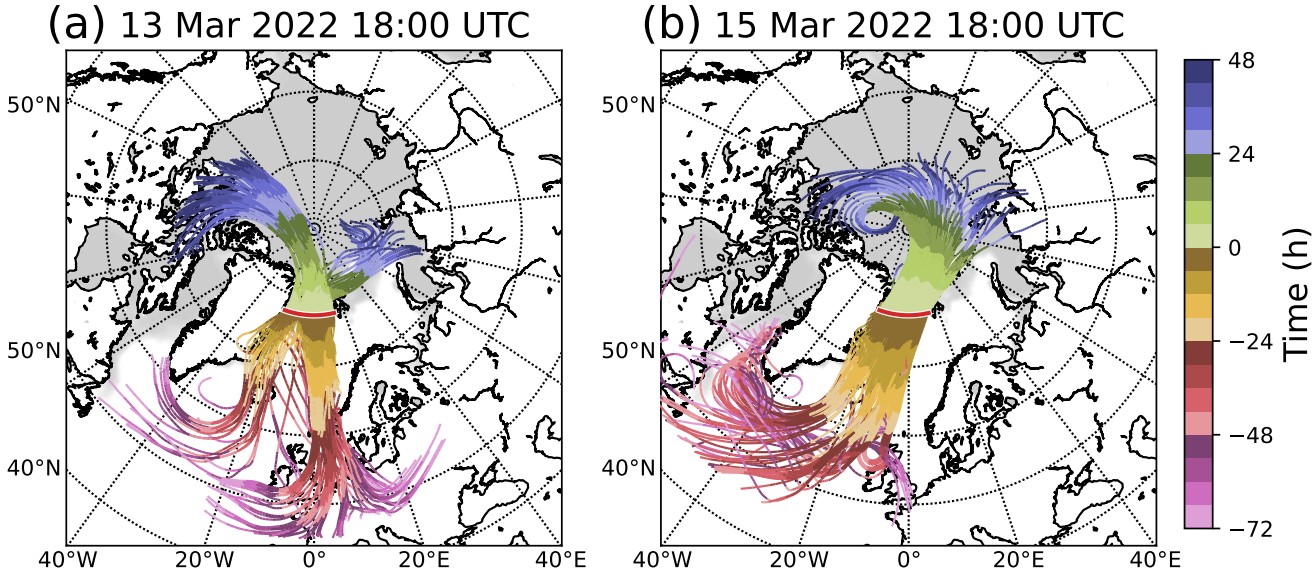

**Figure B1.** Three day backward and two day forward trajectories of the AR air masses observed in the central region on (a) 13 March 2022 18:00 UTC, and (b) 15 March 2022 18:00 UTC. at altitudes of 700, 850, and 925 hPa. Starting points are placed at 77.5° N between 20° W and 13° E with a zonal spacing of 1°. Both 72 hour back–trajectories as well as 48 hour forward–trajectories are computed. Grey shading indicates the sea ice concentration from ERA5.



## Appendix C: Marine cold air outbreak

Figure C1 shows the ERA5 based temporal evolution of temperature, relative humidity and wind for the northern region. During the stronger MCAOs in the central region with northerly winds, the airmasses originated from the northern region. Due to the low occurrence of northwesterly winds, the western extension of the northern region is not included. For comparison of turbulent fluxes between the central and southern region, we include Fig. C2.

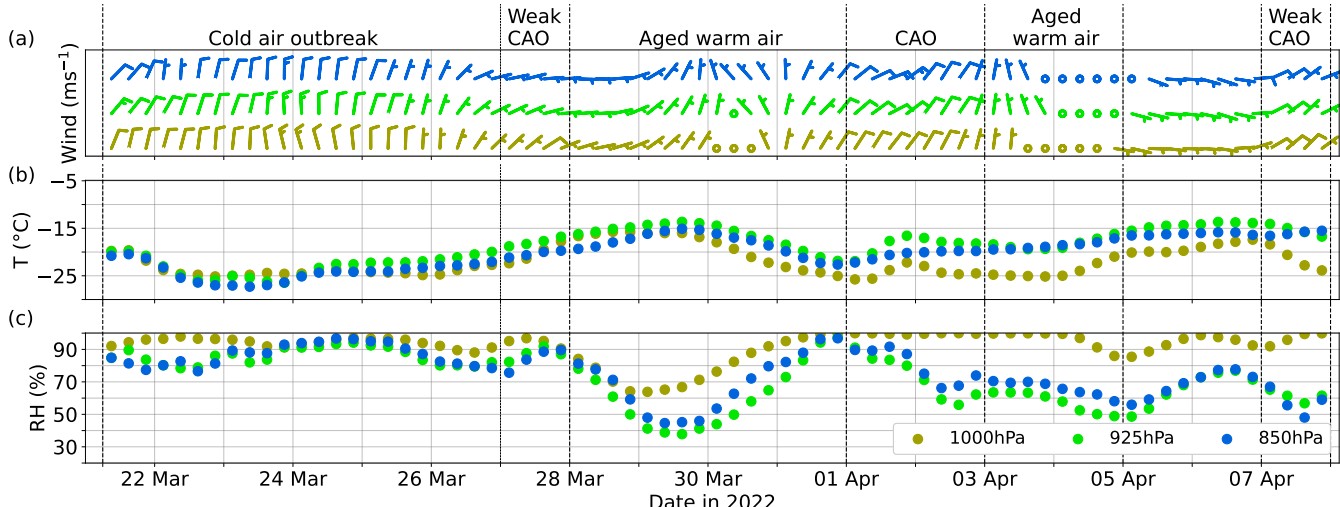

**Figure C1.** Temporal evolution of (a) wind, (b) air temperature (T), and (c) relative humidity (RH) on the 1000, 925, and 850 hPa pressure levels as in Fig. 11, but for the northern region (without western extension). Data is based on ERA5.



**Figure C2.** Temporal evolution of (a) wind, (b) air temperature (T), and (c) relative humidity (RH) on the 1000, 925, and 850 hPa pressure levels, and of (d) low–, (e) mid–, and (f) high–level cloud cover (CC), (g) precipitation, and (h) surface turbulent heat fluxes (STHF) as in Fig. 11 but for the southern region. Data is based on ERA5.



## Appendix D: Polar Low visual observations

Due to the relatively small extent of a Polar Low, we were able to see the entire structure during the HALO research flight on 08 April 2022. The photo was taken on one of the cross sections through the Polar Low, showing the relatively cloud free centre (eye) and the convective cloud band on its (north–)eastern side (Fig. D1).

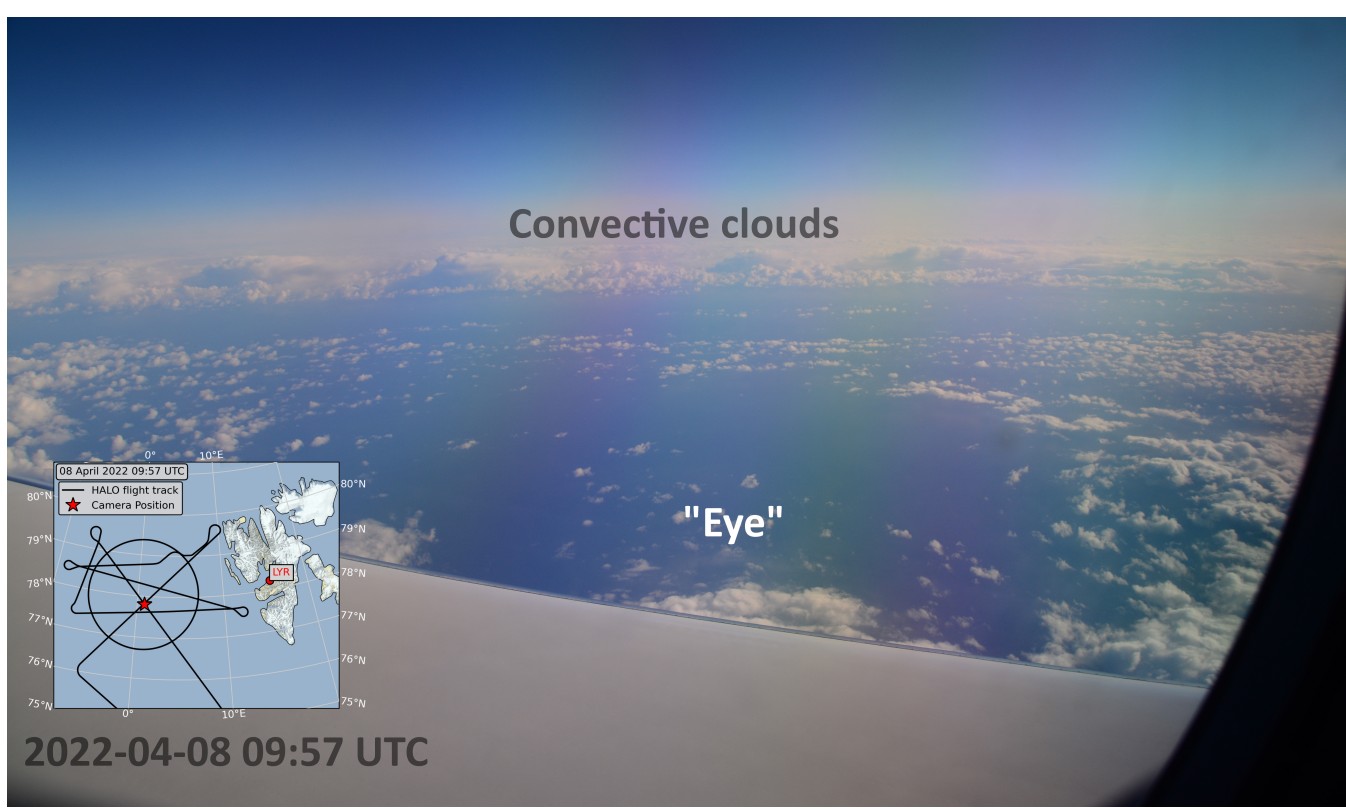

**Figure D1.** Photo of the Polar Low taken from HALO (portside) on 08 April 2022 at 09:57 UTC. The centre (eye) of the Polar Low can be seen in the foreground while the convective cloud band is visible in the background. The map indicates the flight track (black line) and the position of the aircraft at that time (red star). © Andreas Walbröl

*Author contributions.* SC, AE, IG, JanM, MW, and AW conceptualized the manuscript. MW and AW formulated the introduction with comments and input from AE, JanM, and SC. HD, BK, ML, NM, MM, JanR, and AW prepared the description of the data and methods
chapter. SB, HD, BK, ML, JanM, HM, RN, FP, JohR, JanR, IS, NS, and AW provided visualisations and analysed figures. JohM and AW collected codes from co–authors and made them publicly available. AW is the main author of this manuscript and ensured validation. All co–authors reviewed the manuscript.



*Competing interests.* The authors declare that they have no conflict of interest.

*Acknowledgements.* We gratefully acknowledge the funding by the Deutsche Forschungsgemeinschaft (DFG, German Research Foundation) for the ArctiC Amplification: Climate Relevant Atmospheric and SurfaCe Processes, and Feedback Mechanisms (AC)[3] Project Number 268020496 – TRR 172 within the Transregional Collaborative Research Center. We are further grateful for funding of project grant no. 316646266 by the Deutsche Forschungsgemeinschaft (DFG, German Research Foundation) within the framework of Priority Programme SPP 1294 to promote research with HALO. We thank the Institute of Environmental Physics, University of Bremen for the provision of the merged MODIS-AMSR2 sea–ice concentration data at https://seaice.uni-bremen.de/data/modis_amsr2 (last access 2023-04-04). Hersbach

et al. (2018a) and Hersbach et al. (2018b) were downloaded from the Copernicus Climate Change Service (C3S) Climate Data Store. The results contain modified Copernicus Climate Change Service information 2022. Neither the European Commission nor ECMWF is responsible for any use that may be made of the Copernicus information or data it contains. We thank the Alfred Wegener Institute for providing and operating the two aircraft (Polar 5 and Polar 6), the crew, and also the technicians of the Polar 5 and Polar 6 aircraft. We are also grateful for the research aircraft HALO, the pilots and technicians provided and operated by the German Aerospace Centre (Deutsches

Zentrum für Luft– und Raumfahrt).



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
