# Peer review of "Environmental conditions in the North Atlantic sector of the Arctic during the $HALO-(AC)^3$ campaign"

_EGUsphere, 2023_

## Referee Comment (RC1)

This paper presents a detailed account of the weather and sea-ice conditions experienced during te HALO-AC3 campaign. I sympathize with writing such an account; it is very useful as a reference for future work, but at the same time in a scientific journal – rather than a data journal – it should have some science in it to motivate the publication in a scientific journal. This often becomes a compromise and the factor that often suffer is the length and the scientific narrative. That is also the case with this manuscript, which is much too long and unfocused; it is unclear if the paper is describing methods and measurements or – as is claimed in the title – the meteorological conditions during the campaign.

Therefore, I am recommending a major revision focusing on reducing the details on how the different analyzes were obtained, minimizing the repetition of unnecessary information and streamlining the language.

Major concerns

The most stressing concern is the length, the degree of detail and lack of focus. There are to much too many details that would be better suited in special papers dealing with the different aspects whether that be analysis methods or measurement details. Combined with the rather "flowery" language, where the same thing is not rarely and unnecessarily described in more than one wording makes the reading tiresome; I must confess I gave up reading around page 30 or so. It just has much to many details that are better suited in topic specific analysis papers.

The data and methods section is (5.5 pages) is much to detailed for this scope of this paper and should be shortened by 50%; I'm, sure just condensing the language could do at least half of that. It has an "everything but the kitchen sink" character. For the most important measurement asset – the HALO aircraft – only the dropsondes are discussed (lines 100-104) while the measurements at Ny-Ålesund are twice that long and not really needed; I'm sure these are described elsewhere and can be referenced. The fact that only sea-points where used in various analyses are repeated at least three times; once is enough. Definitions of ARs and MCAOs is also much to detailed and the discussion of the circles flown to estimate vorticity is not nearly enough to really understand how but way too much given how this is used in the upcoming sections.

The painstaking day-to-day-account of the synoptic development on page 11-24 (14 pages!) should be condensed to its main components and shortehed to 30% of the present length. The only section that should actually be longer is the comparison to climatology; this is very useful for papers to come. The Ny-Ålesund section is much to long; I think this paper does not really need it and it could be dropped all together.

The section on specific events is what saves this paper; still at 8 pages also this could probably also be shortened.

Some detailed concerns:

Line 11: Mentioning "Shapiro-Keyser cyclone" in the abstract is complete overkill; I bet less than a third of all potential readers have any clue what this means for the results.

Line 14-15: Isn't it natural that conditions during *any* AR would be warmer than climatology?

Line 15: What is significant in the statement that the SIC was within the 10-90 percentiles; that covesr almost everuything, doesn't it?

Line 31: The connection between a slightly weaker jet stream and a more meandering flow is far from well established; suggest inserting "possibly" somewhere in this sentence.

Line 38: The statement about warm air gliding up on a cold dome is very popular in some circles, yet I would say it is false. If it were true, what happens to the air under the dome over time? I presume it can flow out of the Arctic during MCAOs, but apparently not be replenished by ARs? Wouldn't that be contrary to having a dome in the first place? Instead – as what the hole campaign was designed to study – warm air flowing into the Arctic is transformed to Arctic air by interactions with the surface.

Line 43: All ARs are not "extreme"; suggest using "large" instead.

Line 56: This is a problem not only for climate models; moreover, the Pithan reference argues for the Lagrangian methods applied in HALO-AC#, but provides no evidence for how this is modeled – poorly or otherwise.

Line 62-63: The wording "does not permit" is too strong. A Lagragian method does not by itself ascertain proper observation of the transformation and multiple Eulerian observations along a trajectory may provide some transformation information. Its not black or white…

Table 1: With the figure, this table is not necessary.

Line 100-103: Why this degree of detail for the dropsondes? Not necessary in this paper.

Line 119-126: Too much detail; surely there is a reference!

Line 141: IWV is not really a "basic variable".

Line 142: Strictly speaking this means that all grid points where excluded, since "$\geq 0$" means "larger than or equal to zero". So if land fraction is zero, implying ocean only, it would also be excluded.

Line 144-145: The "north" subscript is confusing and probably unnecessary. The way this is calculated makes northward transport of excess heat or moisture by definition positive; southward negative. Including this subscript raises the question of you ignore southward fluxes.

Line 155 & 161: Why different units?

Line 164-166: Don't understand; if the bar is too high for an event, then you raise the bar?

Line 173: Excluding land points again.

Line 175: Why use temperature to indicate sea ice? There is sea ice in the model output.

Line 183: And excluding land points a third time.

Line 204-206: Unclear: First, the definition of the gradient is pretty obvious and doesn't have to be described. Second, the potential temperature can increase in the layer even if the average is zero, since the gradient is probably < 0 close to the surface or there wouldn't be any convection.

Line 212-213: The gustiness parameterization has nothing to do with the resolution; it is does to turbulence, which you need an LES to resolve.

Table 2: With the text, this table is not necessary; alternatively use the table a do not repeat the details in the text.

Line 218-224: Do we need this description? I can't see that vorticity is used in the description, and moreover, this description is not enough to really understand what you did but way too much for this paper.

Line 268-270: This sounds a bit too simple to be the whole truth, that the delay in surface warming is just because of the slope of the warm front; at least you show this is the case – or drop the argument.

Line 289-281: Drop "records"; this is not a championship.

Line 283-284: Don't understand the caveat; ist it or isn't it and why?

Line 295: I suggest "indicating" instead of "illustrating", since you don't show this.

Line 300-307: Why bring in the Shapiro-Keyser classification? Is it relevant an if so, how is it relevant? I bet a majority of readers doesn't even know what this is.

Line 306-307: Don' t understand; if the heat content is low, why is the meridional heat transport not negative?

Line 326: Suggest "dissipating" rather than "being filled up".

Figure 5: It strikes me that Figure 5 is underutilized; drop it or use it more. Why the change in tilt on 21 March?

---

## Referee Comment (RC2)

Review of : environmental conditions in the N. Atlantic sector of the Arctic during the HALO-AC3 campaign.
Walbrohl et al.
Egusphere-2023-668

This is a generally well-written presentation on the meteorology experienced during the HALO-AC3. I enjoyed reading it and am confident it will be a useful contribution to the larger research community. I like many of the figures.

My main comment is that I rather wished for more description of how well the ERA5 reanalysis can be trusted for this part of the world. The analysis relies heavily on ERA5, including for cloud and precipitation phase. Has there been any comparison of the ERA5 products to the in-situ data yet? It may still be early stages for this, but some of the drop sonde quantities provided in the manuscript would be very easy to compare against ERA5. A literature review of other assessments is mostly absent, other than some in section 2.4.4. My own cursory web search revealed at least these two: Seethala et al., 2021; Loeb et al., 2022, but I would expect there to be more. I was hoping to see a more systematic assessment of the ERA5 quantities. Was the data assimilation consistent for the entire timespan of the ERA5 climatology, so that statements about 'maximum records' (e.g. line 281) are fair to make?

The other main comment is that in several places there are references to place names whose geography the reader may not be aware of, such as Franz Josef land. Fig. 1 might be a place to add some helpful geographic annotations, also for the ocean basins (Fram Strait, Greenland Sea, Barents Sea), as is section 2.1

Specific comments:

Abstract: The acronym ERA5 is spelled out, but that for HALO-AC3 is not. My guess is that more readers will know what ERA5 is, than HALO-AC3. I'd suggest spelling out HALO-AC3 and seeing if the journal will accept ERA5 as is.
Abstract, line 10: include years of the ERA5 climatology.
Abstract, line 11: not a good idea to expect the reader to know what a 'shapiro-keyser' cyclone is, you can leave out the name reference
Abstract, line 16: 'untypically' => 'atypically'

Intro, line 32: my recollection is that the Francis and Vavrus, 2015, was highly debated after it was published, leading to a US CLIVAR report, and spurring other work by e.g. E. Barnes at CSU. A bit more detailed literature review here would make this portion more impactful.

 Line 67: space between performance of

Line 120: Nimbus -> Nimbus

Lines 161-164: this is slightly confusing as written. Do Guan and Wailer use a IVT threshold of 100 kg/m/s and you use 50? Maybe combine those two phrases into one sentence if so.

Line 212: how is the polar low's center determined.
Line 212: using the max 10 m wind gust as opposed to the mean wind assumes ERA5 underestimates polar low wind gusts…do you know this for sure?

Line 218: the drop sonde vorticity calculation: at what time? What was the center of the drop sonde circle? The vorticity calculation should be easy to compare to that from ERA5, how does ERA5 do?

Line 248: southerly winds not obvious for the central region in fig. 3d…would suggest removing 'and central'.

Lines 248-255: it's hard to visualize what you are saying just from fig. 3, would suggest adding in some spatial circulation figures like what you have within fig. 4.

Line 278: the stated maximum IVT_north of 388 kg/s/m doesn't seem consistent w Fig. 2. Is that because the maximum is an hourly-mean?

Line 280-282: what's the difference between 'latitude-averaged' and 'area-averaged' IVT?

Line 283: how are you defining MWAI intensity? Winds?

Line 294: I don't follow "The moisture flux decreased faster than the heat flux". Is this from the atmosphere to the ocean? Or the turbulent fluxes coming off of the ocean?

Line 302: "frontal structure representative of a Shapiro-Keyser cyclone".  Better to just describe the frontal structure, as many readers, including myself, will not know what you are talking about.

Line 385: it could be interesting to discuss how the subsidence is evolving as well, as that would also influence the static stability.

Line 415: not fully following how surface conditions explain a high tropopause height. I think you can just say 'vertical advection lifts the tropopause to 12.9 km' and be done with it.

Line 425: these radiosonde profiles are also an opportunity to assess the corresponding ERA5 profiles.

Line 450: '2023) that' => '2023), '

Line 527: it should be relatively straightforward to figure out if the latent heat fluxes are increases because q_sat-q_air is increasing or because the wind speeds are increasing. How much is the q_sat increasing? I would think the SST would not be changing all that much?

Line 565: 'lied' => 'lay'

Line 573: please include a figure comparing the drop sonde vorticity and wind speed profiles to that from ERA5. The drop sonde circle should also give you a divergence profile and updraft speed that can be compared to that from ERA5 in a figure. Please do so.

Line 605: 'a' => 'an' (in front of easterly)

Line 653: insert 'the' before 'absence'

Figures/Tables

Table 1: the northern part of the northern region is hard to understand initially from this table, however, figure 1 shows the study area very well. I had to look down to figure 1 to understand

the table. It might be best to either have the figure before the table, or to add a 4$^{th}$ column to the table for the northern part of the northern region

Fig. 1: spell out what NYA means in the caption.

Fig. 3: the graphic on the left looks a bit odd, as there is no real need for us to know the day of the week I don't think. I would suggest adding a color bar to the top of the right-hand panels that has the identification information.

Fig. 4: include dates in caption.

Fig. 5: are these hourly values? Would be worth putting in caption.

Fig. 8: some strange overlapping of lat/lon labels in a-c, would suggest just removing a few.

Fig. 9, caption: I don't understand the last sentence, and both the top and far right histograms need a basic description. Also, the values shown in here for

Fig. 10: fewer lat/lon labels and bigger plots would be nice. You could just leave most of the lat labels out.

Fig. 13: including the ERA5 vorticity values on panel a would be nice as would be an additional plot showing the wind speeds/divergences. Panel b needs SLP labels.

References:

Loeb et al., 2022: https://www.frontiersin.org/articles/10.3389/fenvs.2022.866929

Seethala et al., 2021: https://agupubs.onlinelibrary.wiley.com/doi/pdfdirect/10.1029/2021GL094364

---

## Author Comment (AC1)

**Reviewer #1 reply:**

We thank the reviewer for the detailed comments. They helped us to get the message of the manuscript in a more concise and focused way without removing necessary details. Below, we repeat the reviewer's comments in black and write our responses in blue. The line numbers in the line-by-line responses are valid for the revised manuscript.

This paper presents a detailed account of the weather and sea-ice conditions experienced during te HALO-AC3 campaign. I sympathize with writing such an account; it is very useful as a reference for future work, but at the same time in a scientific journal – rather than a data journal – it should have some science in it to motivate the publication in a scientific journal. This often becomes a compromise and the factor that often suffer is the length and the scientific narrative. That is also the case with this manuscript, which is much too long and unfocused; it is unclear if the paper is describing methods and measurements or – as is claimed in the title – the meteorological conditions during the campaign.
Therefore, I am recommending a major revision focusing on reducing the details on how the different analyzes were obtained, minimizing the repetition of unnecessary information and streamlining the language.

➢ We moved details from the methods section to the appendix because we think that having some details available might be helpful for full reproducibility of the study while keeping the main body more concise. We also reduced the length of chapter 3.

Major concerns
The most stressing concern is the length, the degree of detail and lack of focus. There are to much too many details that would be better suited in special papers dealing with the different aspects whether that be analysis methods or measurement details. Combined with the rather "flowery" language, where the same thing is not rarely and unnecessarily described in more than one wording makes the reading tiresome; I must confess I gave up reading around page 30 or so. It just has much to many details that are better suited in topic specific analysis papers.

➢ We agree that the manuscript was too unfocused and described too many small details. Therefore, we reduced the text length in the revised manuscript to bring the message across more efficiently. We would like to keep the description of all relevant weather events as this manuscript aims to be a comprehensive overview of the HALO-(AC)³ weather conditions. Duplicates of descriptions have been removed.

The data and methods section is (5.5 pages) is much to detailed for this scope of this paper and should be shortened by 50%; I'm, sure just condensing the language could do at least half of that. It has an "everything but the kitchen sink" character. For the most important measurement asset – the HALO aircraft – only the dropsondes are discussed (lines 100-104) while the measurements at Ny-Ålesund are twice that long and not really needed; I'm sure these are described elsewhere and can be referenced. The fact that only sea-points where used in various analyses are repeated at least three times; once is enough. Definitions of ARs and MCAOs is also much to detailed and the discussion of the circles flown to estimate

vorticity is not nearly enough to really understand how but way too much given how this is used in the upcoming sections.

> Thank you for identifying sections where details can be removed. We reduced the length of the methods section and kept only the most important descriptions. We would like to keep the Ny-Alesund measurement descriptions so that the reader gets a brief overview of the measurements used in the analysis. The Atmospheric River tracking algorithm has been replaced by references to literature.

The painstaking day-to-day-account of the synoptic development on page 11-24 (14 pages!) should be condensed to its main components and shortehed to 30% of the present length. The only section that should actually be longer is the comparison to climatology; this is very useful for papers to come. The Ny-Ålesund section is much to long; I think this paper does not really need it and it could be dropped all together.

> We agree that a description of the development day-by-day is too much for a scientific journal. We therefore condensed the weather development description without losing the core information. To avoid adding more length to the manuscript, we did not add information here. However, due to the reduction of the synoptic description, the balance should be improved. We also removed details in the Ny-Alesund but keep it in the manuscript as the additional measurements form this research station might be included in future HALO-(AC)³-related studies. Then having the connection of the weather conditions between the central measurement region of HALO-(AC)³ and Ny-Alesund would prove beneficial.

The section on specific events is what saves this paper; still at 8 pages also this could probably also be shortened.

> We agree that the section is also too long and tried to shorten it.

Some detailed concerns:
Line 11: Mentioning "Shapiro-Keyser cyclone" in the abstract is complete overkill; I bet less than a third of all potential readers have any clue what this means for the results.
> We changed it to "a strong cyclone" (line 10 of the revised manuscript).

Line 14-15: Isn't it natural that conditions during *any* AR would be warmer than climatology?
> True, we rephrased it to: "due to the strong influence of the ARs " to set the focus more clearly on the effect of ARs (line 12-13).

Line 15: What is significant in the statement that the SIC was within the 10-90 percentiles; that covesr almost everuything, doesn't it?
> Here, we wanted to express, that SIC was not extremely low or high, but rather normal. We rephrased it to "the sea ice concentration (SIC) was well within the climatological variability, staying within the 10-90th percentiles over the campaign duration" (line 13-14).

Line 31: The connection between a slightly weaker jet stream and a more meandering flow is far from well established; suggest inserting "possibly" somewhere in this sentence.

> ➢ Agreed, "possibly" has been inserted (line 28).

Line 38: The statement about warm air gliding up on a cold dome is very popular in some circles, yet I would say it is false. If it were true, what happens to the air under the dome over time? I presume it can flow out of the Arctic during MCAOs, but apparently not be replenished by ARs? Wouldn't that be contrary to having a dome in the first place? Instead – as what the hole campaign was designed to study – warm air flowing into the Arctic is transformed to Arctic air by interactions with the surface.

> ➢ We rephrased it to avoid the confusion with the Arctic cold air dome: "When the warm air is pushed upwards over cold Arctic air masses, deep cloud... " (line 37-38)

Line 43: All ARs are not "extreme"; suggest using "large" instead.

> ➢ Agreed, "extreme" certainly sends a wrong message here and is not appropriate. We replaced it by "strong" (line 42).

Line 56: This is a problem not only for climate models; moreover, the Pithan reference argues for the Lagrangian methods applied in HALO-AC#, but provides no evidence for how this is modeled – poorly or otherwise.

> ➢ We were unsure how to respect this comment in the revision. We would be grateful for an elaborated comment.

Line 62-63: The wording "does not permit" is too strong. A Lagragian method does not by itself ascertain proper observation of the transformation and multiple Eulerian observations along a trajectory may provide some transformation information. Its not black or white…

> ➢ Agreed. We changed it to: "To observe air mass transformation processes along their meridional pathway in a Eulerian view, multiple research stations that are exactly aligned with the wind direction would be needed." (line 61-62).

Table 1: With the figure, this table is not necessary.

> ➢ For full reproducibility, the exact coordinates might be helpful because i.e., the southern limit of the southern region (70.6 °N) cannot accurately be determined from the figure. We shifted this table to the new Appendix A.

Line 100-103: Why this degree of detail for the dropsondes? Not necessary in this paper.

> ➢ We agree that this can be erased and therefore removed details the specification of measurement accuracies (line 93-96).

Line 119-126: Too much detail; surely there is a reference!

> ➢ In general, we agree. But, we think that at least the product names should be included in the main text. The description of the sensors has been removed from the main text (line 106-115).

Line 141: IWV is not really a "basic variable".

> ➢ To reduce the length of the manuscript, we removed this subsection (former 2.4.1) as the data processing was minor and already written in section 3.1. Nevertheless, we removed "basic" in line 184 (section 3.1).

Line 142: Strictly speaking this means that all grid points where excluded, since ">0" means "larger than or equal to zero". So if land fraction is zero, implying ocean only, it would also be excluded.

➢ Well spotted typo. We meant ">0", instead of ">=0" and therefore changed it (line 125).

Line 144-145: The "north" subscript is confusing and probably unnecessary. The way this is calculated makes northward transport of excess heat or moisture by definition positive; southward negative. Including this subscript raises the question of you ignore southward fluxes.

➢ IVT_north (or sometimes also vIVT because of the meridional wind, which is often termed "v") is a common expression for meridional moisture transport in the northern hemisphere. With "north", we just wanted to say that in this case, we did not consider zonal transports, just meridional. We rewrote it to "Woods and Caballero (2016) detects moist air intrusions into the Arctic when the vertically integrated meridional moisture flux (IVT_north) at 70 °N exceeds ... " (line 130-131).

Line 155 & 161: Why different units?

➢ We directly used the IVT product provided in ERA5 data, which is in kg m-1 s-1, while they used a similar but not identical product. When converting the Tg day-1 deg-1 to kg m-1 s-1, the threshold Woods and Caballero 2016 used is 60.6 kg m-1 s-1. We added this information: "... at 70 °N exceeds 200 Tg d-1 deg-1 (60.6 kg m-1 s-1) over a duration ... " (line 131).

Line 164-166: Don't understand; if the bar is too high for an event, then you raise the bar?

➢ For brevity, we removed the AR detection algorithm description because it is sufficient to refer to literature here. In the old manuscript, we only described the way the revised version of the Guan and Waliser algorithm works. It makes sure that not every wide blob of WAI is detected as AR, which is supposed to have a certain geometric shape. A strong WAI may embed an AR, which can be found when increasing the water vapour transport percentile threshold.

Line 173: Excluding land points again.

➢ We kept this information because this sea ice mask is different than the others to keep it consistent with Dahlke et al. (2022).

Line 175: Why use temperature to indicate sea ice? There is sea ice in the model output.

➢ We kept the sea ice mask consistent with Dahlke et al. (2022). We asked them why they decided for skin temperature instead of the ERA5 sea ice model output. It was merely for convenience reasons as the skin temperature is used in the computation of the index anyway. They found that the differences between the skin temperature based and sea ice concentration based mask were negligible.

Line 183: And excluding land points a third time.

➢ Has been removed.

Line 204-206: Unclear: First, the definition of the gradient is pretty obvious and doesn't have to be described. Second, the potential temperature can increase in the layer even if the

average is zero, since the gradient is probably < 0 close to the surface or there wouldn't be any convection.

➢ We agree that there were too many details regarding the methodology. Therefore, the formula-based description of the vertical potential temperature gradient has been removed (line 154-159). We considered vertical mean, max and min to get an idea of the range of the vertical potential temperature gradient but showed the vertical mean only for brevity. Indeed, the vertical minimum of the gradient is < 0 in a large fraction of the 200 km circle around the Polar Low's centre, indicating convection.

Line 212-213: The gustiness parameterization has nothing to do with the resolution; it is does to turbulence, which you need an LES to resolve.

➢ Here, we wanted to say that the ERA5 10 m mean wind might not capture the maximum wind speed of the Polar Low well because of ERA5 has got a relatively coarse resolution. Therefore, we rather considered the gust to have a more realistic view on winds on sub-ERA5-grid scale. We did not intend to have it understood as if the gustiness depends on resolution. It has been changed to: "We decided to use the maximum 10 m wind gust instead of mean wind to get a better estimate of the near-surface wind field that might be hidden due to the coarse resolution of ERA5." (line 160-162)

Table 2: With the text, this table is not necessary; alternatively use the table a do not repeat the details in the text.

➢ We cut the details in the text and moved the table to section 4.3, where we added another column indicating whether a condition is fulfilled or not. The presence of the table in section 4.3 might be convenient as the reader is reminded of the meanings of the acronyms C1-C6.

Line 218-224: Do we need this description? I can't see that vorticity is used in the description, and moreover, this description is not enough to really understand what you did but way too much for this paper.

➢ It is correct that the vorticity is not used in the synoptic description. However, in section 4.3 we compare dropsonde vorticity estimates to ERA5 model output (line 445-455). We reduced the details of the description in the manuscript and refer to literature (line 166-168).

Line 268-270: This sounds a bit too simple to be the whole truth, that the delay in surface warming is just because of the slope of the warm front; at least you show this is the case – or drop the argument.

➢ We dropped the argument.

Line 289-281: Drop "records"; this is not a championship.

➢ This part has also been rewritten for brevity and now reads as: "Simultaneously, the latitude-averaged IHT_north and IVT_north exceeded the previous maxima from 1996 (9.44 * 10^10 W m-1 vs. 9.32 * 10^10 W m-1, and 388 kg m-1 s-1 vs. 384 kg m-1 s-1, Fig. 5)" (line 210-211)

Line 283-284: Don't understand the caveat; ist it or isn't it and why?

➢ We wanted to mention that we only focussed on certain regions (those boxes), but not the entire Arctic. Other regions may have experienced stronger MWAIs in that time period but we didn't have a look at those other regions. We dropped the caveat as it does not provide information relevant for the key message of the manuscript.

Line 295: I suggest "indicating" instead of "illustrating", since you don't show this.
➢ Agreed. We rephrased it to: "After the AR, much drier but still relatively warm air followed, leading to a strong reduction in IVT_north and a slight reduction in IHT_north (Fig. 5). " (line 216-217)

Line 300-307: Why bring in the Shapiro-Keyser classification? Is it relevant an if so, how is it relevant? I bet a majority of readers doesn't even know what this is.
➢ We understand that not every reader might be familiar with the term but during the campaign, this event was always called "Shapiro-Keyser" cyclone. In upcoming studies, it might be that this term would also be used as it represents a turning point in the campaign and would be lost in other cyclones if this classification was removed. We dropped the brief description of the Shapiro-Keyser cyclone characteristics (line 219-221) for brevity as this can be found in the literature.

Line 306-307: Don' t understand; if the heat content is low, why is the meridional heat transport not negative?
➢ For brevity, this sentence has been removed as it only provided details of minor importance. Heat transport was negative but not strongly negative. Large negative values require large amounts of heat being transported southwards. If no (or extremely small amounts of) heat is transported southwards (for example during cold temperatures), we'll have only slightly negative values.

Line 326: Suggest "dissipating" rather than "being filled up".
➢ Agreed. Has been changed to " As the Shapiro-Keyser cyclone stayed over the Barents Sea while dissipating, IWV dropped..." (line 240).

Figure 5: It strikes me that Figure 5 is underutilized; drop it or use it more. Why the change in tilt on 21 March?

➢ We understand that this figure might appear underutilized. However, we would like to keep this figure as it shows the longitudinal position of the meridional air mass transports and the record breaking IVT_north and IHT_north values. For brevity, the heat fluxes are only discussed around the main synoptic events. Regarding the question: Do you refer to the tilt of the contour lines? They basically depend on the wind regime: If we have weak westerlies, the tilt would be many degrees from the horizontal. For totally meridional winds, the contour lines of certain features (WAI) would be vertical (because of missing zonal propagation). For extremely strong westerlies, the contour lines of a feature would be quite horizontal.

---

## Author Comment (AC2)

**Reviewer #2 reply:**

We thank the reviewer for the supportive revision of the manuscript. We appreciate the detailed comments about the usage of ERA5 and its potential performance issues in the Arctic. We revised the manuscript including a more thorough literature research regarding ERA5 performance and adding minor observation-reanalysis comparisons. Below, we repeat the reviewer's comments in black and write our response in blue. The line numbers in the responses are valid for the revised manuscript.

This is a generally well-written presentation on the meteorology experienced during the HALO-AC3. I enjoyed reading it and am confident it will be a useful contribution to the larger research community. I like many of the figures. My main comment is that I rather wished for more description of how well the ERA5 reanalysis can be trusted for this part of the world. The analysis relies heavily on ERA5, including for cloud and precipitation phase. Has there been any comparison of the ERA5 products to the in-situ data yet? It may still be early stages for this, but some of the drop sonde quantities provided in the manuscript would be very easy to compare against ERA5. A literature review of other assessments is mostly absent, other than some in section 2.4.4. My own cursory web search revealed at least these two: Seethala et al., 2021; Loeb et al., 2022, but I would expect there to be more. I was hoping to see a more systematic assessment of the ERA5 quantities.

> We performed a more dedicated literature research regarding the performance of ERA5 in the Arctic (line 116-123). For the HALO-(AC)³ campaign, ERA5 data can probably be more trusted than in other years because dropsondes have been assimilated. We also added this information to the manuscript: "During HALO-(AC)³, dropsonde measurements launched from HALO were assimilated into ERA5." (line 123-124). We briefly compared ERA5 IVT of the Atmospheric Rivers to dropsondes (line 369-370 and Appendix B), and IWV from Ny-Alesund radiosondes to ERA5 (line 309-311). For brevity and to focus on the analysis of the weather events and their climatological context, we keep the comparison of ERA5 with observations to a minimum. Regarding the precipitation phase: It is a known problem due to the lack of observations in this area. Here, we can say that we observed liquid precipitation over the sea ice with HALO's cloud radar during the Atmospheric River events. We added this information to the conclusion to ensure the reader that ERA5 can be trusted in this case: "Liquid precipitation over sea ice was also observed by the cloud radar onboard HALO during research flights. " (line 503-504)

Was the data assimilation consistent for the entire timespan of the ERA5 climatology, so that statements about 'maximum records' (e.g. line 281) are fair to make?

> ERA5 assimilates various sources of information and the used number of observational data has increased steadily throughout time. Therefore, the quality of

reanalysis in general has increased over time especially as more satellite data are assimilated. However, we are not aware of any better climatological information on the Arctic and using reanalyses has become an established approach.

The other main comment is that in several places there are references to place names whose geography the reader may not be aware of, such as Franz Josef land. Fig. 1 might be a place to add some helpful geographic annotations, also for the ocean basins (Fram Strait, Greenland Sea, Barents Sea), as is section 2.1

> ➤ We agree that the reader can much better follow our analysis when we point out the major land marks in Figure 1. Therefore, we added the ocean basin labels. Franz Josef land has not been included to avoid the figure to become overloaded.

Specific comments:

Abstract: The acronym ERA5 is spelled out, but that for HALO-AC3 is not. My guess is that more readers will know what ERA5 is, than HALO-AC3. I'd suggest spelling out HALO-AC3 and seeing if the journal will accept ERA5 as is.

> ➤ We would like to spell out the acronym HALO-(AC)³ but due to its length we decided to explain it in the introduction instead. It would be required to explain both the acronym behind the research aircraft HALO and the science project (AC)³.

Abstract, line 10: include years of the ERA5 climatology.

> ➤ The years of the ERA5 climatology is embedded in the abstract as follows: "Compared to the ERA5 climatology (1979-2022), record breaking vertically integrated poleward heat and moisture fluxes ..." (line 7-8)

Abstract, line 11: not a good idea to expect the reader to know what a 'shapiro-keyser' cyclone is, you can leave out the name reference

> ➤ We agree that the detailed characteristics of this cyclone are not needed here and thus replaced it by "strong cyclone" (line 10).

Abstract, line 16: 'untypically' => 'atypically'

> ➤ Agreed, we changed it in the revised sentence, which should also express the message more efficiently: "However, during the warm period, an atypically large polynya opened north of Svalbard." (line 14-15).

Intro, line 32: my recollection is that the Francis and Vavrus, 2015, was highly debated after it was published, leading to a US CLIVAR report, and spurring other work by e.g. E. Barnes at CSU. A bit more detailed literature review here would make this portion more impactful.

> ➤ We extended the literature review regarding the impact of climate change on the jet stream: "A more meandering jet would result in an increasing number of poleward

moist and warm air intrusions (MWAIs) and southward cold air outbreaks (CAOs). However, the tropical upper troposphere warms while the Arctic lower stratosphere cools, reducing  meridional temperature gradients at higher altitudes (Lee et al., 2019, Stendel et al., 2021). The frequency of meridional transport through the North Atlantic has increased during the last decades while it stayed constant or even decreased in other regions (Mewes and Jacobi, 2019). You et al. (2022) found a positive trend in the frequency and duration of atmospheric blocking over the Barents Sea especially in winter, supporting the statement of an enhanced North Atlantic pathway for meridional transport. " (line 29-35)

Line 67: space between performance of

➢ Well spotted. Space has been added.

Line 120: Nimbus -> Nimbus

➢ Thank you for spotting also this typo, but we removed the description of satellite sensors for brevity.

Lines 161-164: this is slightly confusing as written. Do Guan and Wailer use a IVT threshold of 100 kg/m/s and you use 50? Maybe combine those two phrases into one sentence if so.

➢ Indeed, for the Arctic, we use the threshold 50 kg m-1 s-1 while it is 100 kg m-1 s-1 in the original Guan and Waliser revised Atmospheric River detection algorithm. We rephrased the second sentence to: "In this study, ARs were identified with a global algorithm by Guan and Waliser (2015) in its revised version (Guan et al., 2018), adapted to the lower moisture content of the Arctic (Lauer et al., 2023)." (line 134-136)

Line 212: how is the polar low's center determined.

➢ Thank you for pointing out that we missed giving this information. We added "pressure minimum" in paranthesis to clarify this: "We analyse the environment for Polar Low formation with a set of conditions (C1--C6) suggested by Radovan et al. (2019) and Terpstra et al. (2016) in a 200 km radius around the Polar Low's centre (pressure minimum):" (line 154-155)

Line 212: using the max 10 m wind gust as opposed to the mean wind assumes ERA5 underestimates polar low wind gusts…do you know this for sure?

➢ We did not intend this impression and realized that this could be formulated more clearly. We did not use mean wind because this quantity is expected to yield lower wind speeds that actually present at some places in the Polar Low due to ERA5's coarse resolution. Using either a higher resolution reanalysis (like CARRA) or using wind gusts instead capture the mean wind speed in a rather small scale feature like a Polar Low better. We rephrased it to: "We decided to use the maximum 10 m wind

gust instead of mean wind to get a better estimate of the near–surface wind field that might be hidden due to the coarse resolution of ERA5. " (line 160-162)

Line 218: the drop sonde vorticity calculation: at what time? What was the center of the drop sonde circle? The vorticity calculation should be easy to compare to that from ERA5, how does ERA5 do?

> The dropsondes have been launched between 06:55 and 07:53 UTC on 08 April. This information has been added to chapter 4.3, where also other time information regarding the Polar Low is given: "Dropsonde measurements between 06:55 and 07:53 UTC show high values of relative vorticity in the lowest 2 km and above 6 km, indicating cyclonic rotation." (line 445-446). We added a brief comparison of the ERA5 vorticity to the dropsonde vorticity: "When averaging ERA5 vorticity over the grid points closest to the dropsonde positions, we find a disagreement to the dropsonde measurements below and good agreement above 4 km height. (...)The disagreement between ERA5 and the dropsondes could be due to a misrepresentation of the Polar Low's wind field in the reanalysis or due to spatio-- temporal mismatches of its position." (line 446-447, 449-450). The centre of the dropsonde circle was the centre of the circle flown by HALO and thus slightly off the pressure minimum seen in ERA5. This can be seen in Figure 13b.

Line 248: southerly winds not obvious for the central region in fig. 3d…would suggest removing 'and central'.

> We agree that this was not as clearly visible in the central region for 07-09 March. Therefore, when we shortened this section, summarized the typical wind pattern of the warm period as: "This pressure constellation resulted in a consistent southerly and southwesterly flow with only a few short--lived interruptions in the three measurement regions. The interruptions can be seen as near--surface temperature drops and wind direction change (i.e., 11 March, Fig. 3c, d)." (line 195-197)

Lines 248-255: it's hard to visualize what you are saying just from fig. 3, would suggest adding in some spatial circulation figures like what you have within fig. 4.

> As this is only preconditioning the HALO-(AC)³ period, we did not include a spatial circulation figure. This part has been dropped to reduce the length of this section and to decrease the focus on the preconditioning.

Line 278: the stated maximum IVT_north of 388 kg/s/m doesn't seem consistent w Fig. 2. Is that because the maximum is an hourly-mean?

> In Figure 2, regional (or area) averages have been computed averaging over both latitudes and longitudes. However, this 388 kg m-1 s-1 value from Figure 5 is a latitude average only (thus, keeping longitude information).

Line 280-282: what's the difference between 'latitude-averaged' and 'area-averaged' IVT?

➢ Latitude-averaged is only a 1-dimensional averaging while area-averaged is 2-dimensional and respects the increasing data point density of a regular lat-lon grid with increasing latitudes.

Line 283: how are you defining MWAI intensity? Winds?

➢ We distinguish between weak and strong MWAIs through IVT_north thresholds (strong if exceeding central-region-average of 100 kg m-1 s-1). We rephrased the distinction between weak and strong MWAIs more clearly: "An MWAI is considered weak (strong) when IVT_north is below (equal or above) 100 kg m-1 s-1. " (line 133)

Line 294: I don't follow "The moisture flux decreased faster than the heat flux". Is this from the atmosphere to the ocean? Or the turbulent fluxes coming off of the ocean?

➢ To clarify that we mean the atmospheric heat and moisture fluxes, we rephrased this part to: "After the AR, much drier but still relatively warm air followed, leading to a strong reduction in IVT_north and a slight reduction in IHT_north (Fig. 5). " (line 216-217)

Line 302: "frontal structure representative of a Shapiro-Keyser cyclone". Better to just describe the frontal structure, as many readers, including myself, will not know what you are talking about.

➢ We understand that not every reader might be familiar with the term but during the campaign, this event was always called "Shapiro-Keyser" cyclone. In upcoming studies, it might be that this term would also be used as it represents a turning point in the campaign and would be lost in other cyclones if this classification was removed. We dropped the brief description of the Shapiro-Keyser cyclone characteristics (line 219-221) for brevity as this can be found in the literature.

Line 385: it could be interesting to discuss how the subsidence is evolving as well, as that would also influence the static stability.

➢ We analyzed ERA5 based vertical velocity anomalies at 850, 700 and 500 hPa over the cold period compared to the 1979-2022 climatology. We found a slight positive subsidence anomaly at 850, 700 and 500 hPa in the central Arctic, coinciding with the enhanced static stability in this region (cold anomaly at 2m, warm anomaly at 850 hPa). This information has been added in line 275-276: "This area also shows slightly positive subsidence anomalies at 850 hPa (not shown). "

Line 415: not fully following how surface conditions explain a high tropopause height. I think you can just say 'vertical advection lifts the tropopause to 12.9 km' and be done with it.

➢ The vertical extension of the troposphere also depends on the heat and moisture content (but, of course, not just at the surface). Therefore, we rephrased this part to: "Together with high temperatures and moisture load (IWV > 10 kg m-2, Fig. 7b), the

troposphere extended up to a tropopause height up to 12.9 km (measured by the 12 UTC radiosonde on 12 March, Fig. 7a)." (line 300-302)

Line 425: these radiosonde profiles are also an opportunity to assess the corresponding ERA5 profiles.

➢ We understand that a comparison between ERA5 and observations is beneficial for scientists using ERA5, but we would like to keep the focus of the manuscript on the weather (and sea ice condition) analysis and climatological context. Nevertheless, we added the ERA5 based IWV estimate close to Ny-Alesund for 24 March 2022: "Northerly winds corresponding to the MCAO period led to extremely dry conditions with IWV down to 1.1 kg m-2 (closest grid point in ERA5 with land--fraction < 0.25: 1.5 kg m-2) on 24 March at 06 UTC (Fig. 7b)." (line 309-311)

Line 450: '2023) that' => '2023), '

➢ Well spotted. We corrected it: "Although the liquid precipitation on snow alters the signal of the microwave radiometry and might have increased the uncertainty of the SIC product (Stroeve et al., 2022, Rückert et al., 2023), that SIC reduction was obvious in visual satellite images as well (e.g., NASA Worldview, not shown)." (line 330-333)

Line 527: it should be relatively straightforward to figure out if the latent heat fluxes are increases because q_sat-q_air is increasing or because the wind speeds are increasing. How much is the q_sat increasing? I would think the SST would not be changing all that much?

➢ We investigated the regional distribution of q_sat - q_air and found that the southern region indeed featured higher differences between q_sat and q_air (especially later on 01 April and on 02 April 2022). This supports the assumption that q_sat – q_air is rather responsible for enhanced latent heat fluxes than wind speeds. We added this information in section 4.3: "Larger differences between the specific humidity of the air and specific humidity at saturation in the southern compared to the central region support this assumption (not shown)." (line 401-402)

Line 565: 'lied' => 'lay'

➢ Has been corrected: "As the northwestern part of the circle lay over sea ice, the vertically averaged lower tropospheric lapse rate (C3) indicated... " (line 436-437)

Line 573: please include a figure comparing the drop sonde vorticity and wind speed profiles to that from ERA5. The drop sonde circle should also give you a divergence profile and updraft speed that can be compared to that from ERA5 in a figure. Please do so.

➢ We added ERA5 vorticity to the plot (see response to your comment on line 218 of the unrevised manuscript). Also with respect to the other reviewer, we do not show divergence profiles (and comparisons to ERA5 in this respect) to avoid adding more content to the manuscript.

Line 605: 'a' => 'an' (in front of easterly)

> Well spotted. The error has been corrected: "This layer is associated with higher wind speeds up to 25 m s-1 and a shift in wind direction from an easterly flow in the lower 3 km to a southerly flow in the upper troposphere." (line 476-477)

Line 653: insert 'the' before 'absence'

> Thank you for spotting also this error. It has been corrected: "This combination resulted in an isolated humidity layer including Arctic cirrus in the absence of low-- and mid--level clouds." (line 520)

Figures/Tables

Table 1: the northern part of the northern region is hard to understand initially from this table, however, figure 1 shows the study area very well. I had to look down to figure 1 to understand the table. It might be best to either have the figure before the table, or to add a 4th column to the table for the northern part of the northern region

> We agree that a fourth column helps to separate the extension from the main part of the northern region. The table has also been moved to the new Appendix A, which contains detailed methods, because the exact coordinates would only be necessary for full reproducibility.

Fig. 1: spell out what NYA means in the caption.

> We added the description of "NYA" in the caption: "The orange label NYA in the zoomed domain marks the location of Ny-Alesund."

Fig. 3: the graphic on the left looks a bit odd, as there is no real need for us to know the day of the week I don't think. I would suggest adding a color bar to the top of the right-hand panels that has the identification information.

> We agree that the calendar like graphic has some redundant information like the weekday. However, events can be more directly attributed to a certain date while it might be more difficult to read the exact date in a colorbar-like time series (i.e., as illustrated below in the same colours used in Fig. 3a). In case we did not understand your idea, we would kindly ask for an elaborated comment.

[Figure]

Fig. 4: include dates in caption.

- Dates are now included in the caption: "Maps of mean sea level pressure (white contour lines), 500 hPa geopotential height (black contour lines), and 850 hPa equivalent--potential temperature (shading and grey contours) from ERA5 data for representative days of the main weather conditions at 12 UTC. 13 March (a) and 15 March (b) represent the moist and warm air intrusions with AR characteristics, (c) shows the Shapiro--Keyser cyclone on 21 March that marked the beginning of the cold air outbreak period, 28 March (d) and 01 April (e) represent the persistent northeasterlies with varying cold air advection strength, and (f) features the Polar Low west of Svalbard on 08 April. ..."

Fig. 5: are these hourly values? Would be worth putting in caption.

- Yes, theses are hourly values. We added it to the caption: "Hovmöller diagram of hourly vertically integrated meridional fluxes of (a) heat (IHT_north), and (b) moisture (IVT_north) during HALO-(AC)³, averaged over the central region latitudes."

Fig. 8: some strange overlapping of lat/lon labels in a-c, would suggest just removing a few.

- We removed the overlapping lat and lot labels.

Fig. 9, caption: I don't understand the last sentence, and both the top and far right histograms need a basic description. Also, the values shown in here for

- We think that you referred to the horizontal lines that should mark the measurement region boundaries? We changed the colours to the respective colours used in Fig. 3.

Fig. 10: fewer lat/lon labels and bigger plots would be nice. You could just leave most of the lat labels out.

- We reduced the number of lat and lon labels and increased the subplot size.

Fig. 13: including the ERA5 vorticity values on panel a would be nice as would be an additional plot showing the wind speeds/divergences. Panel b needs SLP labels.

- We added the ERA5 vorticity to the plot but do not show wind speeds or divergence due to the length of the manuscript. Panel b now has mean sea level pressure labels.

---

## Author Comment (AC3)

**Reviewer #1 reply:**

We thank the reviewer for the detailed comments. They helped us to get the message of the manuscript in a more concise and focused way without removing necessary details. Below, we repeat the reviewer's comments in black and write our responses in blue. The line numbers in the line-by-line responses are valid for the revised manuscript.

This paper presents a detailed account of the weather and sea-ice conditions experienced during te HALO-AC3 campaign. I sympathize with writing such an account; it is very useful as a reference for future work, but at the same time in a scientific journal – rather than a data journal – it should have some science in it to motivate the publication in a scientific journal. This often becomes a compromise and the factor that often suffer is the length and the scientific narrative. That is also the case with this manuscript, which is much too long and unfocused; it is unclear if the paper is describing methods and measurements or – as is claimed in the title – the meteorological conditions during the campaign.
Therefore, I am recommending a major revision focusing on reducing the details on how the different analyzes were obtained, minimizing the repetition of unnecessary information and streamlining the language.

➢ We moved details from the methods section to the appendix because we think that having some details available might be helpful for full reproducibility of the study while keeping the main body more concise. We also reduced the length of chapter 3 keeping the main focus on the description of environmental conditions and the climatological context.

Major concerns
The most stressing concern is the length, the degree of detail and lack of focus. There are to much too many details that would be better suited in special papers dealing with the different aspects whether that be analysis methods or measurement details. Combined with the rather "flowery" language, where the same thing is not rarely and unnecessarily described in more than one wording makes the reading tiresome; I must confess I gave up reading around page 30 or so. It just has much to many details that are better suited in topic specific analysis papers.

➢ We agree that the manuscript was too unfocused and described too many small details. Therefore, we reduced the text length in the revised manuscript to bring the message across more efficiently. We keep a shorter version of the description of all relevant weather events as this manuscript aims to be a comprehensive overview of the HALO-(AC)³ weather conditions. Duplicates of descriptions have been removed.

The data and methods section is (5.5 pages) is much to detailed for this scope of this paper and should be shortened by 50%; I'm, sure just condensing the language could do at least half of that. It has an "everything but the kitchen sink" character. For the most important measurement asset – the HALO aircraft – only the dropsondes are discussed (lines 100-104) while the measurements at Ny-Ålesund are twice that long and not really needed; I'm sure these are described elsewhere and can be referenced. The fact that only sea-points where used in various analyses are repeated at least three times; once is enough. Definitions of ARs

and MCAOs is also much to detailed and the discussion of the circles flown to estimate vorticity is not nearly enough to really understand how but way too much given how this is used in the upcoming sections.

➢ Thank you for identifying sections where details can be removed. We reduced the length of the methods section and kept only the most important descriptions. We kept but reduced the Ny-Alesund measurement descriptions so that the reader gets a brief overview of the measurements used in the analysis. The Atmospheric River tracking algorithm has been replaced by references to literature.

The painstaking day-to-day-account of the synoptic development on page 11-24 (14 pages!) should be condensed to its main components and shortehed to 30% of the present length. The only section that should actually be longer is the comparison to climatology; this is very useful for papers to come. The Ny-Ålesund section is much to long; I think this paper does not really need it and it could be dropped all together.

➢ We agree that a description of the development day-by-day is too much for a scientific journal. We therefore condensed the weather development description to its core information. To avoid adding more length to the manuscript, we did not add information here. However, due to the reduction of the synoptic description, the balance between synoptic description and climatology has been improved. We did not fully remove the weather description at Ny-Alesund but reduced it to its essentials as the additional measurements form this research station might be included in future HALO-(AC)³-related studies.

The section on specific events is what saves this paper; still at 8 pages also this could probably also be shortened.

➢ We agree that the section is also too long and tried to shorten it.

Some detailed concerns:
Line 11: Mentioning "Shapiro-Keyser cyclone" in the abstract is complete overkill; I bet less than a third of all potential readers have any clue what this means for the results.
➢ We changed it to "a strong cyclone" (line 11 of the revised manuscript).

Line 14-15: Isn't it natural that conditions during *any* AR would be warmer than climatology?
➢ True, we rephrased it to: "due to the strong influence of the ARs " to set the focus more clearly on the effect of ARs (line 14).

Line 15: What is significant in the statement that the SIC was within the 10-90 percentiles; that covesr almost everyuthing, doesn't it?
➢ Here, we wanted to express, that SIC was not extremely low or high, but rather normal. We rephrased it to "the sea ice concentration (SIC) was well within the climatological variability, staying within the 10-90th percentiles over the campaign duration" (line 14-15).

Line 31: The connection between a slightly weaker jet stream and a more meandering flow is far from well established; suggest inserting "possibly" somewhere in this sentence.

➢ Agreed, "possibly" has been inserted (line 30).

Line 38: The statement about warm air gliding up on a cold dome is very popular in some circles, yet I would say it is false. If it were true, what happens to the air under the dome over time? I presume it can flow out of the Arctic during MCAOs, but apparently not be replenished by ARs? Wouldn't that be contrary to having a dome in the first place? Instead – as what the hole campaign was designed to study – warm air flowing into the Arctic is transformed to Arctic air by interactions with the surface.

➢ We rephrased it to avoid the confusion with the Arctic cold air dome: "When the warm air is pushed upwards over cold Arctic air masses, deep cloud... " (line 39-40)

Line 43: All ARs are not "extreme"; suggest using "large" instead.

➢ Agreed, "extreme" sends a wrong message here and is not appropriate. We replaced it by "strong" (line 46).

Line 56: This is a problem not only for climate models; moreover, the Pithan reference argues for the Lagrangian methods applied in HALO-AC#, but provides no evidence for how this is modeled – poorly or otherwise.

➢ Agreed, this problem is not restricted to climate models. We rephrased this sentence for clarification: "This cloud evolution is not well represented in models but an important feature (Pithan et al., 2018). " (line 57-58)

Line 62-63: The wording "does not permit" is too strong. A Lagragian method does not by itself ascertain proper observation of the transformation and multiple Eulerian observations along a trajectory may provide some transformation information. Its not black or white…

➢ Agreed. We changed it to: "To observe air mass transformation processes along their meridional pathway in a Eulerian view, multiple research stations that are exactly aligned with the wind direction would be needed." (line 64-65).

Table 1: With the figure, this table is not necessary.

➢ For full reproducibility, the exact coordinates might be helpful because i.e., the southern limit of the southern region (70.6 °N) cannot accurately be determined from the figure. We shifted this table to the new Appendix A.

Line 100-103: Why this degree of detail for the dropsondes? Not necessary in this paper.

➢ We agree that this can be erased and therefore removed details of the specification of measurement accuracies (line 96-98).

Line 119-126: Too much detail; surely there is a reference!

➢ In general, we agree. But, we think that at least the product names should be included in the main text. The description of the sensors has been removed from the main text (line 106-114).

Line 141: IWV is not really a "basic variable".

➢ To reduce the length of the manuscript, we removed this subsection (former 2.4.1) as the data processing was minor and already written in section 3.1.

Line 142: Strictly speaking this means that all grid points where excluded, since ">0" means "larger than or equal to zero". So if land fraction is zero, implying ocean only, it would also be excluded.

➤ Well spotted typo. We meant ">0", instead of ">=0" and therefore changed it (line 128).

Line 144-145: The "north" subscript is confusing and probably unnecessary. The way this is calculated makes northward transport of excess heat or moisture by definition positive; southward negative. Including this subscript raises the question of you ignore southward fluxes.

➤ IVT_north (also IVT_v because of the meridional wind "v") is a common expression for meridional moisture transport in the northern hemisphere. We rewrote it to "Woods and Caballero (2016) detect moist air intrusions into the Arctic when the vertically integrated meridional moisture flux (IVT_north) at 70 °N exceeds ... " (line 133-134).

Line 155 & 161: Why different units?

➤ We directly used the IVT product provided in ERA5 data, which is in kg m-1 s-1, while Woods and Caballero (2016) used a similar but not identical product. When converting the Tg day-1 deg-1 to kg m-1 s-1, the threshold Woods and Caballero (2016) used is 60.6 kg m-1 s-1. We added this information: "... at 70 °N exceeds 200 Tg d-1 deg-1 (60.6 kg m-1 s-1) over a duration ... " (line 134).

Line 164-166: Don't understand; if the bar is too high for an event, then you raise the bar?

➤ For brevity, we removed the description of the AR detection algorithm and refer to literature instead.

Line 173: Excluding land points again.

➤ Has been removed.

Line 175: Why use temperature to indicate sea ice? There is sea ice in the model output.

➤ We kept the sea ice mask consistent with Dahlke et al. (2022). They decided for skin temperature instead of the ERA5 sea ice model output for convenience reasons as the skin temperature is used in the computation of the MCAO index. They investigated the differences between the skin temperature based and sea ice concentration based mask and found that differences were negligible.

Line 183: And excluding land points a third time.

➤ Has been removed.

Line 204-206: Unclear: First, the definition of the gradient is pretty obvious and doesn't have to be described. Second, the potential temperature can increase in the layer even if the average is zero, since the gradient is probably < 0 close to the surface or there wouldn't be any convection.

➤ We agree that there were too many details regarding the methodology. Therefore, the formula-based description of the vertical potential temperature gradient has been removed. We considered vertical mean, max and min to get an idea of the range of the vertical potential temperature gradient but showed the vertical mean

only for brevity. Indeed, the vertical minimum of the gradient is < 0 in a large fraction of the 200 km circle around the Polar Low's centre, indicating convection.

Line 212-213: The gustiness parameterization has nothing to do with the resolution; it is does to turbulence, which you need an LES to resolve.

> Here, we wanted to say that the ERA5 10 m mean wind might not capture the maximum wind speed of the Polar Low well because of ERA5's relatively coarse resolution. Therefore, we rather considered the gust to have a more realistic view on winds on sub-ERA5-grid scale. We added the following information and rephrased it to: "Wahl et al. (2017) [1] found that scales of multiples of the grid cell spacing are required to realistically represent the energy spectrum of a wind field. We decided to use the maximum 10 m wind gust instead of mean wind to get a better estimate of the near-surface wind field of this small-scale phenomenon that might be hidden due to the coarse resolution of ERA5." (line 163-166)
> [1]: Wahl, S., Bollmeyer, C., Crewell, S., Figura, C., Friederichs, P., Hense, A., Keller, J. D., and Ohlwein, C. (2017): A novel convective-scale regional reanalysis COSMO-REA2: Improving the representation of precipitation. *Meteorologische Zeitschrift* 26 (4), 345-361, doi: 10.1127/metz/2017/0824.

Table 2: With the text, this table is not necessary; alternatively use the table a do not repeat the details in the text.

> We cut the details in the text and moved the table to section 4.3, where we added another column indicating whether a condition is fulfilled or not. The presence of the table in section 4.3 might be convenient as the reader is reminded of the meanings of the acronyms C1-C6.

Line 218-224: Do we need this description? I can't see that vorticity is used in the description, and moreover, this description is not enough to really understand what you did but way too much for this paper.

> It is correct that the vorticity is not used in the synoptic description. However, in section 4.3 we compare dropsonde vorticity estimates to ERA5 model output (line 455-465). We reduced the details of the description in the manuscript and refer to literature (line 170-172).

Line 268-270: This sounds a bit too simple to be the whole truth, that the delay in surface warming is just because of the slope of the warm front; at least you show this is the case – or drop the argument.

> We dropped the argument.

Line 289-281: Drop "records"; this is not a championship.

> This part has also been rewritten for brevity and now reads as: "Simultaneously, the latitude-averaged IHT_north and IVT_north exceeded the previous maxima from 1996 (9.44 * $10^{10}$ W m-1 vs. 9.32 * $10^{10}$ W m-1, and 388 kg m-1 s-1 vs. 384 kg m-1 s-1, Fig. 5)" (line 216-217)

Line 283-284: Don't understand the caveat; ist it or isn't it and why?

> We wanted to mention that we only focussed on certain regions (boxes), but not the entire Arctic. Other regions may have experienced stronger MWAIs in that time

period but we didn't have a look at those other regions.  We dropped the caveat as it does not provide information relevant for the key message of the manuscript.

Line 295: I suggest "indicating" instead of "illustrating", since you don't show this.

➢ Agreed. We rephrased it to: "After the AR, much drier but still relatively warm air followed, leading to a strong reduction in IVT_north and a slight reduction in IHT_north (Fig. 5). " (line 222-223)

Line 300-307: Why bring in the Shapiro-Keyser classification? Is it relevant an if so, how is it relevant? I bet a majority of readers doesn't even know what this is.

➢ We understand that not every reader might be familiar with the term but during the campaign, this event was always called "Shapiro-Keyser" cyclone. In upcoming studies, it might be that this term would also be used as it represents a turning point in the campaign and would be lost in other cyclones if this classification was removed. We dropped the brief description of the Shapiro-Keyser cyclone characteristics (line 225-227) for brevity as this can be found in the literature.

Line 306-307: Don' t understand; if the heat content is low, why is the meridional heat transport not negative?

➢ For brevity, this sentence has been removed as it only provided details of minor importance. Heat transport was negative but not strongly negative. Large negative values require large amounts of heat being transported southwards. If no (or extremely small amounts of) heat is transported southwards (for example during cold temperatures), we'll have only slightly negative values.

Line 326: Suggest "dissipating" rather than "being filled up".

➢ Agreed. Has been changed to: "As the Shapiro-Keyser cyclone stayed over the Barents Sea while dissipating, IWV dropped..." (line 246)

Figure 5: It strikes me that Figure 5 is underutilized; drop it or use it more. Why the change in tilt on 21 March?

➢ We understand that this figure might appear underutilized. For brevity, the heat and moisture fluxes in this figure are only discussed around the main synoptic events. However, we would like to keep this figure as it shows the longitudinal position of the meridional air mass transports and the record breaking IVT_north and IHT_north values.
The tilt of the contour lines depends on the wind regime: For totally meridional winds, the contour lines of certain features (WAI) would be vertical (because of missing zonal propagation).

---

## Author Comment (AC4)

**Reviewer #2 reply:**

We thank the reviewer for the supportive review of the manuscript. We appreciate the detailed comments about the usage of ERA5 and its potential performance issues in the Arctic. We revised the manuscript including a more thorough literature research regarding ERA5 performance and adding minor observation-reanalysis comparisons. Below, we repeat the reviewer's comments in black and write our response in blue. The line numbers in the responses are valid for the revised manuscript.

This is a generally well-written presentation on the meteorology experienced during the HALO-AC3. I enjoyed reading it and am confident it will be a useful contribution to the larger research community. I like many of the figures. My main comment is that I rather wished for more description of how well the ERA5 reanalysis can be trusted for this part of the world. The analysis relies heavily on ERA5, including for cloud and precipitation phase. Has there been any comparison of the ERA5 products to the in-situ data yet? It may still be early stages for this, but some of the drop sonde quantities provided in the manuscript would be very easy to compare against ERA5. A literature review of other assessments is mostly absent, other than some in section 2.4.4. My own cursory web search revealed at least these two: Seethala et al., 2021; Loeb et al., 2022, but I would expect there to be more. I was hoping to see a more systematic assessment of the ERA5 quantities.

> We performed a more dedicated literature research regarding the performance of ERA5 in the Arctic (line 122-126). For the HALO-(AC)³ campaign, ERA5 data can probably be more trusted than in other years because our dropsondes have been assimilated. We also added this information to the manuscript: "During HALO-(AC)³, dropsonde measurements launched from HALO were assimilated into ERA5." (line 126-127). We now include a comparison of IVT of the Atmospheric Rivers from ERA5 to dropsondes (line 377-379 and Appendix B), and IWV from Ny-Alesund radiosondes to values from ERA5 (line 316-317). For brevity and to focus on the analysis of the weather events and their climatological context, we keep the comparison of ERA5 with observations to a minimum. More detailed comparisons are expected to be part of upcoming publications.
> The precipitation phase is a known problem even with observations due to deficits in model microphysics. We observed liquid precipitation over the sea ice with HALO's cloud radar during the Atmospheric River events. We added this information to the precipitation paragraph of the warm air intrusions and Atmospheric Rivers chapter (4.1) to ensure the reader that ERA5 can be trusted in this case: "Liquid precipitation over sea ice was also observed by the cloud radar onboard HALO during research flights. " (line 389-390)

Was the data assimilation consistent for the entire timespan of the ERA5 climatology, so that statements about 'maximum records' (e.g. line 281) are fair to make?

> ERA5 assimilates various sources of information and the used number of observational data has increased steadily throughout time. Therefore, the quality of reanalysis in general has increased over time especially as more satellite data are assimilated. With the percentile information in the manuscript, we assess if a value is close or far from a new maximum or minimum.

The other main comment is that in several places there are references to place names whose geography the reader may not be aware of, such as Franz Josef land. Fig. 1 might be a place to add some helpful geographic annotations, also for the ocean basins (Fram Strait, Greenland Sea, Barents Sea), as is section 2.1

> We agree that the reader can much better follow our analysis when we point out the major land marks in Figure 1. Therefore, we added the ocean basin labels. The location of Franz Josef Land relative to Svalbard is now described in the manuscript ("...over Scandinavia and around Franz Josef Land (northeast of Svalbard, Fig. 6b).", line 232-233) but did not fit in Figure 1 without overloading it.

Specific comments:

Abstract: The acronym ERA5 is spelled out, but that for HALO-AC3 is not. My guess is that more readers will know what ERA5 is, than HALO-AC3. I'd suggest spelling out HALO-AC3 and seeing if the journal will accept ERA5 as is.

> Agreed. We now briefly explain the acronyms HALO and (AC)3 to clarify the origin of the campaign name: "Centered around the High Altitude and Long Range (HALO) research aircraft and the collaborative research project on Arctic Amplification (AC)3, the airborne field campaign HALO-(AC)3 took place from 07 March to 12 April 2022. " (line 2-3)

Abstract, line 10: include years of the ERA5 climatology.

> The years of the ERA5 climatology is embedded in the abstract as follows: "Compared to the ERA5 climatology (1979-2022), record breaking vertically integrated poleward heat and moisture fluxes ..." (line 8-9)

Abstract, line 11: not a good idea to expect the reader to know what a 'shapiro-keyser' cyclone is, you can leave out the name reference

> We agree that the detailed characteristics of this cyclone are not needed here and thus replaced it by "strong cyclone" (line 11).

Abstract, line 16: 'untypically' => 'atypically'

> Done.

Intro, line 32: my recollection is that the Francis and Vavrus, 2015, was highly debated after it was published, leading to a US CLIVAR report, and spurring other work by e.g. E. Barnes at CSU. A bit more detailed literature review here would make this portion more impactful.

➢ We extended the literature review regarding the impact of climate change on the jet stream: "A more meandering jet would result in an increasing number of poleward moist and warm air intrusions (MWAIs) and southward cold air outbreaks (CAOs). However, the tropical upper troposphere warms while the Arctic lower stratosphere cools, reducing meridional temperature gradients at higher altitudes (Lee et al., 2019, Stendel et al., 2021). The frequency of meridional transport through the North Atlantic has increased during the last decades while it stayed constant or even decreased in other regions (Mewes and Jacobi, 2019). You et al. (2022) found a positive trend in the frequency and duration of atmospheric blocking over the Barents Sea especially in winter, supporting the statement of an enhanced North Atlantic pathway for meridional transport. " (line 31-37)

Line 67: space between performance of

➢ Well spotted. Space has been added.

Line 120: Nimbus -> Nimbus

➢ Thank you for spotting also this typo, but we removed the description of satellite sensors for brevity.

Lines 161-164: this is slightly confusing as written. Do Guan and Wailer use a IVT threshold of 100 kg/m/s and you use 50? Maybe combine those two phrases into one sentence if so.

➢ Indeed, for the Arctic, we use the threshold 50 kg m-1 s-1 while it is 100 kg m-1 s-1 in the original Guan and Waliser revised Atmospheric River detection algorithm. We rephrased the second sentence to: "In this study, ARs were identified with a global algorithm by Guan and Waliser (2015) in its revised version (Guan et al., 2018), adapted to the lower moisture content of the Arctic (Lauer et al., 2023)." (line 137-139)

Line 212: how is the polar low's center determined.

➢ Thank you for pointing out that we missed giving this information. We added "pressure minimum" in paranthesis to clarify this: "We analyse the environment for Polar Low formation with a set of conditions (C1-C6) suggested by Radovan et al. (2019) and Terpstra et al. (2016) in a 200 km radius around the Polar Low's centre (pressure minimum):" (line 157-158)

Line 212: using the max 10 m wind gust as opposed to the mean wind assumes ERA5 underestimates polar low wind gusts…do you know this for sure?

> We did not intend this impression and realized that this could be formulated more clearly. We did not use mean wind because this quantity is expected to yield lower wind speeds than actually present at some places in the Polar Low due to ERA5's coarse resolution. Using either a higher resolution reanalysis (like CARRA) or using wind gusts instead capture the mean wind speed in a rather small scale feature like a Polar Low better. We added the information that "Wahl et al. (2017) [1] found that scales of multiples of the grid cell spacing are required to realistically represent the energy spectrum of a wind field. We decided to use the maximum 10 m wind gust instead of mean wind to get a better estimate of the near-surface wind field of this small-scale phenomenon that might be hidden due to the coarse resolution of ERA5." (line 163-166).
> [1]: Wahl, S., Bollmeyer, C., Crewell, S., Figura, C., Friederichs, P., Hense, A., Keller, J. D., and Ohlwein, C. (2017): A novel convective-scale regional reanalysis COSMO-REA2: Improving the representation of precipitation. *Meteorologische Zeitschrift* 26 (4), 345-361, doi: 10.1127/metz/2017/0824.

Line 218: the drop sonde vorticity calculation: at what time? What was the center of the drop sonde circle? The vorticity calculation should be easy to compare to that from ERA5, how does ERA5 do?

> The dropsondes have been launched between 06:55 and 07:53 UTC on 08 April. This information has been added to chapter 4.3, where also other time information regarding the Polar Low is given: "Dropsonde measurements between 06:55 and 07:53 UTC show high values of relative vorticity in the lowest 2 km and above 6 km, indicating cyclonic rotation." (line 455-456). We added a brief comparison of the ERA5 vorticity to the dropsonde vorticity: "When averaging ERA5 vorticity over the grid points closest to the dropsonde positions, we find a disagreement to the dropsonde measurements below and good agreement above 4 km height. (...)The disagreement between ERA5 and the dropsondes could be due to a misrepresentation of the Polar Low's wind field in the reanalysis or due to spatio--temporal mismatches of its position." (line 456-457, 459-460). The centre of the dropsonde circle was the centre of the circle flown by HALO and thus slightly off the pressure minimum seen in ERA5. This can be seen in Figure 13b.

Line 248: southerly winds not obvious for the central region in fig. 3d…would suggest removing 'and central'.

> We agree that this was not as clearly visible in the central region for 07-09 March. Therefore, when we shortened this section, we summarized the typical wind pattern of the warm period as: "This pressure constellation resulted in a consistent southerly and southwesterly flow with only a few short--lived interruptions in the three measurement regions. The interruptions can be seen as near--surface temperature drops and wind direction change (i.e., 11 March, Fig. 3c, d)." (line 201-203)

Lines 248-255: it's hard to visualize what you are saying just from fig. 3, would suggest adding in some spatial circulation figures like what you have within fig. 4.

➤ As this is only preconditioning the HALO-(AC)³ period, we did not include a spatial circulation figure. This part has been dropped to reduce the length of this section as suggested by reviewer 1 and to decrease the focus on the preconditioning.

Line 278: the stated maximum IVT_north of 388 kg/s/m doesn't seem consistent w Fig. 2. Is that because the maximum is an hourly-mean?

➤ In Figure 2, regional (or area) averages have been computed averaging over both latitudes and longitudes. To make it more clear that latitude averages (not area averages) are meant, we rephrased this part to: "Simultaneously, the latitude--averaged IHT_north and IVT_north exceeded the previous maxima from 1996 ($9.44*10^{10}$ W m-1 vs. $9.32*10^{10}$ W m-1, and 388 kg m-1 s-1 vs. 384 kg m-1 s-1, Fig. 5)." (line 216-217)

Line 280-282: what's the difference between 'latitude-averaged' and 'area-averaged' IVT?

➤ Latitude-averaged is only a 1-dimensional averaging while area-averaged is 2-dimensional and respects the increasing data point density of a regular lat-lon grid with increasing latitudes. To stress the latitude averaging, we rephrased this part to: "In Fig. 5, we show latitude--averages of IVT_north and the vertically integrated meridional heat flux IHT_north over the central region to .... " (line 189-190)

Line 283: how are you defining MWAI intensity? Winds?

➤ We distinguish between weak and strong MWAIs through IVT_north thresholds (strong if exceeding central-region-average of 100 kg m-1 s-1). We rephrased the distinction between weak and strong MWAIs more clearly: "An MWAI is considered weak (strong) when IVT_north is below (equal or above) 100 kg m-1 s-1. " (line 136)

Line 294: I don't follow "The moisture flux decreased faster than the heat flux". Is this from the atmosphere to the ocean? Or the turbulent fluxes coming off of the ocean?

➤ To clarify that we mean the atmospheric heat and moisture fluxes, we rephrased this part to: "After the AR, much drier but still relatively warm air followed, leading to a strong reduction in IVT_north and a slight reduction in IHT_north (Fig. 5). " (line 222-223)

Line 302: "frontal structure representative of a Shapiro-Keyser cyclone". Better to just describe the frontal structure, as many readers, including myself, will not know what you are talking about.

➤ We understand that not every reader might be familiar with the term but during the campaign, this event was always called "Shapiro-Keyser" cyclone. In upcoming studies, this term might also be used as it represents a turning point in the campaign

and would be lost in other cyclones if this classification was removed. We dropped the brief description of the Shapiro-Keyser cyclone characteristics for brevity and only refer to literature.

Line 385: it could be interesting to discuss how the subsidence is evolving as well, as that would also influence the static stability.

> We analyzed ERA5 based vertical velocity anomalies at 850, 700 and 500 hPa over the cold period compared to the 1979-2022 climatology. We found a slight positive subsidence anomaly at 850, 700 and 500 hPa in the central Arctic, coinciding with the enhanced static stability in this region (cold anomaly at 2m, warm anomaly at 850 hPa). This information has been added in line 281-282: "This area also shows slightly positive subsidence anomalies at 850 hPa (not shown). "

Line 415: not fully following how surface conditions explain a high tropopause height. I think you can just say 'vertical advection lifts the tropopause to 12.9 km' and be done with it.

> Agreed, we rephrased this part to: "Vertical advection of heat and moisture lifted the tropopause to 12.9 km (measured by the 12 UTC radiosonde on 12 March, Fig. 7a)." (line 307-308)

Line 425: these radiosonde profiles are also an opportunity to assess the corresponding ERA5 profiles.

> We understand that a comparison between ERA5 and observations is beneficial for scientists using ERA5, but we would like to keep the focus of the manuscript on the weather (and sea ice condition) analysis and climatological context. Additionally, we currently do not know which sondes were assimilated and which not, which makes a fair comparison difficult. Additionally, the orography around Ny-Alesund is very complex, adding to the difficulties for the ERA5-radiosonde comparison. Nevertheless, we added the ERA5 based IWV estimate close to Ny-Alesund for 24 March 2022: "Northerly winds corresponding to the MCAO period led to extremely dry conditions with IWV down to 1.1 kg m-2 (closest grid point in ERA5 with land--fraction < 0.25: 1.5 kg m-2) on 24 March at 06 UTC (Fig. 7b)." (line 316-318)

Line 450: '2023) that' => '2023), '

> Well spotted. We corrected it

Line 527: it should be relatively straightforward to figure out if the latent heat fluxes are increases because q_sat-q_air is increasing or because the wind speeds are increasing. How much is the q_sat increasing? I would think the SST would not be changing all that much?

> We investigated the regional distribution of q_sat - q_air and found that the southern region indeed featured higher differences between q_sat and q_air (especially later on 01 April and on 02 April 2022). As the sea surface temperature over the southern

region is higher (mostly by 3-4 K) than over the central region, q_sat is increased in the south. This shows that q_sat – q_air is responsible for enhanced latent heat fluxes. We added this information in section 4.2: "We found that larger differences between the specific humidity of the air and specific humidity at saturation in the southern compared to the central region were responsible for the increased latent heat fluxes (not shown)." (line 411-412)

Line 565: 'lied' => 'lay'

➢ Has been corrected.

Line 573: please include a figure comparing the drop sonde vorticity and wind speed profiles to that from ERA5. The drop sonde circle should also give you a divergence profile and updraft speed that can be compared to that from ERA5 in a figure. Please do so.

➢ We added ERA5 vorticity to the plot (see response to your comment on line 218 of the unrevised manuscript). Also with respect to the other reviewer, we do not show divergence profiles (and comparisons to ERA5 in this respect) to avoid adding more content to the manuscript. Furthermore, a detailed publication on the divergence measurements from the research flights is underway.

Line 605: 'a' => 'an' (in front of easterly)

➢ Has been corrected.

Line 653: insert 'the' before 'absence'

➢ Has been corrected.

Figures/Tables

Table 1: the northern part of the northern region is hard to understand initially from this table, however, figure 1 shows the study area very well. I had to look down to figure 1 to understand the table. It might be best to either have the figure before the table, or to add a 4th column to the table for the northern part of the northern region

➢ We agree that a fourth column helps to separate the extension from the main part of the northern region. The table has also been moved to the new Appendix A, which contains detailed methods, because the exact coordinates would only be necessary for full reproducibility.

Fig. 1: spell out what NYA means in the caption.

➢ We added the description of "NYA" in the caption: "The orange label NYA in the zoomed domain marks the location of Ny-Alesund."

Fig. 3: the graphic on the left looks a bit odd, as there is no real need for us to know the day of the week I don't think. I would suggest adding a color bar to the top of the right-hand panels that has the identification information.

➢ We agree that the calendar like graphic has some redundant information like the weekday. However, events can be more directly attributed to a certain date while it might be more difficult to read the exact date in a colorbar-like time series (i.e., as illustrated below in the same colours used in Fig. 3a). In case we did not understand your idea, we would kindly ask for an elaborated comment.

[Figure]

Fig. 4: include dates in caption.

➢ Done.

Fig. 5: are these hourly values? Would be worth putting in caption.

➢ Yes, theses are hourly values. We added it to the caption: "Hovmöller diagram of hourly vertically integrated meridional fluxes of (a) heat (IHT_north), and (b) moisture (IVT_north) during HALO-(AC)³, averaged over the central region latitudes."

Fig. 8: some strange overlapping of lat/lon labels in a-c, would suggest just removing a few.

➢ We removed the overlapping lat and lot labels.

Fig. 9, caption: I don't understand the last sentence, and both the top and far right histograms need a basic description. Also, the values shown in here for

➢ We think that you referred to the horizontal lines that should mark the measurement region boundaries? We changed the colours to the respective colours used in Fig. 3.

Fig. 10: fewer lat/lon labels and bigger plots would be nice. You could just leave most of the lat labels out.

➢ We reduced the number of lat and lon labels and increased the subplot size.

Fig. 13: including the ERA5 vorticity values on panel a would be nice as would be an additional plot showing the wind speeds/divergences. Panel b needs SLP labels.

➢ We added the ERA5 vorticity to the plot but do not show wind speeds or divergence due to the length of the manuscript. Panel b now has mean sea level pressure labels. We also corrected an error regarding the dropsonde vorticity uncertainty computation.

---

## Referee Report (RR1)

The revision has improved the manuscript, the writing flows better and the efforts to reduce the length has also improved it. That said it is clearly not written by someone whose first language is English, and it would benefit from edits by one of the more experienced writers on the long authorship list. The introduction is improved but could still use more polish. I also strongly object to the continued use of the term 'Shapiro-Keyser' without including clear language on what distinguishes a 'Shapiro-Keyser' cyclone from a regular cyclone. This work will rightly appeal to more than just the AC3 research cohort and should be written for a larger audience. Some readers may think differently about cyclones.

Abstract:
Line 16: what is 'aged subpolar warm air' - in particular what does the 'aged' refer to? I suspect the adjective can be removed without loss.

Line 19: agree w reviewer 1 that 'staying within the 10-90th percentiles' can be removed, you already say 'within the climatological variability' , that's enough.

Line 27: 'temporal shifts up to one day' - not sure what this means. Temporal shifts in what?

'Potential future studies' (line 22) and 'future analyses' (line 31) is redundant. Would suggest removing the first mention although I personally think the last sentence can be removed. It is what we call in the US a 'motherhood and apple pie' statement - something so general, bland and well-accepted that it doesn't need mention.

Introduction:
Line 40: 'the tendency' -> 'the jet stream's tendency'
Line 45: "the frequency of meridional transport": what does this mean? Transport of what? Frequency measured how? At what altitude ?
Line 50: 'meridional transport' of what? Probably also need to relate transport to a budget, e.g. if the meridional temperature gradient decreases while the tropics warm up, the meridional transport of temperature could increase but still have less impact on the high-latitude temperature budget than when the gradient was stronger.

I am not an expert on the larger-scale perspective either, and a larger lack of confidence is helping to drive Arctic research. I would suggest attempting to communicate that 'questions remain' as opposed to the current more assertive comments, e.g, 'xxx hypothesize that weaker, meandering jet streams will result in….' Then end the paragraph with something on 'most research has relied on global models unable to resolve mesoscale features well, leaving open questions on….'

Perhaps the more senior authors can step up here, to help polish the introduction a bit further? The first paragraph on p.4 is now confusing because the authors communicate instead a focus on MCAOs as opposed to the ARs of the previous paragraph. I think the paragraph may just need an additional connecting sentence to make the segue.

p. 9 top of page: please recognize somewhere that comparing a recent time period from ERA5 will assimilated soundings to earlier years with less data assimilation introduces a form of bias.

p. 13 line 323: please remove the presumptive description 'Omega block' here . Such terms get thrown around colloquially in weather discussions but unless they are a major focus of the writing they should not be used. Just describe what you have to say.

p. 17, top of page: sorry, not okay to just throw out the term 'Shapiro-Keyser' without describing what it is. Again it may be something that was tossed around within the AC3

weather discussions by a small group of people, but the audience for this manuscript can be anticipated to be larger, some of whom are likely to think about cyclones differently from the AC3 weather forecasting team. Currently what this writing communicates is an author list hiding behind a shorthand they can't explain, which then suggests that perhaps they also don't understand. Give this another try and also try not to rely on this term quite so much, it still appears in many places.

Line 423: what makes air 'aged' ? I find this an odd term. Does it matter?

p. 20-21, lines 445-450: again this reliance on a short-hand - Polar Low - that just comes across as slang. How is this mesoscale cycle a 'Polar Low' as opposed to simply a mesoscale cyclone? Do circulation features need to meet a quantitative criteria? This is all discussed in more depth in section 4.3, I would suggest just calling it a mesoscale cyclone at this stage in the manuscript.

p. 24 line 526: again this strange habit of calling air 'aged'. What the heck does this mean? Aerosol can be 'aged' but what distinguishes 'aged' air from 'fresh' air?

p. 28 line 609: nice to see some comparison between ERA5 and the dropsondes, showing that even with the data assimilation ERA5 isn't getting the full moisture flux. This is worth mentioning I feel.

Section 4.3: so fulfilling 5 of the 7 (or 4 out of 6) conditions put forth by Radowan qualifies a cyclone as a Polar Low? I would suggest stating that explicitly if so. As written the authors are appearing to presume the cyclone is a polar low and then just characterize it using Radowan's criteria. The discussion between lines 709-715 is nice and I'm fine with the system being called a Polar Low but would suggest rewriting the language so that it is not so initially presumptive.

---

## Author Response (AR2)

**University of Cologne**

[Figure]

**Institute for Geophysics**

**and Meteorology**

**MSc. Andreas Walbröl**

Phone: +49 (0)221 470-3678

E-mail: a.walbroel@uni-koeln.de

Pohligstr. 3

50969 Cologne

Germany

Cologne, 11 April 2024

**Author's response:**

Dear ACP editorial team,

thank you for extending the deadline for the submission of the revised manuscript "Environmental conditions in the North Atlantic sector of the Arctic during the HALO–(AC)³ campaign". According to the comments of the reviewers (see line by line response below) the manuscript has been revised thoroughly. Its focus has been refined and now emphasizes on how the meteorological conditions of the HALO-(AC)³ campaign time period relate to the long-term climatology. Because of the shift of the focus of the study, we have changed the title to "Unusual atmospheric and sea ice conditions in the North Atlantic sector of the Arctic during the HALO-(AC)³ campaign".

To address the main concerns of the reviewers the revised manuscript

- has been shortened significantly with a more concise description of the methods, and synoptic sections,
- now focusses on the main meridional atmospheric transport processes (moist and warm air intrusions, and cold air outbreaks),
- has an improved structure, which better separates the synoptic description from the climatological comparison, and
- includes a new analysis to better relate the marine cold air outbreaks to the long-term climatology

Neither this manuscript or substantial parts of it have been published elsewhere in English or any other language, nor is it presently under consideration for publication by any other journal.

Sincerely,

Andreas Walbröl

**Reviewer #1 reply:**

We thank the reviewer for the detailed comments. They helped us to get the message of the manuscript in a more concise and focused way without removing necessary details. We also improved the structure to avoid repetitions and unclear definitions. Below, we repeat the reviewer's comments in black and write our responses in blue. The line numbers in the line-by-line responses are valid for the revised manuscript.

*This is a revised version of a manuscript that is expressedly intended to describe the meteorological conditions during the HALO-AC3 campaign. It started out as a very unfocused manuscript, where now some parts have been transferred to a series of Appendices. What remains, however, is still problematic and cannot be published in this form. I've been struggling with what to recommend this time, and as the "unfocusedness" from the previous version was not really reduced, it tells me the author needs more incentive to think anew; hence I now recommend a rejection hoping that the material behind the manuscript can be reshaped into something coherent.*

➢ We have taken the point of the reviewer very seriously. The manuscript has been revised thoroughly to address a single focus, i.e., the question of how representative the conditions during the HALO-(AC)³ campaign were compared to the long-term climatology. By analyzing the meridional heat transport by Atmospheric Rivers and marine cold air outbreak events in the Fram Strait in the context of climatology, we are able to conclude that the conditions were suitable to pursue the campaign's research goals. The description of the conditions is now only a short but necessary part of understanding the setting of the atmosphere. The manuscript, significantly reduced in size, should be now be much more coherent.

*Major concerns: I think the main question here is "why". Why write this paper and why with this content? Like I mentioned in my earlier review I do sympathize writing a summary paper from a field campaign; I've been there myself. It is very useful to have one as a reference for all future papers from the campaign! But it will still have to stand by itself as a science contribution. A campaign description – which this is not – should hold something about strategy and thinking; the "how" and "why" and not just the "what". A description of what happened is even more difficult; the science has not yet been done and just an analysis of the meteorology during a time period, just because there happened to be a field campaign during that time, is not really a good enough reason to write a science paper about. Unless some real science is actually revealed in the analysis, which is unfortunately not the case here. So while I can see the usefulness for future reference, I do not see enough science here to warrant publication in a science journal. As a project report, yes; as a paper no. Moreover, most of the data analyzed – sometimes even cleverly analyzed – is ERA5-data and not observations at all. So I struggle with why! Why should this paper be published, except for its values as a useful future reference?*

➢ Thank you for the detailed description of your concerns. As outlined above, we strengthened the focus of the study. As also reflected in the new title, our detailed analysis clearly reveals the extraordinary meridional atmospheric transports (Atmospheric Rivers and marine cold air outbreaks) and their impact on the sea ice concentration in the Fram Strait. Without knowing how the specific conditions of the

campaign relate to the long-term climatology, the interpretation of the airborne observations might become misleading.

*In addition to this, I still find the paper poorly organized and confusing to read. Section 4 is mostly a repetition of Section 3, with the addition of the Polar Low and cirrus analysis, although both are mentioned also in section 3, and are both mostly superficial in Section 4; both sections are not needed and represents somewhat different writing structures. Above all, there is a lack of a narrative and reason; there is no connection between the developing meteorology and when the aircrafts were deployed – and how they were deployed.*

➢ We agree that the content of Sections 3 and 4 were similar in some parts. The new manuscript was reorganized and significantly shortened, especially Section 2 (data and methods). Sections 3 and 4 were merged and the "side stories" on Polar Low and Arctic Cirrus were dropped.

*The text is often confusing and different definitions are used in the different parts. For example, in Section 4, at the start on line 357, the dates 12-13, 14, 15 and 20 March are listed as having atmospheric rivers (ARs). But in Figure 3a the 14th is not classified at all and the 20th is classified as a cold-air outbreak (CAO). Looking at Figure 2 and the definitions used for an AR (lines 134-136, 60.5 kg m-1 s-1), there is one AR starting, probably, on the 12th continuing until the 17th with two "strong" peaks (line 136, > 100 kg m-1 s-1). The adopted definition moreover requires a duration of 1.5 days, which the 14, 15 and 20 March episodes would not fulfil being only one day each, and a width of 9deg which is not commented at all.*

➢ There was a misunderstanding of the definitions as different criteria are used for Atmospheric Rivers (AR) and warm air intrusions. The 60.5 kg m-1 s-1 threshold and the minimum duration requirement were not used to detect ARs, but Woods and Caballero (2016) used them to detect moist and warm air intrusions. As stated in Section 2, we used the Guan and Waliser detection algorithm for ARs and define a moist and warm air intrusion by an area average >= 0 over the central domain.

➢ We improved the description of the detection algorithms for the different event types to avoid ambiguity. The different dates regarding the events were due to different regions considered in the respective sections. To avoid this confusion, we only highlight the ARs that passed through the central domain (Fram Strait) in Sect. 3 and 4. Note that also different averaging times can lead to different values.

*There are simply too many things that I need to question. The definition of "aged air" is very fuzzy and seemed very "hand waiving" to me. Where does it come from, what ius aging and how do you know? The use fo circles flights to detect synoptic scale divergence needs a better evaluation before being used in a descriptive paper and here it doesn't really add any value. I suggest dropping it for now. The same goes for the sections on Ny Ålesund observations. Although these are the only atmospheric observations in this paper, it doesn't add any value to the content. The conclusion that there is a one day difference is too simplistic to be useful and the conclusion that one needs to be careful with easterly flow is trivial.*

➢ We dropped the rather exclusive synoptic terms of aged air, but also Shapiro-Keyser cyclone and Polar Low. Because we refined the focus of the study, we dropped the analysis of divergence observations as suggested by the reviewer.

➢ The analysis of Ny-Alesund measurements has been changed. We now use the data at the beginning of Section 3 to introduce the campaign weather conditions because this is the only radiosonde station that provides long-term atmospheric observations in our study area. Further on, the reanalysis data (ERA5) is used to expand the view to the entire study domain.

*Detailed concerns: Line 30: "… at most heights" is better.* Done.

*Lines 32-34: The argument that the upper tropical troposphere also has an amplified warming is intriguing and interesting; however, it never leads anywhere. Follow up or drop it.*

➢ We dropped the discussion about the jet stream response.

*Lines 38-43: This discussion reveals a troubling, however, unpronounced conceptual model. If the warm air was to glide up on top of the cold Arctic air (line 39), where does Arctic air come from? It can exit the Arctic in CAOs but southerly air cannot make it into the Arctic because it glides on top of the cold Arctic air. Moreover, to achieve the suggested downward long wave radiation effects, requires a low cloud base with a temperature close to that of the surface or higher and ascending air cools off. Of course this is not what happens! Southerly air masses are transformed to Arctic air masses by interactions with the new surface.*

➢ We agree that the discussion was misleading and would require a more detailed description of the radiative processes and surface influence on the moist and warm air intrusion. As this would be beyond the scope, we dropped the discussion.

*Line 57-58: The sentence "This cloud … , 2018)" seems to be incomplete.*

➢ We reformulated the sentence to: "This cloud evolution is difficult to capture by atmospheric models (Pithan et al., 2018) motivating dedicated measurement campaigns (Geerts et al., 2022; Lloyd et al., 2018)." (lines 51-53)

*Lines 65-68: Long and complicated sentence! I suggest cutting in two: "Therefore, … the Arctic." And then "This motivated … (AC)3".*

➢ Agreed. We split the sentence.

*Lines 74-75: Operating in a quasi-Lagrangian fashion does not necessarily require longer range, although long range is always good to have. The idea is to repeat visits to the same airmass at multiple missions. The long range comes in handy when following an airmass in over the pack ice, as staging is from far away in Kiruna.*

➢ Agreed. We now say that HALO "has an operating range of 9000 km in altitudes up to 15 km, which is beneficial for quasi-Lagrangian air mass observations" (lines 66-67).

*Line 81: Drop "basis".* Done

*Line 95: This heading just must be a misrepresentation of the HALO contribution to the observations of the atmosphere!*

➢ We changed the heading to "Atmospheric measurements" (line 87).

*Line 107: So which is it; a satellite pixel or a model grid cell?*

➢ We deleted the model as this section focuses on satellite measurements only. "Sea ice concentration (SIC), i.e., the percentage of a satellite pixel covered by sea ice..." (line 105)

*Line 152: Swap: "turbulent surface heat fluxes" is better.*  Thanks, done.

*Line 164: Confusing; what is referred to hear is the fact that a model cannot resolve energy at scales below several time its spatial resolution (theoretically always > 2; in practice often 5-10 times). The criteria applied here all come from the model resolved scale and including more grid points doesn't add anything.*

➢ We dropped the Polar Low discussion and therefore also the respective method section.

*Line 170-172: Suggest dropping this. It is not really a proven method and the circle used is of the same scale as a Polar Low and therefore this cannot be used to detect one. The results shown here is also less than clear.*  Done.

*Lines 173-180: No trajectories are used in the paper, so I suggest you drop this sub-Section.*

➢ Agreed. Trajectory calculations have been used in the background for our analysis but are no longer shown in the paper. Therefore, we dropped it.

*Section 3: This section moves on to the nitty-gritty much to fast. Discuss Figures 2-4 in more general terms so that the reader is familiarized with how these figures are builkt and can then use them for the more detailed description.*

➢ We removed many details, compressed the synoptic description, and changed the order of the figures to improve the flow and focus of the study.

*Lines 188-196: Mostly conentless rambling; use the space more efficiently to really motivate the division into the two main periods.*

➢ We dropped the discussion of the latitude averaged integrated heat and moisture fluxes and made the motivation to split the campaign into two periods more concise (lines 191-197).

*Line 198: Drop "corresponding"; not necessary.*  Thanks, done.

*Figure 2: Are the ticks at 00 or 12 UTC. It seems to this reviewer there is only one long AR according to the definition, with a dip in strength around the 15th.*

➢ Note that this figure shows daily averages while ARs are identified hourly. The x tick labels are at 00 UTC but the data is visualized at 12 UTC to represent the 'middle of the day'. The dip of the IVT$_{north}$ (daily mean and area-average) is therefore on 14 March 2022, which was partly between the two strong AR events (AR I and AR II). Our definition still classifies this day as a moist and warm air intrusion.

*Line 218-221: Everything here is from ERA5 so we don't really know that temperature "was" above freezing, in fact I doubt it, and the liquid precipitation was not "documented".*

➢ Liquid precipitation over the sea ice (up to about 80.5°N) was observed with HALO's cloud radar during research flights. Thus, the observations confirm the ERA5 precipitation phase in a certain region. This is now elaborated in the text: "Liquid precipitation at high latitudes over sea ice was also observed by the cloud radar onboard HALO as we detected a distinct bright band at about 0.5–1 km height during research flights. Thus, the observations confirm the presence of liquid precipitation at least in some regions over sea ice. " (lines 250-252)

*Line 224: I have a problem orienting myself here; I can't find that "low geopotential" which "persisted".* Has been reformulated.

*Line 226 and elsewhere: It is unclear to this reviewer if a "Shapiro-Keyser cyclone" is a real physical feature or just an alternative conceptual mode. In either case, drop this or explain it properly. What makes it an Shapiro-Keyser cyclone if indeed that is something different to any other extratropical cyclone or Polar Low.*

➢ For clarification: As the respective cyclone was a warm core cyclone and did not feature the typical occlusion front as a Norwegian cyclone, this could be classified as Shapiro-Keyser cyclone. However, we dropped the use of this term because it was not critical for the discussion and to reach a broader audience.

*Line 235: I see no evidence of how the "Arcytic inversion" looked like or was changed; this seems out of place.*

➢ We used the anomaly maps of 2 m and 850 hPa temperatures to get an idea of the lower tropospheric stability anomaly. However, we dropped the term 'inversion' here as we did not analyze the vertical temperature structure in detail.

*Lines 248-251: These seems to be very large changes in the statistics for a change in M from 9 to 10?*

➢ Note that the M values in the figure refer to daily averages over the central domain and thus only few cases with a spatio-temporal average of M larger than 11 K occurred in the ERA5 record.

*Line 254: Unclear what is meant by "sub-polar" here; what air mass is this, from where does it come and what is meant by aging? And how do you know?* Has been removed.

*Line 259: Confusing; isn't this esterly flow? Then downstream would be western Fran Strait which is ice covered and would be on-ice flow.*

➢ For clarification: It was a northeasterly flow at this time and the ice edge in the western Fram Strait was also roughly southwest-northeast. In a small region, the wind flow was actually off-ice. To avoid confusion, we dropped this.

*Line 260: Is "attribute" the right word here? This means finding the reason for; maybe use "classification"?*     Sentence has been dropped.

*Line 262: What do you mean by "another weak"; the previously discussed onbes were not weak?*

➢ For conciseness, the MCAO event description has been condensed.

*Lines 266-267: How can enhanced IWV predominantly due to water vapor close to the surface, where it is warm, be advected by "upper-level flow"?*

➢ For clarification: The enhanced IWV (above 10 kg m-2) was due to a weak moist air intrusion over parts the southern domain. Then, in the North Atlantic between Norway and Greenland, a strong vertical wind shear was present where upper level winds transported the moist air northwards into the western Fram Strait. However, we dropped the description of this weak moist air intrusion event.

*Page 15, para 1: Thois whole paragraph preceeds the climatology discussion that begins in the next sub-section.*     We improved the structure.

*Line 289-290: I see no reason why there should be a balance here; in fact there should nopt be a balance since there has to be a climatological heat transport northward to compensate for the radiative net loss at the Arctic TOA.*     We dropped this argument.

*Line 294: Maybe I misunderstand but I don't see the lowest 500 hPa anomalies over Greenland?*

➢ We exchanged the 500 hPa geopotential height anomalies by mean sea level pressure anomalies because this is a more intuitive variable. However, the overall picture remains the same. During the warm period (11-20 March), strongly negative anomalies can be found over Greenland (Fig. 6 b). This has been reforumlated to: "The blocking is evident in Fig. 6b, showing the strong anomalies of more than 10 hPa in the MSLP field over the whole warm period with lower pressure over Greenland and the central Arctic, and higher pressure over Scandinavia. " (lines 201-202)

*Lines 298-299: Again, why should there be a balance and what relevance are these dates in other years?*     We removed this statement.

*Section 3.2 Drop this; doesn't add anything to the manuscript.*     Done.

*Line 251: What are "weather related filters" and "smearing uncertainty"?*

➢ We now explain both terms in the manuscript: "Note that uncertainties of derived SIC in the marginal ice zone are especially large as a result of tempral and spatial interpolation (smearing uncertainty), and due to so-called weather filters. Weather filters remove the atmospheric contribution from the satellite signal to remove false sea ice in open-water regions but run the risk of removing true sea ice as well, especially in the MIZ." (lines 313-316)

*Line 365-370: It is unclear if these results and the discussion refers to ARs that start at a given latitude and ends in the Arctic or if some of them in fact ends much earlier. Colder air more northward holds less water vapor and naturally must have lower IWT.*

➢ The definition of ARs is a tricky issue, especially due to the moisture decrease with temperature. Percentiles are used to circumvent this. Here, we do not want to get into a discussion of AR algorithm performance but only explain how to interpret the figure (decrease of IVT with latitude) and why higher latitudes feature lower IVT.

*Line 373: What do you mean by time steps?*

➢ For clarification, we reformulated this as: "AR I and AR II both represent strong cases in terms of mean IVT as they partly lie outside the 25th percentile in latitude-IVT space [...]." (lines 231-232)

*Line 375-377: Unclear sentence starting with "While…"; why "while"?*     Has been changed.

*Line 381-382: And hence their length becomes dependent on a somewhat arbitrary definition.*

➢ Correct, but this is how the Guan and Waliser algorithm works.

*Figure 10: Doesn't make any sense to me! The definition of the "deviation" by necessity makes it zero whenever the 2022 precipitation rate is zero. Presumably, the deviations can be substantial also when this is the case.*

➢ The figure has been simplified for clarity. Deviations are now absolute anomalies.

**Reviewer #2 reply:**

We thank the reviewer for the supportive review of the manuscript. We appreciate the concerns regarding specific synoptic terms that are not needed for the discussion. Dropping them led to a clearer focus of the study. Below, we repeat the reviewer's comments in black and write our responses in blue. The line numbers in the line-by-line responses are valid for the revised manuscript.

*The revision has improved the manuscript, the writing flows better and the efforts to reduce the length has also improved it. That said it is clearly not written by someone whose first language is English, and it would benefit from edits by one of the more experienced writers on the long authorship list. The introduction is improved but could still use more polish. I also strongly object to the continued use of the term 'Shapiro-Keyser' without including clear language on what distinguishes a 'Shapiro-Keyser' cyclone from a regular cyclone. This work will rightly appeal to more than just the AC3 research cohort and should be written for a larger audience. Some readers may think differently about cyclones.*

➢ The paper was thoroughly revised emphasizing on the climatological aspect. We dropped the term 'Shapiro-Keyser' cyclone for simplicity and also deleted the part about the Polar Low.

*Abstract:*
*Line 16: what is 'aged subpolar warm air' - in particular what does the 'aged' refer to? I suspect the adjective can be removed without loss.*

➢ For clarification: 'Aged' is a synoptic term that refers to the aging of an air mass when it propagates away from its origin (e.g., aged subpolar air mass refers to an air mass that originated from the subpolar regions but warmed or cooled when it was transported to lower or higher latitudes, respectively). However, we dropped this term to avoid confusion.

*Line 19: agree w reviewer 1 that 'staying within the 10-90th percentiles' can be removed, you already say 'within the climatological variability' , that's enough.* Done.

*Line 27: 'temporal shifts up to one day' - not sure what this means. Temporal shifts in what?*

➢ We dropped the term.

*'Potential future studies' (line 22) and 'future analyses' (line 31) is redundant. Would suggest removing the first mention although I personally think the last sentence can be removed. It is what we call in the US a 'motherhood and apple pie' statement - something so general, bland and well-accepted that it doesn't need mention.* Done.

*Introduction:*
*Line 40: 'the tendency' -> 'the jet stream's tendency'*

➢ Thank you for the suggestion. It would make the sentence clearer. However, we condensed the discussion about the jet stream responses and therefore dropped this term.

*Line 45: "the frequency of meridional transport': what does this mean? Transport of what? Frequency measured how? At what altitude ?*        Has been dropped.

*Line 50: 'meridional transport' of what? Probably also need to relate transport to a budget, e.g. if the meridional temperature gradient decreases while the tropics warm up, the meridional transport of temperature could increase but still have less impact on the high-latitude temperature budget than when the gradient was stronger.*        Has been dropped.

*I am not an expert on the larger-scale perspective either, and a larger lack of confidence is helping to drive Arctic research. I would suggest attempting to communicate that 'questions remain' as opposed to the current more assertive comments, e.g, 'xxx hypothesize that weaker, meandering jet streams will result in….' Then end the paragraph with something on 'most research has relied on global models unable to resolve mesoscale features well, leaving open questions on….'*

➢ Agreed. Due to the open questions regarding this topic, we replaced the discussion by: "Especially the role of linkages between Arctic and mid-latitudes is still under debate. " (line 30)

*Perhaps the more senior authors can step up here, to help polish the introduction a bit further? The first paragraph on p.4 is now confusing because the authors communicate instead a focus on MCAOs as opposed to the ARs of the previous paragraph. I think the paragraph may just need an additional connecting sentence to make the segue.*

➢ We reformulated the paragraphs to make our motivation more clear.

*p. 9 top of page: please recognize somewhere that comparing a recent time period from ERA5 will assimilated soundings to earlier years with less data assimilation introduces a form of bias.*

➢ We now discuss the potential bias caused by the different amount and type of assimilated data in the summary: "However, it must be noted that the ERA5 climatology may have systematic differences in 2022 compared to previous years as measurements from HALO-(AC)³ dropsondes were assimilated." (lines 347-348)

*p. 13 line 323: please remove the presumptive description 'Omega block' here . Such terms get thrown around colloquially in weather discussions but unless they are a major focus of the writing they should not be used. Just describe what you have to say.*

➢ We understand your point that using specific synoptic terms may lead to confusion among some readers. However, as the blocking situation was relevant for the Atmospheric Rivers' path through the North Atlantic (Woods et al. 2013), we would like to keep describing it as 'blocking'. However, as a compromise, we deleted the term 'Omega block'.

*p. 17, top of page: sorry, not okay to just throw out the term 'Shapiro-Keyser' without describing what it is. Again it may be something that was tossed around within the AC3 weather discussions by a small group of people, but the audience for this manuscript can be anticipated to be larger, some of whom are likely to think about cyclones differently from the AC3 weather forecasting team. Currently what this writing communicates is an author list hiding behind a shorthand they can't explain, which then suggests that perhaps they also don't understand. Give this another try and also try not to rely on this term quite so much, it still appears in many places.*

➤ We agree that the type of cyclone is not really relevant to our analysis and, therefore, dropped the term.

*Line 423: what makes air 'aged' ? I find this an odd term. Does it matter?*

➤ We dropped the term to avoid confusion.

*p. 20-21, lines 445-450: again this reliance on a short-hand - Polar Low - that just comes across as slang. How is this mesoscale cycle a 'Polar Low' as opposed to simply a mesoscale cyclone? Do circulation features need to meet a quantitative criteria? This is all discussed in more depth in section 4.3, I would suggest just calling it a mesoscale cyclone at this stage in the manuscript.*

➤ We dropped the discussion about the Polar Low to enhance the focus of the manuscript on the main meridional atmospheric transport events.

*p. 24 line 526: again this strange habit of calling air 'aged'. What the heck does this mean? Aerosol can be 'aged' but what distinguishes 'aged' air from 'fresh' air?*

➤ We dropped the term.

*p. 28 line 609: nice to see some comparison between ERA5 and the dropsondes, showing that even with the data assimilation ERA5 isn't getting the full moisture flux. This is worth mentioning I feel.* Thank you for your comment.

*Section 4.3: so fulfilling 5 of the 7 (or 4 out of 6) conditions put forth by Radowan qualifies a cyclone as a Polar Low? I would suggest stating that explicitly if so. As written the authors are appearing to presume the cyclone is a polar low and then just characterize it using Radowan's criteria. The discussion between lines 709-715 is nice and I'm fine with the system being called a Polar Low but would suggest rewriting the language so that it is not so initially presumptive.* Has been dropped.

---

## Author Response (AR3)

**University of Cologne**

[Figure]

Cologne, 14 May 2024

**Author's response:**

**Institute for Geophysics**

**and Meteorology**

**MSc. Andreas Walbröl**

Phone: +49 (0)221 470-3678

E-mail: a.walbroel@uni-koeln.de

Pohligstr. 3

50969 Cologne

Germany

Dear ACP editorial team,

we have revised the manuscript "Unusual atmospheric and sea ice conditions in the North Atlantic sector of the Arctic during the HALO-(AC)³ campaign" according to the comments of the reviewers (see line by line response below). As suggested by the reviewer, we changed the title with the intention to highlight the contrast between the warm and cold conditions: "Contrasting extreme warm and long-lasting cold air anomalies in the North Atlantic sector of the Arctic during the HALO-(AC)³ campaign".

We also analyzed the agreement of ERA5 with the dropsonde measurements. The results (integrated as a new figure) show an excellent agreement in accordance with the assimilation as more than 50 % of the dropsondes into ERA5. As suggested, we added a figure on radar measurements showing the bright band during one of the Atmospheric River (AR) events to stress that liquid precipitation was indeed observed over sea ice at this time of the year. An Arctic-wide synoptic map showing the average pressure and geopotential height conditions over the warm and the cold period has been added to the appendix to support the chapters focussing on AR events and marine cold air outbreaks.

Neither this manuscript or substantial parts of it have been published elsewhere in English or any other language, nor is it presently under consideration for publication by any other journal.

Sincerely,

Andreas Walbröl

**Reviewer reply:**

We thank the reviewer for the supportive review of the manuscript. The comments helped us to refine some of the analyses by adding supportive visualizations. Below, we repeat the reviewer's comments in black and write our responses in blue. The line numbers in the line-by-line responses are valid for the revised manuscript.

*The manuscript flows well now and I like the figures. I remain puzzled why a manuscript focused on the HAO-AC3 field campaign contains absolutely no analysis of the HALO data. We see the dropsonde locations in Fig. 1, then on line 175 are told how scarce the measurements are and diverted to reanalysis. Why not use all the dropsondes to assess if ERA5 indeed possesses a moisture deficit, for example? It seems like a relatively easy assessment and something society funds field campaigns to do. So I feel that I am missing something - maybe the ERA5 assessment using HALO dropsonde data is a dedicated project someone else is undertaking?*

➢ We were reluctant to show this as more than half of the sondes were assimilated into ERA5. However, we agree with the reviewer that the comparison is important for the reader. Therefore, we added a figure in Sect. 3.2, which shows excellent agreement of dropsondes and ERA5.

*I like the focus on the extremely warm AR event, it is something to organize the paper around and the authors have done that by and large. I wonder if AR1 and AR2 couldn't simply be grouped together - although the Guan/Waliser discrimination identifies two, I don't see the meteorology as being different and in practice they are grouped together. Just calling AR1+AR2, AR1, could simplify the writing.*

➢ We understand the idea of merging the two AR events for simplification. However, as they were connected to different pressure systems, we prefer to keep them separated.

*The new title is an improvement in that it sounds like less of a report, but 'Unusual' sounds vague to me - why not try to summarize what is the unusual condition? I believe that is the AR, as the MCAOs aren't all that extreme - the domain is still warmer in the mean after all. Thus I suggest the authors reconsider the title (or defend their current choice further), to focus more on the anomalous warmth of the conditions sampled during HALO-AC3.*

➢ Thank you for the comment on the title. It motivated us to change it to " Contrasting extreme warm and long-lasting cold air anomalies in the North Atlantic sector of the Arctic during the HALO-(AC)³ campaign", which now stresses the contrast of the warm and cold conditions that both occurred during the campaign.

*Another recommendation is to highlight the large-scale circulation features further. An SLP high over Scandinavia and SLP low over Greenland flip, around March 20. A map of the entire Arctic, showing the baroclinic wave structure for the 2 regimes, would be of interest. Of further interest of course is whether the atmospheric blocking is more pronounced than climatology, to help explain the AR, if that can be done.*

➢ Thank you for the suggestion. We added synoptic maps averaged over the warm and the cold period in the appendix to give a wider perspective for interested people. The plot also supports the atmospheric blocking remark in chapter 4 and can explain the longevity of the cold air outbreak conditions in chapter 5. Regarding the last point: The unusual strength of the blocking can be, although implicitly, inferred from the strength of the pressure anomalies. Such a statement has been integrated into the discussion. A detailed analysis of the blocking strength, for which no clear definition exists, would be beyond the scope of the manuscript.

*Line 175 'scarcity of measurements' is a strange phrase here. Can this be stated in a more nuanced fashion, for example by referencing some of the genuine measurement papers?*

➢ We speficied this statement and added that there are no permanent radiosonde stations north of 82.5°N (Rinke et al., 2019) to support the scarcity of measurements in the Arctic: "Given the scarcity of measurements, e.g., no radiosonde stations north of 82.5° N (Rinke et al., 2019), reanalysis data are used to characterize the conditions over the whole study domain. " (lines 175-176)

*Line 187: is the typical Arctic SLP established elsewhere in the manuscript?*

➢ Good point. We have not yet mentioned what the typical Arctic sea level pressure is for this season. Thus, we have added a sentence that refers to a climatological pressure field of the Arctic: "The high MSLP of the central Arctic agrees well with the climatological pressure shown for April in Fig. 4.8 of Serreze and Barry (2014)." (lines 201-202)

*Line 14 in abstract: mention the maxima values are in the ERA5 record (and not measurements)*

➢ We made it more clear that these maxima were indeed measurements: "These warm and moist air masses caused the highest measured 2 m temperatures (5.5 °C) and daily precipitation rates (42 mm day-1) at Ny-Alesund for March since the beginning of the record (1993)." (lines 13-14)

*Line 228: can ARs can start in the arctic? That seems strange.*

➢ The formulation was indeed a bit imprecise. To be more precise, we changed it to: "The number of strong AR events decreases meridionally (Fig. 7) because of two effects: Firstly, along their northward propagation, ARs generally decline in intensity. Secondly, within the Arctic circle, the likelihood of new AR formation strongly decays, as the moisture uptake from the ocean is substantially reduced (Papritz et al., 2022)." (lines 243-246)

*Line 252: why not treat the reader to a radar image of the brightband here? Also include mention of whatever retrieved precipitation values are available from the HALO datasets, I would think some preliminary values at least are available, after 2 years?*

➢ We agree that it is a good idea to visualize the bright band. Therefore, we added a plot showing the bright band, which is best captured in the radar linear depolarization ratio

observations: "Liquid precipitation at high latitudes over sea ice was also observed by the cloud radar onboard HALO as we detected a distinct bright band in the linear depolarization ratio at about 0.5-1 km height on 13 March (Fig. 10)." (lines 266-267). We often observed high LDR values close to the surface, which suggests the presence of tumbling particles. As high LDR values indicate the presence of complex melting particles, no simple relation between reflectivity and rain rate can be applied.

*Line 281: a 19 day mcao cant just be one strong cyclone passage, and it's also hard to see the 19day extent in Fig. 4. An Arctic wide 500 hPa Geopotential height analysis, averaged over the warm and cold periods, might help bring this out.*

➢ We agree that the average pressure over the cold period can illustrate the potential longevity of MCAO conditions. However, as we did not want to disrupt the current flow of the paper, we only added the figure to the appendix and related the pressure constellation to the longevity of MCAO conditions in one sentence in the main text: "The longevity of the MCAO conditions can be explained by the persistence of low pressure over Scandinavia, the Barents Sea and Russia, and high pressure over Greenland and the central Arctic during the cold period (Fig. B1b in Appendix B)." (lines 298-300)
We also rearranged the paragraph to make it more clear that this 19 day MCAO event consists of multiple waves of changing MCAO conditions related to the presence of cyclones near Svalbard.

*Overall the figures are nice, Fig 7, 9, 10 in particular . Within Fig. 4, the white contour lines indicating the SLP can be a little faint. Suggest thickening a few key lines, and including 'L' and 'H' to indicate pronounced closed circulations.* Done.

---

## Author Response (AR4)

**University of Cologne**

[Figure]

Cologne, 25 May 2024

**Institute for Geophysics**

**and Meteorology**

**MSc. Andreas Walbröl**

Phone: +49 (0)221 470-3678

E-mail: a.walbroel@uni-koeln.de

Pohligstr. 3

50969 Cologne

Germany

**Author's response:**

Dear ACP editorial team,

thank you for accepting our manuscript "Contrasting extremely warm and long-lasting cold air anomalies in the North Atlantic sector of the Arctic during the HALO-(AC)³ campaign". As final changes, we have now also added the reference to the codes for the reproduction of the figures in the *Code and data availability* section. Could this manuscript please be added to the HALO-(AC)³ special issue (https://acp.copernicus.org/articles/special_issue1272.html)?

On behalf of all authors and with best regards,

Andreas Walbröl